# LancBiO: Dynamic Lanczos-aided Bilevel Optimization via Krylov Subspace

**Yan Yang**
Academy of Mathematics and Systems Science
Chinese Academy of Sciences
University of Chinese Academy of Sciences
yangyan@amss.ac.cn

**Bin Gao**[*] **& Ya-xiang Yuan**
Academy of Mathematics and Systems Science
Chinese Academy of Sciences
{gaobin,yyx}@lsec.cc.ac.cn

## Abstract

Bilevel optimization, with broad applications in machine learning, has an intricate hierarchical structure. Gradient-based methods have emerged as a common approach to large-scale bilevel problems. However, the computation of the hyper-gradient, which involves a Hessian inverse vector product, confines the efficiency and is regarded as a bottleneck. To circumvent the inverse, we construct a sequence of low-dimensional approximate Krylov subspaces with the aid of the Lanczos process. As a result, the constructed subspace is able to dynamically and incrementally approximate the Hessian inverse vector product with less effort and thus leads to a favorable estimate of the hyper-gradient. Moreover, we propose a provable subspace-based framework for bilevel problems where one central step is to solve a small-size tridiagonal linear system. To the best of our knowledge, this is the first time that subspace techniques are incorporated into bilevel optimization. This successful trial not only enjoys $\mathcal{O}(\epsilon^{-1})$ convergence rate but also demonstrates efficiency in a synthetic problem and two deep learning tasks.

## 1 Introduction

Bilevel optimization, in which upper-level and lower-level problems are nested with each other, mirrors a multitude of applications, e.g., game theory (Stackelberg, 1952), hyper-parameter selection (Ye et al., 2023), data poisoning (Liu et al., 2024), meta-learning (Bertinetto et al., 2018), neural architecture search (Liu et al., 2018; Wang et al., 2022), adversarial training (Wang et al., 2021), reinforcement learning (Chakraborty et al., 2024; Thoma et al., 2024; Yang et al., 2025), computer vision (Liu et al., 2021a). In this paper, we consider the bilevel problem:

$$
\begin{aligned}
\min_{x \in \mathbb{R}^{d_x}} \quad & \varphi(x) := f\left(x, y^*(x)\right) \\
\text{s.t.} \quad & y^*(x) \in \arg\min_{y \in \mathbb{R}^{d_y}} g(x, y),
\end{aligned}
\tag{1}
$$

where the upper-level function $f$ and the lower-level function $g$ are defined on $\mathbb{R}^{d_x} \times \mathbb{R}^{d_y}$. $\varphi$ is called the *hyper-objective*, and the gradient of $\varphi(x)$ is referred to as the *hyper-gradient* (Pedregosa, 2016; Grazzi et al., 2020; Yang et al., 2023; Chen et al., 2024) if it exists. In contrast to standard single-level optimization problems, bilevel optimization is inherently challenging due to its intertwined structure. Specifically, the formulation (1) underscores the crucial role of the lower-level solution $y^*(x)$ in each update of $x$.

One of the focal points in recent bilevel methods has shifted towards nonconvex upper-level problems coupled with strongly convex lower-level problems (Ghadimi and Wang, 2018; Ji et al., 2021; Dagréou et al., 2022; Li et al., 2022; Hong et al., 2023; Hu et al., 2024). This configuration ensures that $y^*(x)$ is a single-valued function of $x$, i.e., $y^*(x) = \arg\min_{y \in \mathbb{R}^{d_y}} g(x, y)$. Subsequently, $\nabla\varphi(x)$ can be computed via the implicit function theorem following (Ghadimi and Wang, 2018),

$$
\nabla\varphi(x) = \nabla_x f\left(x, y^*(x)\right) - \nabla^2_{xy} g\left(x, y^*(x)\right) \left[\nabla^2_{yy} g\left(x, y^*(x)\right)\right]^{-1} \nabla_y f\left(x, y^*(x)\right).
\tag{2}
$$

---

[*]Corresponding author.

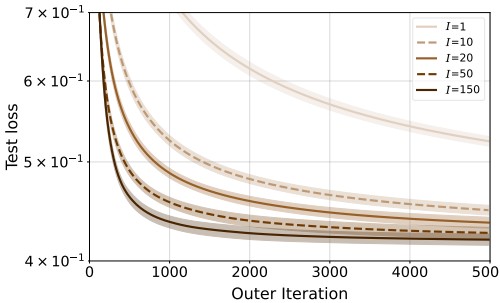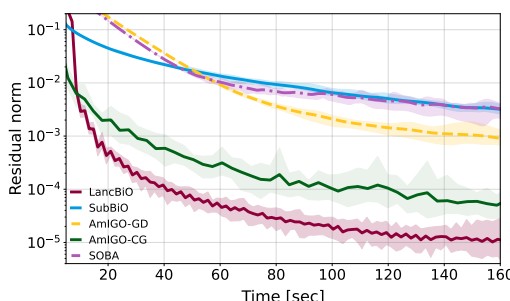

Figure 1: **Left:** test loss for the method stocBiO (Ji et al., 2021) with different inner iterations $I$ to approximate the Hessian inverse vector product; **Right:** Estimation error of the Hessian inverse vector product in hyper-data cleaning task with corruption rate $0.5$ for different methods: LancBiO and SubBiO (ours), AmIGO (Arbel and Mairal, 2022), and SOBA (Dagréou et al., 2022).

Gradient methods based on the hyper-gradient, $x_{k+1} = x_k - \lambda \nabla \varphi(x_k)$, are known as the approximate implicit differentiation (AID) based methods (Ji et al., 2021; Liu et al., 2023b; Huang, 2024). However, the computation of the hyper-gradient (2) suffers from two pains: 1) solving the lower-level problem to obtain $y^*(x)$; 2) assembling the *Hessian inverse vector product*

$$v^*(x) := \left[ \nabla^2_{yy} g\left(x, y^*\left(x\right)\right) \right]^{-1} \nabla_y f\left(x, y^*\left(x\right)\right), \tag{3}$$

or equivalently, solving a large linear system in terms of $v$,

$$\nabla^2_{yy} g(x, y^*(x)) v = \nabla_y f(x, y^*(x)). \tag{4}$$

To this end, it is beneficial to adopt a few inner iterations to approximate $y^*(x)$ and $v^*(x)$ within each outer iteration (i.e., the update of $x$). Note that the approximation accuracy of $v^*(x)$ is crucial for AID-based methods; see (Ji et al., 2022; Li et al., 2022). Specifically, the left of Figure 1 confirms that the more inner iterations, the higher quality of the estimate of $v^*$, and the more enhanced descent of the objective function within the same number of outer iterations.

**Approximation:** Existing efforts are dedicated to approximating $v^*$ in different fashions by regulating the number of inner iterations, e.g., the Neumann series approximation (Ghadimi and Wang, 2018; Ji et al., 2021) for the inverse, gradient descent (Arbel and Mairal, 2022; Dagréou et al., 2022) and conjugate gradient descent (Pedregosa, 2016; Yang et al., 2023) for the linear system.

**Amortization:** Moreover, there are studies aimed at amortizing the cost of approximation through outer iterations. These methods include using the inner estimate from the previous outer iteration as a warm start for the current outer iteration (Ji et al., 2021; Arbel and Mairal, 2022; Dagréou et al., 2022; Ji et al., 2022; Li et al., 2022; Xiao et al., 2023), or employing a refined step size control (Hong et al., 2023).

Subspace techniques, widely adopted in nonlinear optimization (Yuan, 2014), approximately solve large-scale problems in lower-dimensional subspaces, which not only reduce the computational cost significantly but also enjoy favorable theoretical properties as in full space models. Taking into account the above two principles, it is reasonable to consider subspace techniques in bilevel optimization. Specifically, we can efficiently amortize the construction of low-dimensional subspaces and sequentially solve linear systems (4) in these subspaces to approximate $v^*$ accurately.

## 1.1 CONTRIBUTIONS

In this paper, taking advantage of the Krylov subspace and the Lanczos process, we develop an innovative subspace-based framework—LancBiO, which features an efficient and accurate approximation of the Hessian inverse vector product $v^*$ in the hyper-gradient—for bilevel optimization. The main contributions are summarized as follows.

Firstly, we build up a dynamic process for constructing low-dimensional subspaces that are tailored from the Krylov subspace for bilevel optimization. This process effectively reduces the large-scale

subproblem (4) to the small-size tridiagonal linear system, which draws on the spirit of the Lanczos process. To the best of our knowledge, this is the first time that the subspace technique is leveraged in bilevel optimization.

Moreover, the constructed subspaces enable us to dynamically and incrementally approximate $v^*$ across outer iterations, thereby achieving an enhanced estimate of the hyper-gradient; the right of Figure 1 illustrates that the proposed LancBiO reaches the best estimation error for $v^*$. Hence, we provide a new perspective for approximating the Hessian inverse vector product in bilevel optimization. Specifically, the number of Hessian-vector products averages at $(1 + 1/m)$ per outer iteration with the subspace dimension $m$, which is favorably comparable with the existing methods.

Finally, we offer analysis to circumvent the instability in the process of approximating subspaces, with the result that LancBiO can profit from the benign properties of the Krylov subspace. We prove that the proposed method LancBiO is globally convergent with the convergence rate $\mathcal{O}(\epsilon^{-1})$. In addition, the efficiency of LancBiO is validated by a synthetic problem and two deep learning tasks.

## 1.2 RELATED WORK

A detailed introduction to bilevel optimization methods can be found in Appendix A.

**Krylov subspace methods:** Subspace techniques have gained significant recognition in the realm of numerical linear algebra (Parlett, 1998; Saad, 2011; Golub and Van Loan, 2013) and nonlinear optimization (Yuan, 2014; Liu et al., 2021c). Specifically, numerous optimization methods utilized subspace techniques to improve efficiency, including acceleration technique (Li et al., 2020), diagonal preconditioning (Gao et al., 2023), and derivative-free optimization methods (Cartis and Roberts, 2023). Krylov subspace (Krylov, 1931), due to its special structure,

$$\mathcal{K}_N(A, b) := \mathrm{span}\left\{b, Ab, A^2b, \ldots, A^{N-1}b\right\}$$

with the dimension $N$ for a matrix $A$ and a vector $b$, exhibits advantageous properties in convex quadratic optimization (Nesterov et al., 2018), eigenvalue computation (Kuczyński and Woźniakowski, 1992), and regularized nonconvex quadratic problems (Carmon and Duchi, 2018). Krylov subspace has been widely considered in large-scale optimization such as trust region methods (Gould et al., 1999), trace maximization problems (Liu et al., 2013), and cubic Newton methods (Cartis et al., 2011; Jiang et al., 2024). Lanczos process (Lanczos, 1950) is an orthogonal projection method onto the Krylov subspace, which reduces a dense symmetric matrix to a tridiagonal form. Details of the Krylov subspace and the Lanczos process are summarized in Appendix B.

**Approximation of the Hessian inverse vector product :** It is cumbersome to compute the Hessian inverse vector product in bilevel optimization. To bypass it, several strategies implemented through inner iterations were proposed, e.g., the Neumann series approximation (Ghadimi and Wang, 2018; Ji et al., 2021), gradient descent (Arbel and Mairal, 2022; Dagréou et al., 2022), and conjugate gradient descent (Pedregosa, 2016; Arbel and Mairal, 2022; Yang et al., 2023). Alternatively, the previous information was exploited in (Ji et al., 2021; Arbel and Mairal, 2022) as a warm start for outer iterations; Ramzi et al. (2022) suggested approximating the Hessian inverse in the manner of quasi-Newton; Dagréou et al. (2022) and Li et al. (2022) proposed the frameworks without inner iterations to approximate the Hessian inverse vector product.

## 2 SUBSPACE-BASED ALGORITHMS

In this section, we dive into the development of bilevel optimization algorithms for solving (1), which dynamically construct subspaces to approximate the Hessian inverse vector product.

The (hyper-)gradient descent method carries out the $k$-th outer iteration as $x_{k+1} = x_k - \lambda \nabla \varphi(x_k)$, where the hyper-gradient is exactly computed by $\nabla \varphi(x_k) = \nabla_x f(x_k, y_k^*) - \nabla_{xy}^2 g(x_k, y_k^*) v_k^*$ with $y_k^* := y^*(x_k)$ and $v_k^* := v^*(x_k)$ defined in (3). In view of the computational intricacy of $y_k^*$ and $v_k^*$, it is commonly concerned with the following estimator for the hyper-gradient

$$\widetilde{\nabla}\varphi(x_k, y_k, v_k) := \nabla_x f(x_k, y_k) - \nabla_{xy}^2 g(x_k, y_k) v_k, \tag{5}$$

where $y_k$ is an approximation of $y_k^*$. Denote $A_k = \nabla_{yy}^2 g(x_k, y_k)$ and $b_k = \nabla_y f(x_k, y_k)$. The vector $v_k$ serves as the (approximate) solution of the quadratic optimization problem

$$\min_{v \in \mathcal{S}_k} \frac{1}{2} v^\top A_k v - v^\top b_k, \tag{6}$$

where $\mathcal{S}_k$ is the full space $\mathbb{R}^{d_y}$ and the exact solution is $A_k^{-1} b_k$. Subsequently, in order to reduce the computational cost, it is natural to ask:

> *Can we construct a low-dimensional subspace $\mathcal{S}_k$ such that*
> *the solution of the quadratic problem* (6) *satisfactorily approximates $A_k^{-1} b_k$?*

General subspace constructions introduced in the existing subspace methods (Yuan, 2014; Liu et al., 2021c) are not straightforward and not exploited in the bilevel setting, rendering the exploration of appropriate subspaces challenging. In the following subsections, we construct approximate Krylov subspaces and propose an elaborate subspace-based framework for bilevel problems.

## 2.1 WHY KRYLOV SUBSPACE: THE SUBBIO ALGORITHM

In light of the Neumann series for a suitable $\eta \in \mathbb{R}$, $A^{-1} b = \eta \sum_{i=0}^\infty (I - \eta A)^i b$, it is observed from Appendix B that $A^{-1} b$ belongs to a Krylov subspace for some $N > 0$, i.e.,

$$A^{-1} b \in \mathcal{K}_N(A, b) = \mathcal{K}_N(I - \eta A, b).$$

Hence, it is reasonable to consider a Krylov subspace for the construction of $\mathcal{S}_k$.

Given a constant $n \ll N$, we consider an approximation of $A^{-1} b$ in a lower-dimensional Krylov subspace $\mathcal{K}_n(A, b)$, i.e., $v_n \in \mathcal{K}_n(A, b) = \mathcal{K}_n(I - \eta A, b)$ and $v_n = \sum_{i=0}^{n-1} c_i (I - \eta A)^i b \approx A^{-1} b$. Note that the approximation $v_n$ is composed of the set $\{(I - \eta A)^i b\}_{i=0}^{n-1}$ in the sense of the Neumann series. Moreover, we observe that $(I - \eta A) v_n \in \mathcal{K}_{n+1}(A, b)$ and hence we can recursively choose

$$v_{n+1} \in \mathcal{S}_{n+1} := \text{span}\{b, (I - \eta A) v_n\} \subseteq \mathcal{K}_{n+1}(A, b)$$

since the subspace $\text{span}\{b, (I - \eta A) v_n\}$ includes the information of the increased set $\{(I - \eta A)^i b\}_{i=0}^n$. In summary, we can construct a sequence of two-dimensional subspaces $\{\mathcal{S}_n\}$ that implicitly filters information from the Krylov subspaces. The rationale for this procedure can be illustrated in Figure 2.

In the context of bilevel optimization, we seek the best solution $v_k$ to the subproblem (6) in the subspace

$$\mathcal{S}_k = \text{span}\{b_k, (I - \eta A_k) v_{k-1}\}. \tag{7}$$

Repeating the procedure is capable of dynamically approximating the Hessian inverse vector product, i.e., $v_k$ approximates $A_k^{-1} b_k$. The Krylov Subspace-aided Bilevel Optimization algorithm (SubBiO) is listed in Algorithm 1.

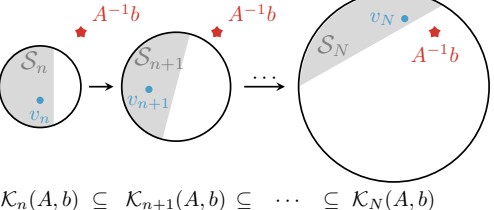

$$\mathcal{K}_n(A, b) \subseteq \mathcal{K}_{n+1}(A, b) \subseteq \cdots \subseteq \mathcal{K}_N(A, b)$$

Figure 2: Illustration of approximating $A^{-1} b \in \mathcal{K}_N(A, b)$ by $v_n$ in the two-dimensional subspace $\mathcal{S}_n \subseteq \mathcal{K}_n(A, b)$.

---

**Algorithm 1** SubBiO

**Input:** iteration threshold $K$, step sizes $\theta, \lambda, \eta$, initialization $x_1, y_1, v_0$
1: **for** $k = 1, 2, \ldots, K$ **do**
2:     $A_k = \nabla_{yy}^2 g(x_k, y_k)$, $b_k = \nabla_y f(x_k, y_k)$
3:     $\mathcal{S}_k = \text{span}\{b_k, (I - \eta A_k) v_{k-1}\}$
4:     $v_k = \arg\min_{v \in \mathcal{S}_k} \frac{1}{2} v^\top A_k v - b_k^\top v$
5:     $x_{k+1} = x_k - \lambda \left( \nabla_x f(x_k, y_k) - \nabla_{xy}^2 g(x_k, y_k) v_k \right)$
6:     $y_{k+1} = y_k - \theta \nabla_y g(x_{k+1}, y_k)$
7: **end for**
**Output:** $(x_{K+1}, y_{K+1})$

---

## 2.2 WHY DYNAMIC LANCZOS: THE LANCBIO FRAMEWORK

Notice that the subproblem in SubBiO (Algorithm 1) can be equivalently reduced to

$$\min_{z \in \mathbb{R}^2} \frac{1}{2} z^\top (S_k^\top A_k S_k) z - b_k^\top S_k z,$$

where $S_k := [b_k \ (I - \eta A_k) v_{k-1}] \in \mathbb{R}^{d_y \times 2}$. The solution $z^* \in \mathbb{R}^2$ results in $v_k = S_k z^*$. It is a two-dimensional subproblem, whereas computing the projection $S_k^\top A_k S_k$ requires two Hessian-vector products, which dominate the cost of the subproblem. Therefore, it is crucial to reduce or amortize the projection cost while preserving the advantages of the Krylov subspace. To this end, we find that the Lanczos process (Appendix B) provides an enlightening way (Lanczos, 1950; Saad, 2011; Golub and Van Loan, 2013) since it allows for the construction of a Krylov subspace and maintaining a tridiagonal matrix as the projection matrix, which significantly reduces the computational cost.

In bilevel optimization, since the quadratic problem (6) evolves through outer iterations, it is difficult to leverage the Lanczos process to amortize the projection cost while updating variables as in SubBiO. Specifically, the Lanczos process is inherently unstable (Paige, 1980), and thus the accumulative difference among $\{A_k\}$ and $\{b_k\}$ will make the Lanczos process invalid.

In order to address the above difficulties, we propose a restart mechanism to guarantee the benign behavior of approximating the Krylov subspace and consider solving residual systems to employ the historical information. In summary, we propose a dynamic Lanczos-aided Bilevel Optimization framework, LancBiO, which is listed in Algorithm 2. The only difference between LancBiO and SubBiO is solving the subproblem (line 3-4 in Algorithm 1).

If we adapt the standard Lanczos process for tridiagonalizing $A_k$, by starting from $q_1 = b_1 / \|b_1\|$, $q_0 = \mathbf{0}$, $\beta_1 = 0$ and using the dynamic matrices $A_j$ for $j = 1, 2, \ldots, k$, the process is as follows,

$$u_j = A_j q_j - \beta_j q_{j-1},$$
$$\alpha_j = q_j^\top u_j,$$
$$\omega_j = u_j - \alpha_j q_j,$$
$$\beta_{j+1} = \|\omega_j\|,$$
$$q_{j+1} = \omega_j / \beta_{j+1},$$

and $T_k$ is a tridiagonal matrix recursively computed from

$$T_j = \begin{pmatrix} & & \vdots & \mathbf{0} \\ & T_{j-1} & \vdots & \\ & & \vdots & \beta_j \\ \cdots & \cdots & \vdots & \cdots \\ \mathbf{0} & \vdots & \beta_j & \alpha_j \end{pmatrix}.$$

**Algorithm 2** LancBiO

**Input:** iteration threshold $K$, step sizes $\theta, \lambda$, initialization $x_1, y_1, v_1$, initial correction $\Delta v_0 = 0$, subspace dimension $m$, initial epoch $h = -1$
1: **for** $k = 1, 2, \ldots, K$ **do**
2:      $A_k = \nabla^2_{yy} g(x_k, y_k)$, $b_k = \nabla_y f(x_k, y_k)$
3:      **if** $(k \bmod m) = 1$ **then**
4:          $h = h + 1$
5:          $\bar{v}_h = v_k$
6:          $w_h = A_k \bar{v}_h$
7:          $Q_{k-1} = (b_k - w_h) / \|b_k - w_h\|$
8:          $T_{k-1} = $ Empty Matrix
9:          $\beta_k = 0$
10:      **end if**
11:      $(T_k, Q_k, \beta_{k+1}) = $ DLanczos$(T_{k-1}, Q_{k-1}, A_k, \beta_k)$
12:      $r_k = b_k - w_h$
13:      $\Delta v_k = Q_k (T_k)^{-1} Q_k^\top r_k$
14:      $v_k = \bar{v}_h + \Delta v_k$
15:      $x_{k+1} = x_k - \lambda \left( \nabla_x f(x_k, y_k) - \nabla^2_{xy} g(x_k, y_k) v_k \right)$
16:      $y_{k+1} = y_k - \theta \nabla_y g(x_{k+1}, y_k)$
17: **end for**
**Output:** $(x_{K+1}, y_{K+1})$

**Restart mechanism:** In contrast to SubBiO, we construct the subspace

$$\mathcal{S}_k = \text{span}(Q_k) = \text{span}([q_1, \ldots, q_k]).$$

Consequently, $Q_k$ approximates the basis of the true Krylov subspace $\mathcal{K}_k(A_k, b_k)$, and $T_k$ approximates the projection of $A_k$ onto it, i.e., $T_k \approx Q_k^\top A_k Q_k$. However, $Q_k$ will lose the orthogonality due to the evolution of $A_j$, and $T_k$ will deviate from the true projection. Based on this observation, we restart the subspace spanned by the matrix $Q$ for every $m$ outer iterations (line 3 to line 10 in Algorithm 2). The restart mechanism allows us to mitigate the accumulation of the difference among $\{A_1, \ldots, A_k\}$, and hence, we can maintain a more reliable basis matrix to approximate Krylov subspaces. The above dynamic Lanczos subroutine, DLanczos, is summarized in Appendix C.

**Residual minimization:** The preceding discussion reveals that the dimension of the subspace $\mathcal{S}_k$ should be moderate to retain the reliability of the dynamic process. However, the limited dimension stagnates the approximation of the subproblem (6) to the full space problem. Therefore, instead of directly solving the quadratic subproblem (6) in the subspace, we intend to find $v \in \mathcal{S}_k = \text{span}(Q_k)$ such that $A_k v \approx b_k$. To this end, after going through $m$ outer iterations, we denote the current approximation by $\bar{v}$. In the subsequent outer iterations, we concentrate on a linear system with a residual in the form

$$A_k \Delta v = b_k - A_k \bar{v}, \qquad \Delta v \in \mathcal{S}_k. \tag{8}$$

Specifically, we inexactly solve the minimal residual subproblem

$$\min_{\Delta v \in \mathcal{S}_k} \left\| (b_k - A_k \bar{v}) - A_k \Delta v \right\|^2 \tag{9}$$

and use the solution $\Delta v_k$ as a correction to $\bar{v}$ (line 12 to line 14 in Algorithm 2), $v_k = \bar{v} + \Delta v_k$.

Consequently, taking into account the two strategies above, we illuminate the framework LancBiO in Figure 3. It is structured into epochs, with each epoch built by $m$ outer iterations. Notably, each epoch restarts by incrementally constructing from a one-dimensional space $Q_1$ to an $m$-dimensional space $Q_m$, aiming to approximate the solution to the residual system within these subspaces. The solution $\Delta v_j$ to the subproblem serves as a correction to enhance the hyper-gradient estimation, which facilitates the $(x, y)$ updating.

The combination of the two strategies, restart mechanism and residual minimization, not only controls the dimension of the subspace but also utilizes historical information to enhance the approximation accuracy. By considering a simplified scenario, we reduce the two strategies into solving a standard linear system problem $Ax = b$ with $A$ and $b$ fixed. Note that, from the perspective of Theorem 1 in (Carmon and Duchi, 2018), the residual associated with solving a linear system in an $m$-dimensional Krylov subspace decays faster than a rate $\mathcal{O}\left(1/m^2\right)$ after each restart. In other words, the estimation error of the Hessian inverse vector product experiences a decay rate of $\mathcal{O}\left(1/m^2\right)$ after every restart, i.e.,

$$\frac{\left\| b - A v_{m(h+1)} \right\|^2}{\left\| b - A v_{mh} \right\|^2} = \frac{\left\| (b - A \bar{v}_h) - A \Delta_{m(h+1)} \right\|^2}{\left\| b - A \bar{v}_h \right\|^2}$$
$$= \mathcal{O}\left(\frac{1}{m^2}\right).$$

*Remark* 2.1. The classic Lanczos process is known for its capability to solve indefinite linear systems (Greenbaum et al., 1999). In a similar fashion, the LancBiO framework can be adapted to the bilevel problems with a nonconvex lower-level problem. Interested readers are referred to Appendix D for details.

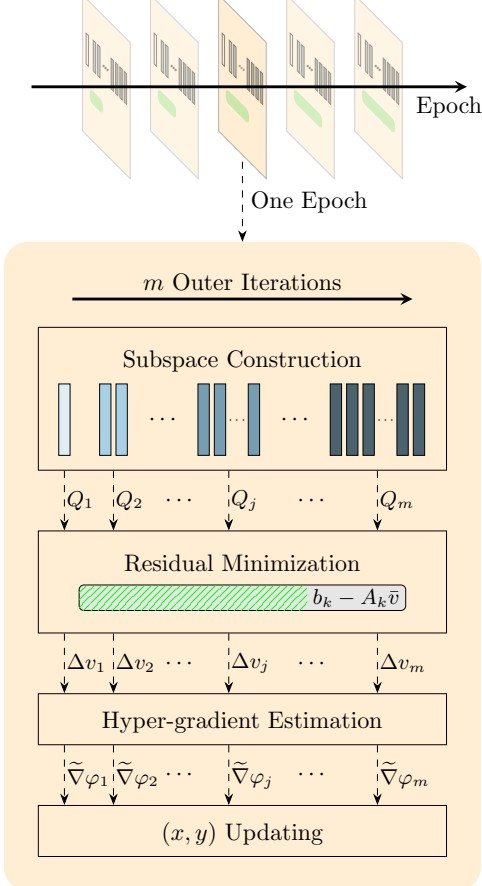

Figure 3: An overview of LancBiO.

## 2.3 RELATION TO EXISTING ALGORITHMS

The proposed SubBiO and LancBiO have intrinsic connections to the existing algorithms. Generally, in each outer iteration, methods such as BSA (Ghadimi and Wang, 2018) and TTSA (Hong et al., 2023) truncate the Neumann series at $N$, exploiting information from an $N$-dimensional Krylov subspace. In contrast, both SubBiO and LancBiO implicitly gather knowledge from a higher-dimensional Krylov subspace with less effort.

SubBiO shares similarities with SOBA (Dagréou et al., 2022) and FSLA (Li et al., 2022). The update rule for the estimator $v$ of the Hessian inverse vector product in SOBA and FSLA is

$$v_k = v_{k-1} - \eta \left( A_k v_{k-1} - b_k \right) = (I - \eta A_k) v_{k-1} + \eta b_k,$$

while the proposed SubBiO constructs a two-dimensional subspace, $\mathcal{S}_k = \text{span}\{b_k, (I - \eta A_k)v_{k-1}\}$ defined in (7). It is worth noting that the updated $v_k$ in SOBA and FSLA belongs to the subspace $\mathcal{S}_k$. Furthermore, in the sense of solving the two-dimensional subproblem (6), SubBiO selects the optimal solution $v$ in the subspace.

In addition, if the subspace dimension $m$ is set to one, LancBiO is simplified to a scenario in which one conjugate gradient (CG) step with the warm start mechanism is performed in each outer iteration, which exactly recovers the algorithm AmIGO-CG (Arbel and Mairal, 2022) with one inner iteration to the update of $v$. Alternatively, if the step size $\lambda$ in Algorithm 2 is set to 0 within each $m$-steps (i.e., only inner iterations are invoked), LancBiO reduces to the algorithm AmIGO-CG (Arbel and Mairal, 2022) with $m$ inner iterations.

## 3 THEORETICAL ANALYSIS

In this section, we provide a non-asymptotic convergence analysis for LancBiO. Firstly, we introduce some appropriate assumptions. Subsequently, to address the principal theoretical challenges, we analyze the properties and dynamics of the subspaces constructed in Section 2. Finally, we prove the global convergence of LancBiO and give the iteration complexity; the detailed proofs are provided in the appendices.

**Assumption 3.1.** The upper-level function $f$ is twice continuously differentiable. The gradients $\nabla_x f(x, y)$ and $\nabla_y f(x, y)$ are $L_{fx}$-Lipschitz and $L_{fy}$-Lipschitz, and $\|\nabla_y f(x, y^*(x))\| \leq C_{fy}$.

**Assumption 3.2.** The lower-level function $g$ is twice continuously differentiable. $\nabla_x g(x, y)$ and $\nabla_y g(x, y)$ are $L_{gx}$-Lipschitz and $L_{gy}$-Lipschitz. The derivative $\nabla^2_{xy} g(x, y)$ and the Hessian matrix $\nabla^2_{yy} g(x, y)$ are $L_{gxy}$-Lipschitz and $L_{gyy}$-Lipschitz.

**Assumption 3.3.** For any $x \in \mathbb{R}^{d_x}$, the lower-level function $g(x, \cdot)$ is $\mu_g$-strongly convex.

The Lipschitz properties of $f, g$ and the strong convexity of the lower-level problem revealed by the above assumptions are standard in bilevel optimization (Ghadimi and Wang, 2018; Chen et al., 2021; Ji et al., 2021; Khanduri et al., 2021; Arbel and Mairal, 2022; Chen et al., 2022; Dagréou et al., 2022; Li et al., 2022; Ji et al., 2022; Hong et al., 2023). These assumptions ensure the smoothness of $\varphi$ and $y^*$; see the following results (Ghadimi and Wang, 2018).

**Lemma 3.4.** *Under the Assumptions 3.2 and 3.3, $y^*(x)$ is $L_{gx}/\mu_g$ -Lipschitz continuous, i.e., for any $x_1, x_2 \in \mathbb{R}^{d_x}$, $\|y^*(x_1) - y^*(x_2)\| \leq \frac{L_{gx}}{\mu_g} \|x_1 - x_2\|$.*

**Lemma 3.5.** *Under the Assumptions 3.1, 3.2 and 3.3, the hyper-gradient $\nabla\varphi(\cdot)$ is $L_\varphi$-Lipschitz continuous, i.e., for any $x_1, x_2 \in \mathbb{R}^{d_x}$, $\|\nabla\varphi(x_1) - \nabla\varphi(x_2)\| \leq L_\varphi \|x_1 - x_2\|$, where $L_\varphi > 0$ is defined in Appendix E.*

**Assumption 3.6.** There exists a constant $C_{fx}$ so that $\|\nabla_x f(x, y)\| \leq C_{fx}$.

Assumption 3.6, commonly adopted in (Ghadimi and Wang, 2018; Ji et al., 2021; Liu et al., 2022; Kwon et al., 2023), is helpful in ensuring the stable behavior of the dynamic Lanczos process; see Section 3.1.

### 3.1 SUBSPACE PROPERTIES IN DYNAMIC LANCZOS PROCESS

In view of the inherent instability of the Lanczos process (Paige, 1980; Meurant and Strakoš, 2006) and the evolution of the Hessian $\{A_k\}$ and the gradient $\{b_k\}$ in LancBiO, the analysis of the constructed subspaces is intricate. Based on the existing work (Paige, 1976; 1980; Greenbaum, 1997), this subsection sheds light on the analysis of the subspaces and the effectiveness of the subproblem in approximating the full space problem in LancBiO.

An *epoch* is constituted of a complete $m$-step dynamic Lanczos process between two restarts, namely, after $h$ epochs, the number of outer iterations is $mh$. Given the outer iterations $k = mh + j$ for $j = 1, 2, \ldots, m$, we denote

$$\varepsilon^h_{st} := \left(1 + \frac{L_{gx}}{\mu_g}\right) \|x_{mh+s} - x_{mh+t}\| + \|y_{mh+s} - y^*_{mh+s}\|$$

for $s, t = 1, 2, \ldots, m$, and $\varepsilon_j^h := \max_{1 \le s, t \le j} \varepsilon_{st}^h$, serving as the accumulative difference. For brevity, we omit the superscript where there is no ambiguity, and we are slightly abusing of notation that at the current epoch, $\{A_{mh+j}\}$ and $\{b_{mh+j}\}$ are simplified by $\{A_j\}$ and $\{b_j\}$ for $j = 1, \ldots, m$. In addition, the approximations in the residual system (8) are simplified by $\bar{v}$ and $\bar{b} := b_1 - A_1 \bar{v}$.

The following proposition demonstrates that the dynamic subspace constructed in Algorithm 2 within an epoch is indeed an approximate Krylov subspace.

**Proposition 3.7.** *At the $j$-th step within an epoch ($j = 1, 2, \ldots, m - 1$), the subspace spanned by the matrix $Q_{j+1}$ in Algorithm 2 satisfies*

$$\mathrm{span}(Q_{j+1}) \subseteq \mathrm{span}\left\{ A_1^{a_1} A_2^{a_2} \cdots A_j^{a_j} \bar{b} \mid a_s = 0 \text{ or } 1, \ s = 1, 2, \ldots, j \right\}.$$

*Specifically, when $A_1 = A_2 = \cdots = A_j = A$ and $Q_{j+1}$ is of full rank, $\mathrm{span}(Q_{j+1}) = \mathcal{K}_{j+1}\left(A, \bar{b}\right)$.*

Denote $A^* = \nabla_{yy}^2 g(x, y^*)$ and $b^* = \nabla_y f(x, y^*)$. Notice that the dynamic Lanczos process in Algorithm 2 centers on $A_j$ instead of $A_j^*$. The subsequent lemma interprets the perturbation analysis for the dynamic Lanczos process in terms of $A_j^*$, which satisfies an approximate three-term recurrence with a perturbation term $\delta Q$.

**Lemma 3.8.** *Suppose Assumptions 3.1 to 3.3 hold. The dynamic Lanczos process in Algorithm 2 with normalized $q_1$ satisfies*

$$A_j^* Q_j = Q_j T_j + \beta_{j+1} q_{j+1} e_j^\top + \delta Q_j, \ \text{for } j = 1, 2, \ldots, m,$$

*where $Q_j = [q_1, q_2, \ldots, q_j]$, $\delta Q_j = [\delta q_1, \delta q_2, \ldots, \delta q_j]$ with $\|\delta q_j\| \le L_{gyy} \varepsilon_j$, and $T_j$ is a $j \times j$ symmetric tridiagonal matrix with diagonal elements $\{\alpha_1, \ldots, \alpha_j\}$ and subdiagonal elements $\{\beta_2, \ldots, \beta_j\}$.*

### 3.2 CONVERGENCE ANALYSIS

To guarantee the stable behavior of the dynamic process, we need the subsequent assumption.

**Assumption 3.9.** *The initialization of $y_1$ in Algorithm 2 satisfies* $\|y_1 - y_1^*\| \le \frac{\sqrt{3} \mu_g}{8(m+1)^3 L_{gyy}}$.

Similar initialization refinement is used in (Hao et al., 2024), which can be achieved by implementing several gradient descent steps for the smooth and strongly convex lower-level problem. The following lemma reveals that the dynamic process yields an improved solution for the subproblem (9).

**Lemma 3.10.** *Suppose Assumptions 3.1, 3.2, 3.3, 3.6 and 3.9 hold. Within each epoch, if we set the step size $\theta \sim \mathcal{O}(1/m)$ a constant for $y$, and the step size for $x$ as zero in the first $m_0 \sim \Omega(1)$ steps and the others as an appropriate constant $\lambda \sim \mathcal{O}(1/m^4)$, we have the following inequality,*

$$\frac{\|\bar{r}_j\|}{\|\bar{r}_0\|} \le 2\sqrt{\tilde{\kappa}(j)} \left( \frac{\sqrt{\tilde{\kappa}(j)} - 1}{\sqrt{\tilde{\kappa}(j)} + 1} \right)^j + \sqrt{j} L_{gyy} \varepsilon_j \tilde{\kappa}(j),$$

*where $\tilde{\kappa}(j) := \frac{L_{gy} + \frac{2\sqrt{3}}{3}(j+1)^3 L_{gyy} \varepsilon_j}{\mu_g - \frac{2\sqrt{3}}{3}(j+1)^3 L_{gyy} \varepsilon_j}$.*

**Theorem 3.11.** *Suppose Assumptions 3.1, 3.2, 3.3, 3.6 and 3.9 hold. Within each epoch, if we set the step size $\theta \sim \mathcal{O}(1/m)$ a constant for $y$, and the step size for $x$ as zero in the first $m_0$ steps and the others as an appropriate constant $\lambda \sim \mathcal{O}(1/m^4)$, the iterates $\{x_k\}$ generated by Algorithm 2 satisfy*

$$\frac{m}{K(m - m_0)} \sum_{\substack{k=0, \\ (k \bmod m) > m_0}}^{K} \|\nabla \varphi(x_k)\|^2 = \mathcal{O}\left( \frac{m\lambda^{-1}}{K(m - m_0)} \right),$$

*where $m_0 \sim \Omega(\log m)$ is a constant and $m$ is the subspace dimension.*

In other words, we prove that the proposed LancBiO is globally convergent, and the average norm square of the hyper-gradient $\|\nabla \varphi(x_k)\|^2$ achieves $\epsilon$ within $\mathcal{O}(\epsilon^{-1})$ outer iterations.

## 4 NUMERICAL EXPERIMENTS

In this section, we conduct experiments in the deterministic setting to empirically validate the performance of the proposed algorithms. We test on a synthetic problem and two deep learning tasks. The selection of parameters and more details of the experiments are deferred to Appendix I. We have made the code available on https://github.com/UCAS-YanYang/LancBiO.

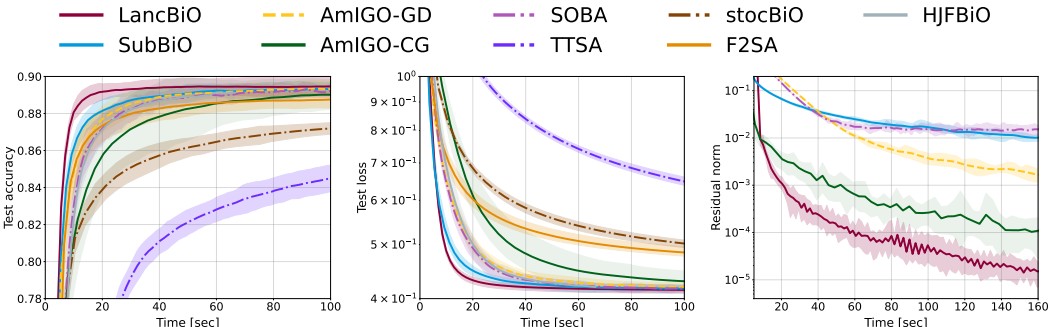

Figure 4: Comparison of the bilevel algorithms on data hyper-cleaning task when $p = 0.8$. **Left:** test accuracy; **Center:** test loss; **Right:** residual norm of the linear system, $\|A_k v_k - b_k\|$.

Table 1: Comparison of the bilevel algorithms on data hyper-cleaning task across two corruption rates $p = 0.5$ and $p = 0.8$. The results are averaged over 10 runs and $\pm$ is followed by the standard deviation. The results are conducted after 40 and 60 seconds running time.

| Algorithm | $p = 0.5$ | | | $p = 0.8$ | | |
|---|---|---|---|---|---|---|
| | Test accuracy (%) | Test loss | Residual | Test accuracy (%) | Test loss | Residual |
| LancBiO | $90.35 \pm 0.1716$ | $0.36 \pm 0.0028$ | $1.20e{-}4 \pm 2.52e{-}5$ | $89.45 \pm 0.2470$ | $0.42 \pm 0.0038$ | $9.18e{-}5 \pm 2.77e{-}5$ |
| SubBiO | $90.21 \pm 0.2159$ | $0.36 \pm 0.0035$ | $2.22e{-}2 \pm 2.63e{-}3$ | $89.22 \pm 0.2587$ | $0.42 \pm 0.0050$ | $2.49e{-}2 \pm 1.93e{-}3$ |
| AmIGO-GD | $90.16 \pm 0.2114$ | $0.37 \pm 0.0044$ | $3.66e{-}2 \pm 4.00e{-}3$ | $89.14 \pm 0.2722$ | $0.43 \pm 0.0044$ | $1.07e{-}2 \pm 8.25e{-}4$ |
| AmIGO-CG | $90.06 \pm 0.2305$ | $0.38 \pm 0.0053$ | $5.89e{-}4 \pm 2.52e{-}4$ | $88.57 \pm 0.5839$ | $0.46 \pm 0.0176$ | $6.32e{-}4 \pm 3.74e{-}4$ |
| SOBA | $90.00 \pm 0.1811$ | $0.37 \pm 0.0051$ | $2.74e{-}2 \pm 8.52e{-}3$ | $88.99 \pm 0.2661$ | $0.42 \pm 0.0054$ | $1.73e{-}2 \pm 1.70e{-}3$ |
| TTSA | $89.35 \pm 0.2747$ | $0.40 \pm 0.0103$ | - | $82.91 \pm 0.4516$ | $0.74 \pm 0.0072$ | - |
| stocBiO | $89.20 \pm 0.1824$ | $0.43 \pm 0.0033$ | - | $86.44 \pm 0.2907$ | $0.54 \pm 0.0064$ | - |
| F2SA | $89.78 \pm 0.1969$ | $0.40 \pm 0.0073$ | - | $88.65 \pm 0.2828$ | $0.51 \pm 0.0055$ | - |
| HJFBiO | $90.21 \pm 0.2027$ | $0.37 \pm 0.0048$ | - | $89.30 \pm 0.3594$ | $0.43 \pm 0.0040$ | - |

**Data hyper-cleaning on three datasets:** The data hyper-cleaning task (Shaban et al., 2019) aims to train a classifier in a corruption scenario, where the labels of the training data are randomly altered to incorrect classification numbers at a probability $p$, referred to as the corruption rate. The results on the MNIST dataset are presented in Figure 4 and Table 1. Note that LancBiO is crafted for approximating the Hessian inverse vector product $v^*$, while the two solid methods, TTSA and stocBiO are not. Consequently, with respect to the residual norm of the linear system, i.e., $\|A_k v_k - b_k\|$, we only compare the results with AmIGO-GD, AmIGO-CG, and SOBA. Observe that the proposed subspace-based LancBiO achieves the lowest residual norm and the best test accuracy, and subBiO is comparable to the other algorithms. Specifically, in Figure 4, the efficiency of LancBiO stems from its accurate approximation of the linear system. Additionally, while AmIGO-CG is also adept at approximating $v^*$, the results in Table 1 indicate that it tends to yield higher variance. Moreover, algorithms are also evaluated on the Fashion-MNIST and Kuzushiji-MNIST datasets; see Figure 10 and Figure 11, respectively. The proposed LancBiO performs better than other algorithms and showcases robustness across various datasets.

**Synthetic problem:** We concentrate on a synthetic bilevel optimization (1) with $d_x = d_y = d$ and

$$f(x, y) := c_1 \cos\left(x^\top D_1 y\right) + \frac{1}{2} \|D_2 x - y\|^2,$$

$$g(x, y) := c_2 \sum_{i=1}^{d} \sin(x_i + y_i) + \log\left(\sum_{i=1}^{d} e^{x_i y_i}\right) + \frac{1}{2} y^\top \left(D_3 + G\right) y.$$

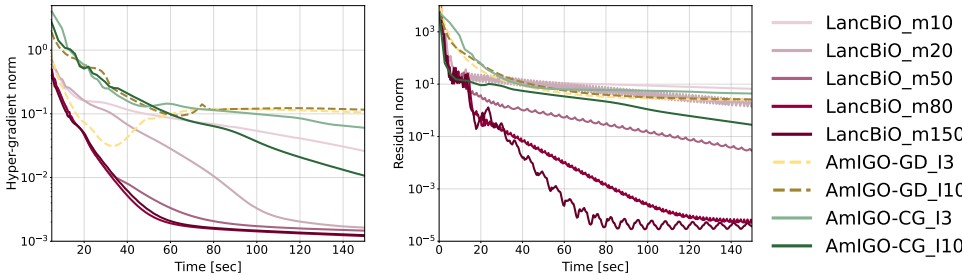

Figure 5: Influence of the subspace dimension $m$ on LancBiO. The post-fix of legend represents the subspace dimension $m$ or the inner iteration $I$. **Left:** norm of the hyper-gradient; **Right**: residual norm of the linear system, $\|A_k v_k - b_k\|$.

It can be seen from Figure 12 that LancBiO achieves the final accuracy the fastest, which benefits from the more accurate $v^*$ estimation. Figure 5 illustrates how variations in $m$ and $I$ influence the performance of LancBiO and AmIGO, tested across a range from 10 to 150 for $m$, and from 2 to 10 for $I$. For clarity, we set the seed of the experiment at 4, and present typical results to encapsulate the observed trends. It is observed that the increase of $m$ accelerates the decrease in the residual norm, thus achieving better convergence of the hyper-gradient, which aligns with the spirit of the classic Lanczos process. Under the same outer iterations, to attain a comparable convergence property, $I$ for AmIGO-CG should be set to 10. Furthermore, given that the number of Hessian-vector products averages at $(1 + 1/m)$ per outer iteration for LancBiO, whereas AmIGO involves $I \geq 2$ calculations, it follows that LancBiO is more efficient. Moreover, to illustrate how the methods scale with increasing dimensions, we present the convergence time and the final upper-level value under different problem dimensions $d = 10^i, i = 1, 2, 3, 4$ in Table 2. The results demonstrate the proposed methods maintain decent performance across different problem dimensions.

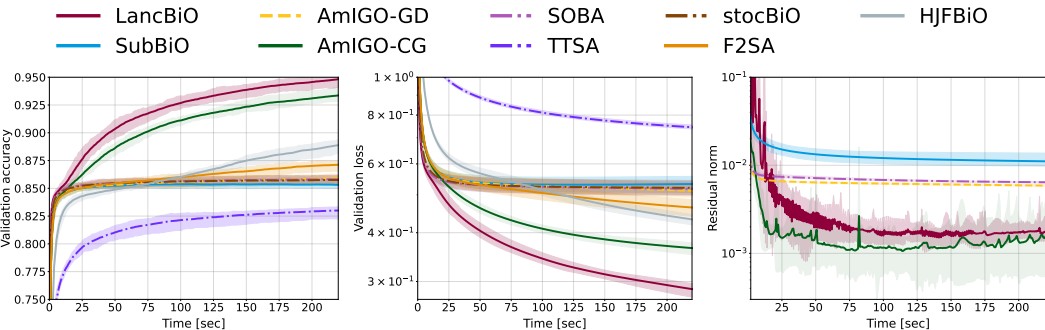

Figure 6: Comparison of the bilevel algorithms on the hyper-parameters selection task. **Left:** validation accuracy; **Center**: validation loss; **Right**: residual norm of the linear system, $\|A_k v_k - b_k\|$.

**Logistic regression on** 20**Newsgroup:** Consider the hyper-parameter selection task on the 20Newsgroups dataset (Grazzi et al., 2020). The goal is to train a linear classifier $w$ and determine the optimal regularization parameter $\zeta$. As shown in Figure 6, AmIGO-CG exhibits slightly better performance in reducing the residual norm. Nevertheless, under the same time, LancBiO implements more outer iterations to update $x$, which optimizes the hyper-function more efficiently.

Generally, to solve standard linear systems, the Lanczos process is recognized for its efficiency and versatility over gradient descent methods. LancBiO, in a sense, reflects this principle in the context of bilevel optimization, underscoring the effectiveness of the dynamic Lanczos-aided approach.

ACKNOWLEDGMENTS

Bin Gao was supported by the Young Elite Scientist Sponsorship Program by CAST. Ya-xiang Yuan was supported by the National Natural Science Foundation of China (grant No. 12288201). The authors are grateful to the Program and Area Chairs and Reviewers for their valuable comments and suggestions.

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

# Appendix

## A    RELATED WORK IN BILEVEL OPTIMIZATION

A variety of bilevel optimization algorithms are based on reformulation (Liu et al., 2021b; Yao et al., 2024). These algorithms involve transforming the lower-level problem into a set of constraints, such as the optimal conditions of the lower-level problem (Dempe and Dutta, 2012; Li et al., 2023), or the optimal value condition (Outrata, 1990; Ye and Zhu, 1995; 2010; Dempe and Zemkoho, 2013; Lin et al., 2014; Xu and Ye, 2014). Furthermore, incorporating the constraints of the reformulated problem as the penalty function into the upper-level objective inspires a series of algorithms (Liu et al., 2022; Hu et al., 2023; Kwon et al., 2023; Lu and Mei, 2024; Kwon et al., 2024). Another category of methods in bilevel optimization is the iterative differentiation (ITD) based method (Maclaurin et al., 2015; Franceschi et al., 2017; Shaban et al., 2019; Grazzi et al., 2020; Liu et al., 2020; Ji et al., 2021), which takes advantage of the automatic differentiation technique. Central to this approach is the construction of a computational graph during each outer iteration, achieved by solving the lower-level problem. This setup facilitates the approximation of the hyper-gradient through backpropagation, and it is noted that parts of these methods share a unified structure, characterized by recursive equations (Ji et al., 2021; Li et al., 2022; Zhang et al., 2024). The approximate implicit differentiation (AID) treats the lower-level variable as a function of the upper-level variable. It calculates the hyper-gradient to implement alternating gradient descent between the two levels (Ghadimi and Wang, 2018; Ji et al., 2021; Chen et al., 2022; Dagréou et al., 2022; Li et al., 2022; Hong et al., 2023). Moreover, extending the spirits of AID to distributed settings has garnered increasing interest in recent years(Kong et al., 2024; He et al., 2024; Zhu et al., 2024).

## B    KRYLOV SUBSPACE AND LANCZOS PROCESS

Krylov subspace (Krylov, 1931) is fundamental in numerical linear algebra (Parlett, 1998; Saad, 2011; Golub and Van Loan, 2013) and nonlinear optimization (Yuan, 2014; Liu et al., 2021c), specifically in the context of solving large linear systems and eigenvalue problems. In this section, we will briefly introduce the Krylov subspace and the Lanczos process, and recap some important properties; readers are referred to Saad (2011); Golub and Van Loan (2013) for more details.

An $N$-dimensional Krylov subspace generated by a matrix $A$ and a vector $b$ is defined as follows,

$$\mathcal{K}_N(A, b) := \text{span} \left\{ b, Ab, A^2b, \dots, A^{N-1}b \right\},$$

and the sequence of vectors $\left\{ b, Ab, A^2b, \dots, A^{N-1}b \right\}$ forms the basis for it. The Krylov subspace is widely acknowledged for its favorable properties in various aspects, including approximating eigenvalues (Kuczyński and Woźniakowski, 1992), solving the regularized nonconvex quadratic problems (Gould et al., 1999; Zhang et al., 2017; Carmon and Duchi, 2018), and reducing computation cost (Brown and Saad, 1990; Bellavia and Morini, 2001; Liu et al., 2013; Jiang et al., 2024).

The Lanczos process (Lanczos, 1950) is an algorithm that exploits the structure of the Krylov subspace when $A$ is symmetric. Specifically, in the $j$-th step of the Lanczos process, we can efficiently maintain an orthogonal basis $Q_j$ of $\mathcal{K}_j(A, b)$, so that $T_j = Q_j^\top A Q_j$ is tridiagonal, which means a tridiagonal matrix $T_j$ approximates $A$ in the Krylov subspace. Consequently, it allows to solve the minimal residual problem or the eigenvalue problem efficiently within the Krylov subspace. There are several equivalent variants of the Lanczos process (Paige, 1971; 1976; Meurant and Strakoš, 2006), and we follow the update rule as shown in Algorithm 3. We now present several key properties of the Krylov subspace and the Lanczos Process from Saad (2011).

**Definition B.1.** The minimal polynomial of a vector $v \in \mathbb{R}^n$ with respect to a matrix $A \in \mathbb{R}^{n \times n}$ is defined as the non-zero monic polynomial $p$ of the lowest degree such that $p(A)v = 0$, where a monic polynomial is a non-zero univariate polynomial with the coefficient of highest degree equal to 1.

*Remark* B.2. The degree of the minimal polynomial $p$ does not exceed $n$ because the set of $n+1$ vectors $\{A^n v, A^{n-1} v, \dots, A^2 v, Av, v\}$ is linearly dependent.

*Remark* B.3. Suppose the minimal polynomial of a vector $v$ with respect to a matrix $A$ is

$$p(x) = x^m + c_{m-1}x^{m-1} + \dots + c_2 x^2 + c_1 x + c_0,$$

---

**Algorithm 3** Lanczos process

---

**Input:** dimension $m$, matrix $A \in \mathbb{R}^{n \times n}$, initial vector $b \in \mathbb{R}^n$
1: **Initialization:** $q_1 = \frac{b}{\|b\|}$, $q_0 = \mathbf{0}$, $\beta_1 = 0$, $Q_0 = T_0 = $ Empty Matrix
2: **for** $j = 1, 2, \ldots, m$ **do**
3: $\quad u_j = Aq_j - \beta_j q_{j-1}$
4: $\quad \alpha_j = q_j^\top u_j$
5: $\quad \omega_j = u_j - \alpha_j q_j$
6: $\quad \beta_{j+1} = \|\omega_j\|$
7: $\quad q_{j+1} = \omega_j / \beta_{j+1}$
8: $\quad Q_j = [Q_{j-1} \ q_j]$
9: $\quad T_j = \begin{pmatrix} T_{j-1} & & \vdots \\ & & \beta_j \\ \hline \cdots & \overline{\beta_j} & \overline{\alpha_j} \end{pmatrix}$
10: **end for**
**Output:** $T_m, Q_m, \|b\| e_1$

---

and has a degree of $m$. If $c_0 \neq 0$ and $A$ is invertible, by Definition B.1,

$$A^m v + c_{m-1} A^{m-1} v + \cdots + c_2 A^2 v + c_1 A v + c_0 v = 0,$$

multiply both sides of the equation by $A^{-1}$ and rearrange the equation,

$$A^{-1} v = -\frac{1}{c_0} \left( A^{m-1} v + c_{m-1} A^{m-2} v + \cdots + c_2 A v + c_1 v \right).$$

In other words, $A^{-1} v$ belongs to the Krylov subspace $\mathcal{K}_m(A, v)$.

**Proposition B.4.** *Denote the $n \times j$ matrix with column vectors $q_1, \ldots, q_j$ by $Q_j$ and the $j \times j$ tridiagonal matrix by $T_j$, all of which are generated by Algorithm 3. Then the following three-term recurrence holds.*

$$AQ_j = Q_j T_j + \beta_{j+1} q_{j+1} e_j^\top,$$
$$Q_j^\top AQ_j = T_j.$$

Based on Proposition B.4, the Lanczos process is illustrated in Figure 7.

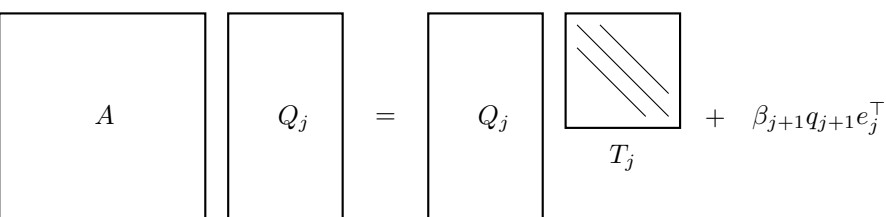

Figure 7: Classic three-term recurrence

## C  DYNAMIC LANCZOS SUBROUTINE

This section lists the DLanczos subroutine (Algorithm 4) invoked in the LancBiO framework (Algorithm 2). One of the main differences between Algorithm 3 and Algorithm 4 is that Algorithm 3 represents the entire $m$-step Lanczos process, while Algorithm 4 serves as a one-step subroutine. Specifically, LancBiO invokes DLanczos once in each outer iteration (line 11 in Algorithm 2), expanding both $T$ and $Q$ by one dimension. Consequently, the inputs of Algorithm 4 are not indexed to avoid confusion, with their corresponding variables $(T_{k-1}, Q_{k-1}, A_k, \beta_k)$ in Algorithm 2 evolving across outer iterations indexed by $k$. Another difference lies in the dynamic property of the DLanczos subroutine, i.e., the matrix $A_k$ passed during each invocation varies in Algorithm 2, while the classic Lanczos process (Algorithm 3) employs a static $A$.

---

**Algorithm 4** Dynamic Lanczos subroutine for LancBiO (`DLanczos`)

---

**Input:** tridiagonal matrix $T$, basis matrix $Q$ with $j$ columns, Hessian matrix $A$, and $\beta$
 1: **if** $j = 1$ **then**
 2:    $[q_1] = Q$, $q_{-1} = \mathbf{0}$
 3: **else**
 4:    $[q_1, q_2, \ldots, q_j] = Q$
 5: **end if**
 6: $u_j = Aq_j - \beta q_{j-1}$
 7: $\alpha_j = q_j^\top u_j$
 8: $\omega_j = u_j - \alpha_j q_j$
 9: $\beta_{j+1} = \|\omega_j\|$
10: $q_{j+1} = \omega_j / \beta_{j+1}$
11: $Q_{j+1} = [Q \; q_{j+1}]$
12: $T_{j+1} = \begin{pmatrix} & T & & \vdots \\ & & & \vdots \; \beta \\ \text{-} \text{-} \text{-} & \bar{\beta} & \text{|} & \bar{\alpha_j} \end{pmatrix}$
**Output:** $T_{j+1}, Q_{j+1}, \beta_{j+1}$

---

# D    EXTENDING LANCBIO TO NON-CONVEX LOWER-LEVEL PROBLEM

The Lanczos process is known for its efficiency of constructing Krylov subspaces and is capable of solving indefinite linear systems (Greenbaum et al., 1999). In this section, we will briefly demonstrate that the dynamic Lanczos-aided bilevel optimization framework, LancBiO, can also handle lower-level problems with the indefinite Hessian.

Suppose $A$ is invertible, and consider solving a standard linear system

$$Ax = b,$$

with initial ponit $x_0$, initial residual $r_0 = b - Ax_0$ and initial error $e_0 = A^{-1}b - x_0$. If the matrix $A$ is positive-definite, the classic Lanczos algorithm is equivalent to the Conjugate Gradient (CG) algorithm (Hestenes et al., 1952), both of which minimize the $A$-norm of the error in an affine space (Greenbaum, 1997; Meurant and Strakoš, 2006), i.e., at the $m$-th step,

$$x_m = \underset{x \in x_0 + \mathcal{K}_m(A,b)}{\arg\min} \left\| A^{-1}b - x \right\|_A.$$

If the matrix $A$ is not positive-definite, MINRES (Paige and Saunders, 1975) is the algorithm recognized to minimize the 2-norm of the residual in an affine space (Greenbaum, 1997; Meurant and Strakoš, 2006), i.e., at the $m$-th step,

$$x_m = \underset{x \in x_0 + \mathcal{K}_m(A,b)}{\arg\min} \|b - Ax\|. \tag{10}$$

Additionally, based on $Q_m$ as the basis of the Krylov subspace $\mathcal{K}_m(A,b)$, $T_m$ as the projection of $A$ onto $\mathcal{K}_m(A,b)$, and the three-term recurrence

$$AQ_m = Q_m T_m + \beta_{m+1} q_{m+1} e_m^\top,$$

we can rewrite (10) as

$$x_m = x_0 + Q_m c_m,$$

with

$$
\begin{aligned}
c_m &= \arg\min_c \|r_0 - AQ_m c\| \\
&= \arg\min_c \|r_0 - Q_{m+1} T_{m+1,m} c\| \\
&= \arg\min_c \|Q_{m+1} (\|r_0\| e_1 - T_{m+1,m} c)\| \\
&= \arg\min_c \| \|r_0\| e_1 - T_{m+1,m} c \|,
\end{aligned}
$$

where

$$T_{m+1,m} := \begin{bmatrix} T_m \\ \beta_{m+1} e_m^\top \end{bmatrix}.$$

In the spirit of MINRES, to address the bilevel problem where the lower-level problem exhibits an indefinite Hessian, the framework LancBiO (Algorithm 2) requires only a minor modification. Specifically, line 13 in Algorithm 2, , which solves a small-size tridiagonal linear system, will be replaced by solving a low-dimensional least squares problem

$$c_k = \arg\min_c \left\| \|r_k\| e_1 - T_{k+1,k} c \right\|^2,$$

and computing the correction

$$\Delta v_k = Q_k c_k,$$

where

$$T_{k+1,k} := \begin{bmatrix} T_k \\ \beta_{k+1} e_k^\top \end{bmatrix}.$$

## E  PROOF OF SMOOTHNESS OF $y^*$ AND $\varphi$

To ensure completeness, we provide detailed proofs for the preliminary lemmas that characterize the smoothness of the lower level solution $y^*$ and the hyper-objective $\varphi$.

**Lemma E.1.** *Under the Assumptions 3.2 and 3.3, $y^*(x)$ is $\frac{L_{gx}}{\mu_g}$-Lipschitz continuous, i.e., for any $x_1, x_2 \in \mathbb{R}^{d_x}$,*

$$\|y^*(x_1) - y^*(x_2)\| \le \frac{L_{gx}}{\mu_g} \|x_1 - x_2\|.$$

*Proof.* The assunption that $\nabla_x g(x,y)$ is $L_{gx}$-Lipschitz reveals $\left\| \nabla^2_{xy} g(x,y) \right\| \le L_{gx}$. Then

$$\|\nabla y^*(x)\| = \left\| \nabla^2_{xy} g(x,y) \left[ \nabla^2_{yy} g(x,y) \right]^{-1} \right\| \le \left\| \nabla^2_{xy} g(x,y) \right\| \left\| \left[ \nabla^2_{yy} g(x,y) \right]^{-1} \right\| \le \frac{L_{gx}}{\mu_g},$$

since $g(x,\cdot)$ is $\mu_g$-strongly convex.  □

**Lemma E.2.** *Under the Assumptions 3.1, 3.2 and 3.3, the hyper-gradient $\nabla\varphi(x)$ is $L_\varphi$-Lipschitz continuous, i.e., for any $x_1, x_2 \in \mathbb{R}^{d_x}$,*

$$\|\nabla\varphi(x_1) - \nabla\varphi(x_2)\| \le L_\varphi \|x_1 - x_2\|,$$

*where $L_\varphi = \left(1 + \frac{L_{gx}}{\mu_g}\right) \left( L_{fx} + \frac{L_{gx} L_{fy} + L_{gxy} C_{fy}}{\mu_g} + \frac{L_{gx} C_{fy} L_{gyy}}{\mu_g^2} \right)$.*

*Proof.* By combining

$$\begin{aligned}
&(A_1^*)^{-1} b_1^* - (A_2^*)^{-1} b_2^* \\
&= (A_1^*)^{-1} b_1^* - (A_1^*)^{-1} b_2^* + (A_1^*)^{-1} b_2^* - (A_2^*)^{-1} b_2^* \\
&= (A_1^*)^{-1} (b_1^* - b_2^*) + (A_1^*)^{-1} (A_2^* - A_1^*) (A_2^*)^{-1} b_2^*
\end{aligned}$$

with the properties revealed by Assumptions 3.1 3.2 and 3.3, we can derive

$$\left\| (A_1^*)^{-1} b_1 - (A_2^*)^{-1} b_2 \right\| \le \frac{L_{fy}}{\mu_g} \left( 1 + \frac{L_{gx}}{\mu_g} \right) \|x_1 - x_2\| + \frac{C_{fy} L_{gyy}}{\mu_g^2} \left( 1 + \frac{L_{gx}}{\mu_g} \right) \|x_1 - x_2\|.$$

In a similar way, the subsequent decomposition holds,

$$\begin{aligned}
\nabla\varphi(x_1) - \nabla\varphi(x_2) = {} & (\nabla_x f(x_1, y^*(x_1)) - \nabla_x f(x_2, y^*(x_2))) \\
& - \nabla^2_{xy} g(x_1, y^*(x_1)) (A_1^*)^{-1} b_1^* + \nabla^2_{xy} g(x_1, y^*(x_1)) (A_2^*)^{-1} b_2^* \\
& + \nabla^2_{xy} g(x_2, y^*(x_2)) (A_2^*)^{-1} b_2^* - \nabla^2_{xy} g(x_1, y^*(x_1)) (A_2^*)^{-1} b_2^*.
\end{aligned}$$

It follows that

$$\|\nabla\varphi(x_1) - \nabla\varphi(x_2)\| \le L_{fx}\left(1 + \frac{L_{gx}}{\mu_g}\right)\|x_1 - x_2\|$$

$$+ L_{gx}\left(1 + \frac{L_{gx}}{\mu_g}\right)\left(\frac{L_{fy}}{\mu_g} + \frac{C_{fy}L_{gyy}}{\mu_g^2}\right)\|x_1 - x_2\|$$

$$+ L_{gxy}\frac{C_{fy}}{\mu_g}\left(1 + \frac{L_{gx}}{\mu_g}\right)\|x_1 - x_2\|$$

$$= L_\varphi\|x_1 - x_2\|,$$

where $L_\varphi := \left(1 + \frac{L_{gx}}{\mu_g}\right)\left(L_{fx} + \frac{L_{gx}L_{fy} + L_{gxy}C_{fy}}{\mu_g} + \frac{L_{gx}C_{fy}L_{gyy}}{\mu_g^2}\right)$. $\qquad\square$

## F   PROPERTIES OF DYNAMIC SUBSPACE IN SECTION 3.1

In this section, we focus on the properties of the basis matrix $Q$ and the tridiagonal matrix $T$ constructed within each epoch of the dynamic Lanczos process. Denote

$$A_k^* = \nabla_{yy}^2 g(x_k, y_k^*) \text{ and } b_k^* = \nabla_y f(x_k, y_k^*).$$

An *epoch* is constituted of a complete $m$-step dynamic Lanczos process between two restarts, namely, after $h$ epochs, the number of outer iterations is $mh$. Given the outer iterations $k = mh + j$ for $j = 1, 2, \ldots, m$, we denote

$$\varepsilon_{st}^h := \left(1 + \frac{L_{gx}}{\mu_g}\right)\|x_{mh+s} - x_{mh+t}\| + \|y_{mh+s} - y_{mh+s}^*\|$$

for $s, t = 1, 2, \ldots, m$ and

$$\varepsilon_j^h := \max_{1 \le s,t \le j} \varepsilon_{st}^h,$$

serving as the accumulative difference. For brevity, we omit the superscript where there is no ambiguity, and we are slightly abusing of notation that at the current epoch, $\{A_{mh+j}\}$ and $\{b_{mh+j}\}$ are simplified by $\{A_j\}$ and $\{b_j\}$ for $j = 1, \ldots, m$. In addition, the approximations in the residual system (8) are simplified by $\bar{v}$ and $\bar{b} := b_1 - A_1\bar{v}$.

We rewrite the dynamic update rule from Section 2.2

$$u_j = A_j q_j - \beta_j q_{j-1}, \tag{11}$$

$$\alpha_j = q_j^\top u_j, \tag{12}$$

$$\omega_j = u_j - \alpha_j q_j, \tag{13}$$

$$\beta_{j+1} = \|\omega_j\|, \tag{14}$$

$$q_{j+1} = \omega_j/\beta_{j+1}, \tag{15}$$

for $j = 1, 2, \ldots, m$ with $q_0 = \mathbf{0}$, $\beta_1 = 0$ and $Q_1 = q_1 = \bar{b}/\|\bar{b}\|$. The following proposition characterizes that the dynamic subspace constructed in Algorithm 2 within an epoch is indeed an approximate Krylov subspace.

**Proposition F.1.** *At the $j$-th step within an epoch ($j = 1, 2, \ldots, m - 1$), the subspace spanned by the matrix $Q_{j+1}$ in Algorithm 2 satisfies*

$$\mathrm{span}(Q_{j+1}) \subseteq \mathrm{span}\left\{A_1^{a_1} A_2^{a_2} \cdots A_j^{a_j}\bar{b} \,\middle|\, \begin{array}{c} a_s = 0 \text{ or } 1 \\ \forall s = 1, 2, \ldots, j \end{array}\right\}. \tag{16}$$

*Specifically, when $A_1 = A_2 = \cdots = A_j = A$ and $Q_{j+1}$ is of full rank,*

$$\mathrm{span}(Q_{j+1}) = \mathcal{K}_{j+1}(A, \bar{b})$$

*Proof.* Note that $Q_1 = \mathrm{span}\{q_1\}$ with $q_1 = \frac{\bar{b}}{\|\bar{b}\|}$ satisfies (16). We will give a proof by induction. Suppose for $i = 1, 2, \ldots, j$, it holds that

$$\mathrm{span}(Q_{i+1}) \subseteq \mathrm{span}\left\{A_1^{a_1} A_2^{a_2} \cdots A_i^{a_i}\bar{b} \,\middle|\, \begin{array}{c} a_s = 0 \text{ or } 1 \\ \forall s = 1, 2, \ldots, i \end{array}\right\}.$$

By the dynamic Lanczos process, it yields

$$q_{j+2} = \frac{1}{\beta_{j+2}} \left( A_{j+1} q_{j+1} - \beta_{j+1} q_j - \alpha_{j+1} q_{j+1} \right).$$ (17)

Since

$$q_{j+1} \in \text{span} \left\{ A_1^{a_1} A_2^{a_2} \cdots A_j^{a_j} \bar{b} \;\middle|\; \begin{matrix} a_s = 0 \text{ or } a_s = 1 \\ \forall s = 1, 2, \ldots, j \end{matrix} \right\},$$

then we have

$$A_{j+1} q_{j+1} \in \text{span} \left\{ A_1^{a_1} A_2^{a_2} \cdots A_{j+1}^{a_{j+1}} \bar{b} \;\middle|\; \begin{matrix} a_s = 0 \text{ or } a_s = 1 \\ \forall s = 1, 2, \ldots, j+1 \end{matrix} \right\}.$$

It follows from (17) that

$$q_{j+2} \in \text{span} \left\{ A_1^{a_1} A_2^{a_2} \cdots A_{j+1}^{a_{j+1}} \bar{b} \;\middle|\; \begin{matrix} a_s = 0 \text{ or } a_s = 1 \\ \forall s = 1, 2, \ldots, j+1 \end{matrix} \right\}.$$

By induction, we complete the proof. $\qquad\square$

Although we can estimate the difference between the basis of the above two subspaces, it is noted that the Krylov subspaces can be very sensitive to small perturbation (Meurant and Strakoš, 2006; Paige, 1976; 1980; Greenbaum, 1997). The next lemma interprets the perturbation analysis for the dynamic Lanczos process in terms of $A_j^*$, which satisfies an approximate three-term recurrence with a perturbation term $\delta Q$.

**Lemma F.2.** *Suppose Assumptions 3.1 to 3.3 hold. The dynamic Lanczos process in Algorithm 2 with normalized $q_1$ and $\alpha_j, \beta_j, q_j$ satisfies*

$$A_j^* Q_j = Q_j T_j + \beta_{j+1} q_{j+1} e_j^\top + \delta Q_j$$ (18)

*for $j = 1, 2, \ldots, m$, where $Q_j = [q_1, q_2, \ldots, q_j]$, $\delta Q_j = [\delta q_1, \delta q_2, \ldots, \delta q_j]$,*

$$T_j = \begin{pmatrix} \alpha_1 & \beta_2 & & & \\ \beta_2 & \alpha_2 & \beta_3 & & \\ & \beta_3 & \ddots & \ddots & \\ & & \ddots & \ddots & \beta_j \\ & & & \beta_j & \alpha_j \end{pmatrix}.$$

*The columns of the perturbation $\delta Q_j$ satisfy*

$$\|\delta q_i\| \le L_{gyy} \varepsilon_j, \text{ for } i = 1, 2, \ldots, j.$$

*Additionally, if we decompose $Q_j$ as*

$$Q_j^\top Q_j = R_j^\top + R_j,$$ (19)

*with $R_j$ as a strictly upper triangular matrix, then*

$$T_j R_j - R_j T_j = \beta_{j+1} Q_j^\top q_{j+1} e_j^\top + \delta R_j,$$ (20)

*where $\delta R_j$ is strictly upper triangular with elements $|\zeta_{st}| \le 2 L_{gyy} \varepsilon_j$, for $1 \le s < t \le j$.*

*Proof.* From

$$\alpha_j = q_j^\top u_j = q_j^\top A_j q_j - \beta_j q_j^\top q_{j-1}$$

and

$$q_{j+1}^\top q_j = \frac{1}{\beta_{j+1}} \omega_j^\top q_j = \frac{1}{\beta_{j+1}} \left( u_j - \alpha_j q_j \right)^\top q_j = \frac{1}{\beta_{j+1}} \left( A_j q_j - \alpha_j q_j - \beta_j q_{j-1} \right)^\top q_j,$$

we can derive

$$q_{j+1}^\top q_j = 0$$ (21)

by induction. Then, we combine equations (11), (12), (13) and (15), and rewrite them in the perturbed form:

$$\beta_{i+1}q_{i+1} = A_i q_i - \beta_i q_{i-1} - \alpha_i q_i = A_j^* q_i - \beta_i q_{i-1} - \alpha_i q_i + \delta q_i, \quad for \ i = 1, 2, \ldots, j, \quad (22)$$

where $\|\delta q_i\| \le L_{gyy}\varepsilon_j$ due to Assumpstions 3.2 and 3.3. Specifically, (22) can be rewritten in a compact form:

$$A_j^* Q_j = Q_j T_j + \beta_{j+1}q_{j+1}e_j^\top + \delta Q_j.$$

Then, we consider the orthogonality of matrix $Q_j$, which is reflected by $R_j$ in (19). Multiply on both sides of (18) by $Q_j^\top$,

$$Q_j^\top A_j^* Q_j = Q_j^\top Q_j T_j + \beta_{j+1} Q_j^\top q_{j+1}e_j^\top + Q_j^\top \delta Q_j.$$

Combining its symmetry with the decomposition (19), we obtain

$$T_j\left(R_j^\top + R_j\right) - \left(R_j^\top + R_j\right)T_j = \beta_{j+1}\left(Q_j^\top q_{j+1}e_j^\top - e_j q_{j+1}^\top Q_j\right) + Q_j^\top \delta Q_j - \delta Q_j^\top Q_j. \quad (23)$$

Denote $M_j = T_j R_j - R_j T_j$ which is upper triangular. Since the consecutive $q_i$ is orthogonal as revealed by (21), we conclude that the diagonal elements of $M_j$ are 0. Furthermore, by extracting the upper triangular part of the right hand side of (23), we can get

$$M_j = T_j R_j - R_j T_j = \beta_{j+1}Q_j^\top q_{j+1}e_j^\top + \delta R_j,$$

where $\delta R_j$ is strictly upper triangular with elements $\zeta_{st}$ satisfying: for $t = 2, 3, \ldots, j$,

$$\begin{cases} \zeta_{t-1,t} &= q_{t-1}^\top \delta q_t - \delta q_{t-1}^\top q_t \\ \zeta_{st} &= q_s^\top \delta q_t - \delta q_s^\top q_t, \end{cases} \quad s = 1, 2, \ldots, t-2.$$

From the boundedness of $\|\delta q_j\|$, it follows that for $1 \le s < t \le j$, $|\zeta_{st}| \le 2L_{gyy}\varepsilon_j$. $\qquad\square$

Lemma F.2 illustrates the influence of the dynamics in Algorithm 2 imposed on the three-term Lanczos recurrence, and as (19) reveals, $R$ reflects the loss of orthogonality of the basis $Q$. However, the following lemmas demonstrate that the range of eigenvalues of the approximate projection matrix $T$ is indeed controllable.

To proceed, we establish the Ritz pairs of $T_j$ as $\left(\mu_i^{(j)}, y_i^{(j)}\right)$ for $i = 1, 2, \ldots, j$, such that

$$T_j Y^{(j)} = Y^{(j)} \operatorname{diag}\left(\mu_1^{(j)}, \mu_2^{(j)}, \ldots, \mu_j^{(j)}\right).$$

where the normalized $\{y_i^{(j)}\}_{i=1}^j$ form the orthogonal matrix $Y^{(j)}$ with the elements $\varsigma_{st}^{(j)}$ for $1 \le s, t \le j$, and we arrange the Ritz values in a specific order,

$$\mu_1^{(j)} > \mu_2^{(j)} > \cdots > \mu_j^{(j)}.$$

We define the $j$-th approximate eigenvector matrix

$$Z^{(j)} := \left[z_1^{(j)}, z_2^{(j)}, \ldots, z_j^{(j)}\right] := Q_j Y^{(j)}.$$

and the corresponding Rayleigh quotients of $A_j^*$

$$\nu_i^{(j)} := \frac{\left(z_i^{(j)}\right)^\top A_j^* z_i^{(j)}}{\left(z_i^{(j)}\right)^\top z_i^{(j)}}, \ \text{for } i = 1, 2, \ldots, j.$$

The subsequent lemma describes the difference of eigenvalues between $T_j$ and some $T_n$ constructed in preceding steps.

**Lemma F.3.** *Suppose Assumptions 3.1 to 3.3 hold. For any eigenpair $\left(\mu_i^{(j)}, y_i^{(j)}\right)$ of $T_j$, there exists an integer pair $(s, n)$ where $1 \le s \le n < j$, such that*

$$\left|\mu_i^{(j)} - \mu_s^{(n)}\right| \le \frac{2j^2 L_{gyy}\varepsilon_j}{\sqrt{3}\left|\left(y_i^{(j)}\right)^\top R_j y_i^{(j)}\right|}. \quad (24)$$

*Proof.* Multiply the extended eigenvector $y_r^{(i)}$ of $T_i$ by $T_j$, where $i < j$,

$$T_j \begin{bmatrix} y_r^{(i)} \\ \mathbf{0} \end{bmatrix} = \begin{bmatrix} \mu_r^{(i)} y_r^{(i)} \\ \beta_{i+1} \varsigma_{ir}^{(i)} e_1 \end{bmatrix}. \tag{25}$$

Then multiply $\left( y_s^{(j)} \right)^\top$ on the both sides of (25),

$$\left( \mu_s^{(j)} - \mu_r^{(i)} \right) \left( y_s^{(j)} \right)^\top \begin{bmatrix} y_r^{(i)} \\ \mathbf{0} \end{bmatrix} = \beta_{i+1} \varsigma_{ir}^{(i)} \varsigma_{i+1,s}^{(j)}, \tag{26}$$

and multiply the eigenvectors $\left( y_s^{(j)} \right)^\top$ and $y_t^{(j)}$ on the left and right of equation (20), respectively,

$$\left( \mu_s^{(j)} - \mu_t^{(j)} \right) \left( y_s^{(j)} \right)^\top R_j y_t^{(j)} = \varsigma_{jt}^{(j)} \beta_{j+1} \left( z_s^{(j)} \right)^\top q_{j+1} + \epsilon_{st}^{(j)}, \tag{27}$$

where we define

$$\epsilon_{st}^{(j)} := \left( y_s^{(j)} \right)^\top \delta R_j y_t^{(j)}. \tag{28}$$

Note that $\left| \epsilon_{st}^{(j)} \right| \leq 2j L_{gyy} \varepsilon_j$ because of Lemma F.2. Specifically, taking $s = t$ in (27),

$$\left( z_s^{(j)} \right)^\top q_{j+1} = -\frac{\epsilon_{ss}^{(j)}}{\beta_{j+1} \varsigma_{js}^{(j)}}, \tag{29}$$

we can rewrite (29) in the matrix form

$$Q_r^\top q_{r+1} = Y^{(r)} c_r, \tag{30}$$

where for $r = 1, \ldots, s$,

$$e_s^\top c_r := -\frac{\epsilon_{ss}^{(r)}}{\beta_{r+1} \varsigma_{rs}^{(j)}}.$$

By observing that $Q_r^\top q_{r+1} = Y^{(r)} c_r$ is the $(r+1)$-th column of $R_j$, we can derive

$$\left( y_i^{(j)} \right)^\top R_j y_i^{(j)} = -\sum_{r=1}^{j-1} \varsigma_{r+1,i}^{(j)} \sum_{t=1}^{r} \frac{\epsilon_{tt}^{(r)}}{\beta_{r+1} \varsigma_{rt}^{(r)}} \left( y_i^{(j)} \right)^\top \begin{bmatrix} y_t^{(r)} \\ \mathbf{0} \end{bmatrix} \tag{31}$$

$$= -\sum_{r=1}^{j-1} \left( \varsigma_{r+1,i}^{(j)} \right)^2 \sum_{t=1}^{r} \frac{\epsilon_{tt}^{(r)}}{\mu_i^{(j)} - \mu_t^{(r)}}, \tag{32}$$

where (31) and (32) follow from (30) and (26) respectively. Consequently, the definition (28) reveals

$$\sum_{s,t=1}^{j} \left( \epsilon_{st}^{(j)} \right)^2 = \|\delta R_j\|_F^2 \leq 2j^2 L_{gyy} \varepsilon_j. \tag{33}$$

Based on (32) and the orthogonality of $Y^{(j)}$,

$$\left| \left( y_i^{(j)} \right)^\top R_j y_i^{(j)} \right| \leq \left| \sum_{r=1}^{j-1} \varsigma_{r+1,i}^{(j)} \sum_{t=1}^{r} \frac{\epsilon_{tt}^{(r)}}{\min_{1 \leq d \leq l < j} \left| \mu_i^{(j)} - \mu_d^{(l)} \right|} \right|$$

$$\leq \frac{1}{\min_{1 \leq d \leq l < j} \left| \mu_i^{(j)} - \mu_d^{(l)} \right|} \left| \sum_{r=1}^{j-1} \sum_{t=1}^{r} \varsigma_{r+1,i}^{(j)} \epsilon_{tt}^{(r)} \right|,$$

it follows from the Cauchy–Schwarz inequality and (33) that

$$\min_{1 \leq d \leq l < j} \left| \mu_i^{(j)} - \mu_d^{(l)} \right| \leq \frac{\sqrt{j} \sqrt{\sum_{r=1}^{j-1} \sum_{t=1}^{r} \left( \epsilon_{tt}^{(r)} \right)^2}}{\left| \left( y_i^{(j)} \right)^\top R_j y_i^{(j)} \right|} \leq \frac{2j^2 L_{gyy} \varepsilon_j}{\sqrt{3} \left| \left( y_i^{(j)} \right)^\top R_j y_i^{(j)} \right|}.$$

$$\square$$

**Lemma F.4.** *Suppose Assumptions 3.1 to 3.3 hold. $Q_j$ and $T_j$ are the basis matrix and the approximate tridiagonal matrix in the $j$-th step and $R_j$ is the strictly upper triangular matrix defined in (19) characterizing the orthogonality of $Q_j$. Given $\mu_i^{(j)}$ the $i$-th eigenvalue of $T_j$, then for $i = 1, \ldots, j$,*

$$\mu_g - \frac{2\sqrt{3}}{3}(j+1)^3 L_{gyy}\varepsilon_j \le \mu_i^{(j)} \le L_{gy} + \frac{2\sqrt{3}}{3}(j+1)^3 L_{gyy}\varepsilon_j. \tag{34}$$

*Proof.* By conducting the left multiplication with $\left(y_i^{(j)}Q_j\right)^\top$ and the right multiplication with $y_i^{(j)}$ on both sides of equation (18), we obtain the following equation,

$$\left(z_i^{(j)}\right)^\top A_j^* z_i^{(j)} - \mu_i^{(j)}\left(z_i^{(j)}\right)^\top z_i^{(j)} = -\epsilon_{ii}^{(j)} + \left(z_i^{(j)}\right)^\top \delta Q_j y_i^{(j)}.$$

Dividing it by $\left(z_i^{(j)}\right)^\top z_i^{(j)}$, we have

$$\left| \frac{\left(z_i^{(j)}\right)^\top A_j^* z_i^{(j)} - \mu_i^{(j)}\left(z_i^{(j)}\right)^\top z_i^{(j)}}{\left(z_i^{(j)}\right)^\top z_i^{(j)}} \right| \le \frac{\left|\epsilon_{ii}^{(j)}\right| + \left|\left(z_i^{(j)}\right)^\top \delta Q_j y_i^{(j)}\right|}{\left|1 + 2\left(y_i^{(j)}\right)^\top R_j y_i^{(j)}\right|}$$

$$\le \frac{\left|\epsilon_{ii}^{(j)}\right| + L_{gyy}\varepsilon_j\sqrt{j\left(\left|1 + 2\left(y_i^{(j)}\right)^\top R_j y_i^{(j)}\right|\right)}}{\left|1 + 2\left(y_i^{(j)}\right)^\top R_j y_i^{(j)}\right|}, \tag{35}$$

where the inequalities come from

$$\left\|z_i^{(j)}\right\|^2 = 1 + 2\left(y_i^{(j)}\right)^\top R_j y_i^{(j)}. \tag{36}$$

**Case I:** If

$$\left|\left(y_i^{(j)}\right)^\top R_j y_i^{(j)}\right| < \frac{3}{8} - \frac{\varepsilon_j}{2} \tag{37}$$

holds, then from (36),

$$\left\|z_i^{(j)}\right\| \ge \frac{1}{2}.$$

Furthermore, (35) reveals that there exists a Rayleigh quotient $\nu^{(j)}$ of $A_j^*$ that satisfies

$$\left|\nu^{(j)} - \mu_i^{(j)}\right| \le 2\left|\epsilon_{ii}^{(j)}\right| + \sqrt{2j}L_{gyy}\varepsilon_j \le \left(4j + \sqrt{2j}\right)L_{gyy}\varepsilon_j.$$

**Case II:** If the condition (37) does not hold, by applying Lemma F.3, we can find an integer pair $(s_1, n_1)$ with $1 \le s_1 \le n_1 < j$ such that

$$\left|\mu_i^{(j)} - \mu_{s_1}^{(n_1)}\right| \le \frac{2j^2 L_{gyy}\varepsilon_j}{\sqrt{3}\left|\left(y_i^{(j)}\right)^\top R_j y_i^{(j)}\right|} \le 2\sqrt{3}j^2 L_{gyy}\varepsilon_j.$$

By observing that $\left(\mu_{s_1}^{(n_1)}, y_{s_1}^{(n_1)}\right)$ and $\left(y_{s_1}^{(n_1)}\right)^\top R_{n_1} y_{s_1}^{(n_1)}$ can also be categorized into **Case I** or **Case II**, we can repeat this process and construct a sequence $\{(s_t, n_t)\}_{t=0}^{l+1}$ with $1 \le n_{l+1} < n_l < \cdots < n_1 < n_0 = j$ until

$$\left|\left(y_{s_{l+1}}^{(n_{l+1})}\right)^\top R_j y_{s_{l+1}}^{(n_{l+1})}\right| < \frac{3}{8} - \frac{\varepsilon_{n_{l+1}}}{2}.$$

Inequality (37) holds when the superscript $j = 1$ since $T_1 = [\alpha_1]$, $y_1^{(1)} = 1$ and $z_1^{(1)} = q_1$. Therefore, we can obtain $\{(s_t, n_t)\}_{t=0}^{l+1}$ in finite steps, resulting in the following estimate.

$$
\begin{aligned}
\left| \bar{\nu}^{(j)} - \mu_i^{(j)} \right| &\le \left| \bar{\nu}^{(j)} - \mu_{s_{l+1}}^{(n_{l+1})} \right| + \sum_{t=0}^{l} \left| \mu_{s_{t+1}}^{(n_{t+1})} - \mu_{s_t}^{(n_l)} \right| \\
&\le \left( 4j + \sqrt{2j} \right) L_{gyy} \varepsilon_j + \sum_{t=0}^{l} 2\sqrt{3} n_t^2 L_{gyy} \varepsilon_j \\
&\le \frac{2\sqrt{3}}{3} (j+1)^3 L_{gyy} \varepsilon_j,
\end{aligned}
$$

for some $\bar{\nu}^{(j)}$.

Any Rayleigh quotient of $A_j^*$ is bounded by its eigenvalues (Parlett, 1998), i.e., for any $\nu^{(j)}$,

$$
\lambda_{min}^{(j)} \le \nu^{(j)} \le \lambda_{max}^{(j)},
$$

where $\lambda_{min}^{(j)}$ and $\lambda_{max}^{(j)}$ are the minimum and maximal eigenvalue of $A_j^*$, respectively. Based on Assumption 3.2 and Assumption 3.3, we complete the proof. □

## G   PROOF OF LEMMA 3.10

We detail the proof for Lemma 3.10 in this section, starting with a proof sketch.

### G.1   PROOF SKETCH

The proof of Lemma 3.10 is structured by four steps.

**Step1: extending $\varepsilon_j$ to $\tilde{\varepsilon}_j$ in the lemmas given in Appendix F.**
In Appendix F, we adopt $A_j^*$ and $b_j^*$ as reference values with each epoch for the analysis, i.e., for $j = 1, 2, \ldots, m$,

$$
A_j^* := \nabla_{yy}^2 g\left(x_j, y_j^*\right), \ b_j^* := \nabla_y f\left(x_j, y_j^*\right), \ \bar{b} := b_1 - A_1 \bar{v}, \ \bar{r}_0 := \bar{b}, \ \bar{r}_j := \bar{b} - A_j^* \Delta v_j.
$$

and

$$
\varepsilon_j := \max_{1 \le s, t \le j} \left( 1 + \frac{L_{gx}}{\mu_g} \right) \|x_{mh+s} - x_{mh+t}\| + \left\| y_{mh+s} - y_{mh+s}^* \right\|.
$$

In parallel, we can also view the $A_1, b_1$ as reference values. In this way, denote the similar quantities

$$
A_j := \nabla_{yy}^2 g\left(x_j, y_j\right), \ b_j := \nabla_y f\left(x_j, y_j\right), \ \bar{b}' := b_1 - A_1 \bar{v}, \ \bar{r}_0' := \bar{b}', \ \bar{r}_j' := \bar{b}' - A_1 \Delta v_j. \quad (38)
$$

and

$$
\tilde{\varepsilon}_j := \max_{1 \le s, t \le j} \|x_{mh+s} - x_{mh+t}\| + \|y_{mh+s} - y_{mh+t}\|. \quad (39)
$$

Consequently, we extend the lemmas in Appendix F. The results listed in Appendix G.2 are the recipe of Step2.

**Step2: upper-bounding the residual $\|\bar{r}_j'\|$.**
In Appendix G.3, we then demonstrate that if the value of $\tilde{\varepsilon}_j$ is not too large, $\|\bar{r}_j'\|$ can be bounded, which is an important lemma for Step3.

**Step3: controlling $\tilde{\varepsilon}_j$ and $\varepsilon_j$ by induction.**
Since the expression of $\tilde{\varepsilon}_j$ does not involve $y^*$, its magnitude can be controlled by adjusting the step size, implied by (39). With the help of the stability of $\tilde{\varepsilon}_j$, we can prove

$$
\mu_g - \frac{2\sqrt{3}}{3} (j+1)^3 L_{gyy} \varepsilon_j > 0. \quad (40)
$$

**Step4: proof of Lemma 3.10**

Based on the conclusion (40) revealing the benign property of the dynamic process, we achieve the proof of Lemma 3.10.

### G.2 EXTENDED LEMMAS FROM APPENDIX F

In this part, we view the $A_1, b_1$ as reference values. In this way, we can institute $A_j^*$ with $A_1$ and $\varepsilon_j$ with $\tilde{\varepsilon}_j$ in the lemmas from Appendix F, resulting in the extended version of lemmas.

**Lemma G.1.** *(Extended version of Lemma F.2) Suppose Assumptions 3.1 to 3.3 hold. The dynamic Lanczos process in Algorithm 2 with normalized $q_1$ satisfies*

$$A_1 Q_j = Q_j T_j + \beta_{j+1} q_{j+1} e_j^\top + \delta Q_j' \tag{41}$$

*for $j = 1, 2, \ldots, m$, where $Q_j = [q_1, q_2, \ldots, q_j]$, $\delta Q_j' = [\delta q_1', \delta q_2', \ldots, \delta q_j']$,*

$$T_j = \begin{pmatrix} \alpha_1 & \beta_2 & & & & \\ \beta_2 & \alpha_2 & \beta_3 & & & \\ & \beta_3 & \ddots & \ddots & & \\ & & \ddots & \ddots & \beta_j \\ & & & \beta_j & \alpha_j \end{pmatrix}.$$

*The columns of the perturbation $\delta Q_j'$ satisfy*

$$\|\delta q_i'\| \le L_{gyy} \tilde{\varepsilon}_j, \text{ for } i = 1, 2, \ldots, j. \tag{42}$$

**Lemma G.2.** *(Extended version of Lemma F.4) Suppose Assumptions 3.1 to 3.3 hold. $Q_j$ and $T_j$ are the basis matrix and the approximate tridiagonal matrix in the $j$-th step. Take $\mu_i^{(j)}$ as the $i$-th eigenvalue of $T_j$, then for $i = 1, \ldots, j$,*

$$\mu_g - \frac{2\sqrt{3}}{3}(j+1)^3 L_{gyy}\tilde{\varepsilon}_j \le \mu_i^{(j)} \le L_{gy} + \frac{2\sqrt{3}}{3}(j+1)^3 L_{gyy}\tilde{\varepsilon}_j. \tag{43}$$

### G.3 PROOF OF STEP2

The following lemma demonstrates that if the value of $\tilde{\varepsilon}_j$ is not too large, $\|\bar{r}_j'\|$ can be bounded.

**Lemma G.3.** *Suppose Assumpsions 3.1 to 3.3 and*

$$\mu_g - \frac{2\sqrt{3}}{3}(j+1)^3 L_{gyy}\tilde{\varepsilon}_j > 0$$

*are satisfied within an epoch. Then, it holds that*

$$\frac{\|\bar{r}_j'\|}{\|\bar{r}_0'\|} \le 2\sqrt{\tilde{\kappa}'(j)}\left(\frac{\sqrt{\tilde{\kappa}'(j)} - 1}{\sqrt{\tilde{\kappa}'(j)} + 1}\right)^j + \sqrt{j}L_{gyy}\varepsilon_j\tilde{\kappa}'(j),$$

*where $\tilde{\kappa}'(j) := \frac{L_{gy} + \frac{2\sqrt{3}}{3}(j+1)^3 L_{gyy}\tilde{\varepsilon}_j}{\mu_g - \frac{2\sqrt{3}}{3}(j+1)^3 L_{gyy}\tilde{\varepsilon}_j}$.*

*Proof.* Denote the solution in the dynamic subspace in the $j$-th step by

$$\Delta\xi_j = (T_j)^{-1}\bar{b} = \|\bar{r}_0'\|(T_j)^{-1}e_1, \tag{44}$$

where $T_j$ is nonsingular because of Lemma G.2. By (38), (41), and (44), we have

$$\bar{r}_j' = \bar{b} - A_1 Q_j \Delta\xi_j = -\beta_{j+1}q_{j+1}e_j^\top \Delta\xi_j - \delta Q_j' \Delta\xi_j.$$

It follows that

$$\frac{\|\bar{r}_j'\|}{\|\bar{r}_0'\|} \le \|\delta Q_j'\| \|(T_j)^{-1}\| + \left|\beta_{j+1}e_j^\top (T_j)^{-1} e_1\right|. \tag{45}$$

The first term on the right side of (45) can be bounded by (42) and (43):

$$\left\| \delta Q'_j \right\| \left\| (T_j)^{-1} \right\| \le \sqrt{j} L_{gyy} \tilde{\varepsilon}_j \frac{L_{gy}}{\mu_g - \frac{2\sqrt{3}}{3} (j+1)^3 L_{gyy} \tilde{\varepsilon}_j} \le \sqrt{j} L_{gyy} \tilde{\varepsilon}_j \tilde{\kappa}'(j). \tag{46}$$

Recall that

$$T_{j+1,j} := \left[ \begin{array}{c} T_j \\ \beta_{j+1} e_j^\top \end{array} \right], \tag{47}$$

and that for any symmetric tridiagonal matrix $T$, where the upper left $(j+1) \times j$ block is $T_{j+1,j}$, the application of the classic Lanczos algorithm to $T$, starting with the initial vector $e_1$ will result in the matrix $T_{j+1,j}$ at the $j$-th step. To construct a suitable $(j+1) \times (j+1)$ symmetric tridiagonal matrix T, we consider a *virtual step* with $\lambda^\diamond = \theta^\diamond = 0$, which leads to $(x^\diamond_{j+1}, y^\diamond_{j+1}) = (x_j, y_j)$, $\tilde{\varepsilon}^\diamond_{j+1} = \tilde{\varepsilon}_j$, and

$$T = \left( \begin{array}{cc} T_{j+1,j} & \vdots \\ & \vdots \\ & \beta_{j+1} \\ & \tilde{\alpha}^\diamond_{j+1} \end{array} \right)$$

By Lemma G.2, given any eigenvalue $\mu$ of $T$

$$\mu_g - \frac{2\sqrt{3}}{3} (j+1)^3 L_{gyy} \tilde{\varepsilon}_j \le \mu \le L_{gy} + \frac{2\sqrt{3}}{3} (j+1)^3 L_{gyy} \tilde{\varepsilon}_j.$$

In this way, $\left| \beta_{j+1} e_j^\top (T_j)^{-1} e_1 \right|$ can be seen as the residual in the $j$-th step of the classic Lanczos process with the positive-definite matrix $T$ and the initial vector $e_1$. Since the eigenvalues of $T$ satisfy (47), it follows from the standard convergence property of the Lanczos process (Greenbaum, 1997) that

$$\left| \beta_{j+1} e_j^\top (T_j)^{-1} e_1 \right| \le 2\sqrt{\tilde{\kappa}'(j)} \left( \frac{\sqrt{\tilde{\kappa}'(j)} - 1}{\sqrt{\tilde{\kappa}'(j)} + 1} \right)^j,$$

which completes the proof. □

### G.4 PROOF OF $\mu_g - \frac{2\sqrt{3}}{3} (j+1)^3 L_{gyy} \varepsilon_j > 0$

In this part, we will give the detailed proof of $\mu_g - \frac{2\sqrt{3}}{3} (j+1)^3 L_{gyy} \varepsilon_j > 0$. At the same time, we demonstrate that, with appropriate step sizes, the auxiliary variable $v_k$ is bounded, thereby showing that the hyper-gradient estimator (5) remains bounded. This ensures the stable behavior of the dynamic Lanczos process,

**Lemma G.4.** *Suppose Assumptions 3.1, 3.2, 3.3, 3.6 and 3.9 hold. If within each epoch, we set the step size $\theta \sim \mathcal{O}(1/m)$ a constant for y, and the step size for x as zero in the first $m_0 \sim \mathcal{O}(1)$ steps and the others as an appropriate constant $\lambda \sim \mathcal{O}(1/m^4)$, then for any epoch,*

$$\mu_g - \frac{2\sqrt{3}}{3} (j+1)^3 L_{gyy} \varepsilon_j > 0, \ for \ j = 1, 2, \ldots, m+1,$$

*and there exists a constant $C_v > 0$ so that $\|v_k\| \le C_v$ for $v_k$ generated by Algorithm 2.*

*Proof.* Consider the iterates within one epoch and the constants

$$0 < \tilde{\varepsilon} < \mu_g \text{ and } \tilde{\kappa} := \frac{L_{gy} + \tilde{\varepsilon}}{\mu_g - \tilde{\varepsilon}}.$$

It follows from Lemma G.3 that if $\tilde{\varepsilon}_j \le \frac{\sqrt{3}\tilde{\varepsilon}}{2(m+1)^3 L_{gyy}}$, then

$$\frac{\|b_1 - A_1 v_j\|}{\|b_1 - A_1 \bar{v}\|} \le 2\sqrt{\tilde{\kappa}} \left( \frac{\sqrt{\tilde{\kappa}} - 1}{\sqrt{\tilde{\kappa}} + 1} \right)^j + \frac{\sqrt{3}\tilde{\varepsilon}}{2(m+1)^2} \tilde{\kappa} \le 3\sqrt{\tilde{\kappa}}. \tag{48}$$

Then, we give a proof by induction. At the beginning of the algorithm, Assumption 3.9 reveals

$$\varepsilon_1 = \|y_1 - y_1^*\| \leq \frac{\sqrt{3}\mu_g}{8\left(m+1\right)^3 L_{gyy}}.$$

Combining it with $\|b_1 - A_1\bar{v}_0\| \leq C_r$ constructs the start of induction within an epoch and induction between epochs. Within an epoch, suppose the following statements hold for $i = 1, 2, \ldots, j$,

$$\|b_1 - A_1 v_i\| \leq 3\sqrt{\tilde{\kappa}}C_r,$$

$$\|v_i\| \leq \frac{1}{\mu_g}\left(3\sqrt{\tilde{\kappa}}C_r + C_{fx}\right),$$

$$\left\|\tilde{\nabla}\varphi_i\right\| \leq C_{fx} + \frac{1}{\mu_g^2}\left(3\sqrt{\tilde{\kappa}}C_r + C_{fx}\right),$$

$$\tilde{\varepsilon}_i \leq \frac{\sqrt{3}\tilde{\varepsilon}}{2(m+1)^3 L_{gyy}},$$

$$\varepsilon_i \leq \frac{\sqrt{3}\mu_g}{4\left(m+1\right)^3 L_{gyy}}.$$

Then by setting the stepsizes

$$\theta \leq \frac{\tilde{\varepsilon}}{\mu_g L_{gy} m},$$

$$\lambda \leq \frac{\sqrt{3}\tilde{\varepsilon}}{4(m+1)^4 L_{gyy}}\left(1 + \theta\frac{L_{gy}L_{gx}}{\mu_g}\right)^{-1}\left(C_{fx} + \frac{1}{\mu_g^2}\left(3\sqrt{\tilde{\kappa}}C_r + C_{fx}\right)\right)^{-1}, \tag{49}$$

and noticing that

$$\begin{aligned}
\|\nabla_y g(x_{i+1}, y_i)\| &= \left\|\nabla_y g(x_{i+1}, y_i) - \nabla_y g(x_{i+1}, y_{i+1}^*)\right\| \\
&\leq L_{gy}\left\|y_i - y_{i+1}^*\right\| \\
&\leq L_{gy}\left(\|y_i - y_i^*\| + \left\|y_i^* - y_{i+1}^*\right\|\right) \\
&\leq L_{gy}\|y_i - y_i^*\| + \frac{L_{gy}L_{gx}}{\mu_g}\|x_i - x_{i+1}\|,
\end{aligned}$$

we can get

$$\begin{aligned}
\tilde{\varepsilon}_{j+1} &\leq \lambda \sum_{i=1}^{i=j}\left\|\tilde{\nabla}\varphi_i\right\| + \theta \sum_{i=1}^{i=j}\|\nabla_y g\left(x_{i+1}, y_i\right)\| \\
&\leq \left(1 + \theta\frac{L_{gy}L_{gx}}{\mu_g}\right)\lambda \sum_{i=1}^{i=j}\left\|\tilde{\nabla}\varphi_i\right\| + \theta L_{gy}\sum_{i=1}^{i=j}\|y_i - y_i^*\| \\
&\leq \left(1 + \theta\frac{L_{gy}L_{gx}}{\mu_g}\right)\lambda \sum_{i=1}^{i=j}\left\|\tilde{\nabla}\varphi_i\right\| + \theta L_{gy}\sum_{i=1}^{i=j}\varepsilon_i \\
&\leq \left(1 + \theta\frac{L_{gy}L_{gx}}{\mu_g}\right)\lambda \sum_{i=1}^{i=j}\left\|\tilde{\nabla}\varphi_i\right\| + \theta L_{gy}\frac{\sqrt{3}\mu_g m}{4\left(m+1\right)^3 L_{gyy}} \\
&\leq \frac{\sqrt{3}\tilde{\varepsilon}}{2(m+1)^3 L_{gyy}}. \tag{50}
\end{aligned}$$

It follows from (48) that

$$\|b_1 - A_1 v_{j+1}\| \leq 3\sqrt{\tilde{\kappa}}C_r, \tag{51}$$

$$\|v_{j+1}\| = \left\|A_1^{-1}\left(b_1 - A_1 v_{j+1} - b_1\right)\right\| \leq \frac{1}{\mu_g}\left(3\sqrt{\tilde{\kappa}}C_r + C_{fx}\right), \tag{52}$$

$$\left\|\tilde{\nabla}\varphi_{j+1}\right\| \leq C_{fx} + \frac{1}{\mu_g^2}\left(3\sqrt{\tilde{\kappa}}C_r + C_{fx}\right). \tag{53}$$

Additionally, the descent property of $\|y_s - y_s^*\|$ and the Lipschitz continuity of $y^*$ reveal that

$$\|y_s - y_s^*\| \le (1 - \theta\mu_g)^{\frac{1}{2}} \|y_{s-1} - y_s^*\| \tag{54}$$

$$\le (1 - \theta\mu_g)^{\frac{1}{2}} \|y_{s-1} - y_{s-1}^*\| + (1 - \theta\mu_g)^{\frac{1}{2}} \left(\frac{L_{gx}}{\mu_g}\right) \|x_s - x_{s-1}\|$$

$$\le (1 - \theta\mu_g)^{\frac{s-1}{2}} \|y_1 - y_1^*\| + (1 - \theta\mu_g)^{\frac{1}{2}} \left(\frac{L_{gx}}{\mu_g}\right) \lambda \sum_{t=1}^{t=s-1} \left\|\widetilde{\nabla}\varphi_t\right\|$$

$$\le \|y_1 - y_1^*\| + (1 - \theta\mu_g)^{\frac{1}{2}} \left(\frac{L_{gx}}{\mu_g}\right) \lambda \sum_{t=1}^{t=s-1} \left\|\widetilde{\nabla}\varphi_t\right\|. \tag{55}$$

Setting

$$\lambda \le \frac{\sqrt{3}\mu_g}{8(m+1)^4 L_{gyy}} \left(C_{fx} + \frac{1}{\mu_g^2}\left(3\sqrt{\tilde{\kappa}}C_r + C_{fx}\right)\right)^{-1} \left(1 + \frac{L_{gx}}{\mu_g} + (1 - \theta\mu_g)^{\frac{1}{2}}\left(\frac{L_{gx}}{\mu_g}\right)\right)^{-1}$$

yields from Assumption 3.9, (52), (55) that

$$\varepsilon_{j+1} \le \left(1 + \frac{L_{gx}}{\mu_g}\right) \max_{1 \le s, t \le j+1} \|x_s - x_t\| + \max_{1 \le s \le j+1} \|y_s - y_s^*\|$$

$$\le \|y_1 - y_1^*\| + \left(1 + \frac{L_{gx}}{\mu_g} + (1 - \theta\mu_g)^{\frac{1}{2}}\left(\frac{L_{gx}}{\mu_g}\right)\right) \lambda \sum_{i=1}^{i=j} \left\|\widetilde{\nabla}\varphi_i\right\|$$

$$\le \|y_1 - y_1^*\| + \left(1 + \frac{L_{gx}}{\mu_g} + (1 - \theta\mu_g)^{\frac{1}{2}}\left(\frac{L_{gx}}{\mu_g}\right)\right) \lambda \sum_{i=1}^{i=j} \left\|\widetilde{\nabla}\varphi_i\right\|$$

$$\le \frac{\sqrt{3}\mu_g}{4(m+1)^3 L_{gyy}}. \tag{56}$$

As for the next epoch, denoting $C_v = \frac{1}{\mu_g}\left(3\sqrt{\tilde{\kappa}}C_r + C_{fx}\right)$, we have

$$\|b_{m+1} - A_{m+1}v_m\| \le (L_{fx} + L_{gyy}C_v)\tilde{\varepsilon}_{m+1} + \|b_1 - A_1 v_m\|$$

$$\le (L_{fx} + L_{gyy}C_v)\tilde{\varepsilon}_{m+1} + \left(2\sqrt{\tilde{\kappa}}\left(\frac{\sqrt{\tilde{\kappa}} - 1}{\sqrt{\tilde{\kappa}} + 1}\right)^m + \sqrt{m}L_{gyy}\tilde{\varepsilon}_{m+1}\tilde{\kappa}\right)C_r$$

$$\le C_r \left(\left(\frac{L_{fx} + L_{gyy}C_v}{C_r}\right)\tilde{\varepsilon}_{m+1} + 2\sqrt{\tilde{\kappa}}\left(\frac{\sqrt{\tilde{\kappa}} - 1}{\sqrt{\tilde{\kappa}} + 1}\right)^m + \sqrt{m}L_{gyy}\tilde{\varepsilon}_{m+1}\tilde{\kappa}\right)$$

$$\le C_r, \tag{57}$$

by choosing $m, \tilde{\varepsilon}$ such that

$$\left(\frac{L_{fx} + L_{gyy}C_v}{C_r}\right)\tilde{\varepsilon}_{m+1} + 2\sqrt{\tilde{\kappa}}\left(\frac{\sqrt{\tilde{\kappa}} - 1}{\sqrt{\tilde{\kappa}} + 1}\right)^m + \sqrt{m}L_{gyy}\tilde{\varepsilon}_{m+1}\tilde{\kappa} \le 1.$$

Moreover, since the step size for $x$ is set as zero at the first $m_0$ steps in the next epoch, we obtain for $i = 1, 2, \ldots, m_0$,

$$\varepsilon_{m+i} = \left\|y_{m+i} - y_{m+i}^*\right\| \le (1 - \theta\mu_g)^{\frac{i}{2}} \|y_m - y_m^*\| \le \varepsilon_m \le \frac{\sqrt{3}\mu_g}{4(m+1)^3 L_{gyy}}. \tag{58}$$

Specifically,

$$\left\|y_{m+m_0} - y_{m+m_0}^*\right\| \le (1 - \theta\mu_g)^{\frac{m_0}{2}} \|y_m - y_m^*\| \le \frac{\sqrt{3}\mu_g}{8(m+1)^3 L_{gyy}} \tag{59}$$

if we choose $m_0$ so that $(1 - \theta\mu_g)^{\frac{m_0}{2}} \le \frac{1}{2}$. Therefore, by induction within an epoch (50), (51), (52), (53), (56) and induction between epochs (57), (58), (59), we conclude the lemma. $\qquad \square$

### G.5 Proof of Lemma 3.10

**Lemma G.5.** *Suppose Assumptions 3.1, 3.2, 3.3, 3.6 and 3.9 hold. If within each epoch, we set the step size $\theta \sim \mathcal{O}(1/m)$ a constant for $y$ and the step size for $x$ as zero in the first $m_0 \sim \mathcal{O}(1)$ steps, and the others as an appropriate constant $\lambda \sim \mathcal{O}(1/m^4)$, we have the following inequality,*

$$\frac{\|\bar{r}_j\|}{\|\bar{r}_0\|} \leq 2\sqrt{\tilde{\kappa}(j)} \left(\frac{\sqrt{\tilde{\kappa}(j)} - 1}{\sqrt{\tilde{\kappa}(j)} + 1}\right)^j + \sqrt{j} L_{gyy} \varepsilon_j \tilde{\kappa}(j),$$

*where $\tilde{\kappa}(j) := \frac{L_{gy} + \frac{2\sqrt{3}}{3}(j+1)^3 L_{gyy}\varepsilon_j}{\mu_g - \frac{2\sqrt{3}}{3}(j+1)^3 L_{gyy}\varepsilon_j}$.*

*Proof.* Lemma G.4 guarantees the condition

$$\mu_g - \frac{2\sqrt{3}}{3}(j+1)^3 L_{gyy}\varepsilon_j > 0$$

is satisfied. The remaining proof can be directly adapted from Lemma G.3. □

## H Proof of The Main Theorem

In this section, we provide proof of the main theorem presented in Section 3.2. Let $A_k = \nabla_{yy}^2 g(x_k, y_k)$ and $b_k = \nabla_y f(x_k, y_k)$, and let the reference values be $A_k^* = \nabla_{yy}^2 g(x_k, y_k^*)$, $b_k^* = \nabla_y f(x_k, y_k^*)$ and $v_k^* = (A_k^*)^{-1} b_k^*$. To begin with, a short proof sketch is provided for guidance, which is structured in four main steps.

**Step1: upper-bounding the residual $\|v_k - v_k^*\|$**
Appendix F and G lay a foundation to bound the residual term $\|v_k - v_k^*\|$ (as a corollary of Lemma G.5).

**Step2: studying the descent property of $\|y_k - y_k^*\|$**
Lemma H.1 reveals the descent property of the estimation error for $y^*$ as follows,

$$\left\|y_{k+1} - y_{k+1}^*\right\|^2 \leq (1+\sigma)(1-\theta\mu_g)\|y_k - y_k^*\|^2 + \left(1 + \frac{1}{\sigma}\right)(1-\theta\mu_g)\left(\frac{L_{gx}}{\mu_g}\right)^2 \|x_{k+1} - x_k\|^2.$$

**Step3: Controlling the hyper-gradient estimation error $\|\widetilde{\nabla}\varphi(x_k) - \nabla\varphi(x_k)\|$**
Defining

$$\delta_k := \left(\frac{L_{fx}^2 \mu_g^2 + L_{gxy}^2 C_{fg}^2}{L_{gx}^2 \mu_g^2}\right)\|y_k - y_k^*\|^2 + \|v_k - v_k^*\|^2,$$

and incorporating the results from the last two steps, then in Lemma H.2 we can establish the upper bound for $\|\widetilde{\nabla}\varphi(x_k) - \nabla\varphi(x_k)\|$ and $\delta_k$ recursively.

**Step4: Assembling the estimations above and achieving the conclusion**
Consider the descent property of $\mathcal{L}_k := \varphi_k + \delta_k$. Substituting the inequalities developed in step1 to step3, and telescoping the index from $0$ to $K$ gives the convergence results.

The following lemma displays the descent property of the iterates $\{y_k\}$.

**Lemma H.1.** *Suppose Assumptions 3.2 and 3.3 hold. Setting $0 < \theta \leq \frac{2}{\mu_g + L_{gy}}$, we have*

$$\left\|y_{k+1} - y_{k+1}^*\right\|^2 \leq (1+\sigma)(1-\theta\mu_g)\|y_k - y_k^*\|^2 + \left(1 + \frac{1}{\sigma}\right)(1-\theta\mu_g)\left(\frac{L_{gx}}{\mu_g}\right)^2 \|x_{k+1} - x_k\|^2,$$

*for any $\sigma > 0$.*

*Proof.* Algorithm 2 executes a single-step gradient descent on the strongly convex function $g(x_{k+1}, \cdot)$ during the outer iteration. Leveraging the established convergence properties of strongly convex functions (Nesterov et al., 2018), we are thus able to derive the following.

$$\left\| y_{k+1} - y_{k+1}^* \right\|^2 \leq (1 - \theta \mu_g) \left\| y_k - y_{k+1}^* \right\|^2.$$

By Young's inequality that $|a + b|^2 \leq (1 + \sigma) |a|^2 + \left(1 + \frac{1}{\sigma}\right) \|b\|^2$ with any $\sigma > 0$,

$$\left\| y_{k+1} - y_{k+1}^* \right\|^2 \leq (1 + \sigma)(1 - \theta\mu_g) \|y_k - y_k^*\|^2 + \left(1 + \frac{1}{\sigma}\right)(1 - \theta\mu_g) \left(\frac{L_{gx}}{\mu_g}\right)^2 \|x_{k+1} - x_k\|^2.$$

$\square$

In the context of bilevel optimization, we define the initial residual in the $(h + 1)$-th epoch as

$$r_{h+1} := b_{mh+1} - A_{mh+1}\bar{v}_h,$$

and the residual in $k$-th step

$$r_k := (b_{mh+1} - A_{mh+1}\bar{v}_h) - A_k^* \Delta v_k.$$

Based on the boundness of $v_k$ in Lemma G.4 we can estimate

$$
\begin{aligned}
\mu_g \|v_k - v_k^*\| &\leq \|(b_k^* - A_k^* \bar{v}_h) - A_k^* \Delta v_k\| \\
&= \|(b_k^* - A_k^* \bar{v}_h) - A_k^* \Delta v_k - r_k + r_k\| \\
&\leq \|b_k^* - b_{mh+1}\| + \|A_k^* - A_{mh+1}\| \|\bar{v}_h\| + \|r_k\| \\
&\leq L_{fy} \|(x_k, y_k^*) - (x_{mh+1}, y_{mh+1})\| \\
&\quad + L_{gyy} \|(x_k, y_k^*) - (x_{mh+1}, y_{mh+1})\| \|\bar{v}_h\| + \|r_k\| \\
&= (L_{fy} + L_{gyy} \|v_{mh}\|) \varepsilon_j^h + \|r_k\| \\
&\leq (L_{fy} + L_{gyy} C_v) \varepsilon_j^h + \|r_k\|,
\end{aligned}
\tag{60}
$$

which comes from $\left\|(A_j^*)^{-1}\right\| \leq \frac{1}{\mu_g}$, $v_k = \bar{v}_h + \Delta v_k$, and $\|v_k\| \leq C_v$.

**Lemma H.2.** *Suppose Assumptions 3.1, 3.2, 3.3, 3.6 and 3.9 hold. Within each epoch, we set the step size $\theta \sim \mathcal{O}(1/m)$ a constant for $y$ and the step size for $x$ as zero in the first $m_0$ steps, and the others as an appropriate constant $\lambda \sim \mathcal{O}(1/m^4)$, then the iterates*

$$\{x_k\} \text{ for } k = mh + j, \ h = 0, 1, 2, \ldots, \text{ and } j = m_0 + 1, m_0 + 2, \ldots, m,$$

*generated by Algorithm 2 satisfy*

$$\left\| \widetilde{\nabla}\varphi(x_k) - \nabla\varphi(x_k) \right\|^2 \leq 3L_{gx}^2 \delta_k,
\tag{61}$$

*and*

$$
\begin{aligned}
\delta_k &\leq \iota^{2(m - m_0)} \left(\delta_{m(h-1)}\right) + \iota^{2m} \delta_{m(h-1)} + \iota^{2m} \delta_{m(h-2)} \\
&\quad + 12m^2 \lambda^2 \omega_\varphi \Big( \|\widetilde{\nabla}\varphi(x_{mh})\|^2 + \|\widetilde{\nabla}\varphi(x_{m(h-1)})\|^2 + \|\widetilde{\nabla}\varphi(x_{m(h-2)})\|^2 \\
&\quad + \sum_{t=m_0}^{j-1} \|\widetilde{\nabla}\varphi(x_{mh+t})\|^2 + \sum_{t=m_0}^{m-1} \|\widetilde{\nabla}\varphi(x_{m(h-1)+t})\|^2 + \sum_{t=m_0}^{m-1} \|\widetilde{\nabla}\varphi(x_{m(h-2)+t})\|^2 \Big),
\end{aligned}
\tag{62}
$$

*where $0 < \iota < 1$, $m_0 \sim \Omega(\log m)$ are constants, $m$ is the subspace dimension,*

$$\delta_k = \left(\frac{L_{fx}^2 \mu_g^2 + L_{gxy}^2 C_{fg}^2}{L_{gx}^2 \mu_g^2}\right) \|y_k - y_k^*\|^2 + \|v_k - v_k^*\|^2.$$

*Proof.* By (60), conclusion from Lemma G.3 and the Young's inequality, it holds that

$$\|v_k - v_k^*\|^2 \le 2 \left( \frac{L_{fy} + L_{gyy}C_v}{\mu_g} \right)^2 (\varepsilon_j^h)^2 + 2\frac{1}{\mu_g^2} \|r_k\|^2$$

$$\le 2 \left( \frac{L_{fy} + L_{gyy}C_v}{\mu_g} \right)^2 (\varepsilon_j^h)^2$$

$$+ 2\frac{1}{\mu_g^2} \left( 2\sqrt{\tilde{\kappa}(j)} \left( \frac{\sqrt{\tilde{\kappa}(j)}-1}{\sqrt{\tilde{\kappa}(j)}+1} \right)^j + \sqrt{j}L_{gyy}\varepsilon_j^h\tilde{\kappa}(j) \right)^2 \|r_{k0}\|^2$$

$$\le 2 \left( \frac{L_{fy} + L_{gyy}C_v}{\mu_g} \right)^2 (\varepsilon_j^h)^2$$

$$+ 4\frac{1}{\mu_g^2} \left( 4\tilde{\kappa}(j) \left( \frac{\sqrt{\tilde{\kappa}(j)}-1}{\sqrt{\tilde{\kappa}(j)}+1} \right)^{2j} + \left( \sqrt{j}\tilde{\kappa}(j)L_{gyy} \right)^2 (\varepsilon_j^h)^2 \right) \|r_{k0}\|^2 .$$

An estimate can be made for $\|r_{h+1}\|$,

$$\|r_{h+1}\| = \|b_{mh+1} - A_{mh+1}\bar{v}_h\|$$

$$= \|b_{mh+1} - b_{mh}^* + A_{mh}^* v_{mh}^* - A_{mh+1}v_{mh}\|$$

$$\le L_{fy} \|(x_{mh+1}, y_{mh+1}) - (x_{mh}, y_{mh}^*)\|$$

$$+ \frac{C_{fy}}{\mu_g}L_{gyy} \|(x_{mh+1}, y_{mh+1}) - (x_{mh}, y_{mh}^*)\| + L_{gy} \|v_{mh}^* - v_{mh}\|$$

$$= L_{gy} \|v_{mh} - v_{mh}^*\| + \frac{\mu_g L_{fy} + C_{fy}L_{gyy}}{\mu_g} \left( 1 + \frac{L_{gx}}{\mu_g} \right) \|x_{mh+1} - x_{mh}\| ,$$

and thus

$$\|v_k - v_k^*\|^2 \le \omega_\varepsilon (\varepsilon_j^h)^2 + \frac{L_{gy}^2}{\mu_g^2}\tilde{\kappa}(j) \left( \frac{\sqrt{\tilde{\kappa}(j)}-1}{\sqrt{\tilde{\kappa}(j)}+1} \right)^{2j} \|v_{mh} - v_{mh}^*\|^2$$

$$+ \frac{\tilde{\kappa}(j)}{\mu_g^2} \left( \frac{\mu_g L_{fy} + C_{fy}L_{gyy}}{\mu_g} \left( 1 + \frac{L_{gx}}{\mu_g} \right) \right)^2 \left( \frac{\sqrt{\tilde{\kappa}(j)}-1}{\sqrt{\tilde{\kappa}(j)}+1} \right)^{2j} \|x_{mh+1} - x_{mh}\|^2 ,$$

$$\tag{63}$$

where

$$\omega_\varepsilon = 2 \left( \frac{L_{fy} + L_{gyy}C_v}{\mu_g} \right)^2$$

$$+ \frac{4}{\mu_g^2} \left( \sqrt{m}\tilde{\kappa}(m)L_{gyy} \right)^2 \left( L_{gy}^2 C_v^2 + \left( \frac{\mu_g L_{fy} + C_{fy}L_{gyy}}{\mu_g} \left( 1 + \frac{L_{gx}}{\mu_g} \right) \right)^2 C_s^2 \right) \sim \mathcal{O}(m) .$$

By definition of $\varepsilon_j^h$, the update rule of $x$, and Young's inequality, it holds that

$$(\varepsilon_j^h)^2 \le 2m\lambda^2 \left( 1 + \frac{L_{gx}}{\mu_g} \right)^2 \sum_{i=1}^{j-1} \left\| \widetilde{\nabla}\varphi(x_{mh+i}) \right\|^2 + 2 \left\| y_{mh+i(j)} - y_{mh+i(j)}^* \right\|^2$$

for some $m_0 + 1 \leq i(j) \leq j$. Then we apply the descent property of $\|y_k - y_k^*\|$ (Lemma H.1) recursively to derive

$$
\begin{aligned}
\left(\varepsilon_j^h\right)^2 &\leq 2m\lambda^2 \left(1 + \frac{L_{gx}}{\mu_g}\right)^2 \sum_{i=1}^{j-1} \left\|\widetilde{\nabla}\varphi\left(x_{mh+i}\right)\right\|^2 + 2\left(1+\sigma\right)^{i(j)}\left(1-\theta\mu_g\right)^{i(j)} \left\|y_{mh} - y_{mh}^*\right\|^2 \\
&\quad + 2\left(1+\frac{1}{\sigma}\right)\left(1-\theta\mu_g\right)\left(\frac{L_{gx}}{\mu_g}\right)^2 \sum_{r=0}^{i(j)} \left(1+\sigma\right)^r\left(1-\theta\mu_g\right)^r \left\|x_{k-r} - x_{k-r-1}\right\|^2 \\
&\leq 2\lambda^2 \left(1 + \frac{L_{gx}}{\mu_g}\right)^2 \left(m + \left(1+\frac{1}{\sigma}\right)\left(1-\theta\mu_g\right)\right) \sum_{i=1}^{j-1} \left\|\widetilde{\nabla}\varphi\left(x_{mh+i}\right)\right\|^2 + 2\left\|y_{mh} - y_{mh}^*\right\|^2 \\
&\leq 2\lambda^2 \left(1 + \frac{L_{gx}}{\mu_g}\right)^2 \left(m + \left(1+\frac{1}{\sigma}\right)\left(1-\theta\mu_g\right)\right) \sum_{i=1}^{j-1} \left\|\widetilde{\nabla}\varphi\left(x_{mh+i}\right)\right\|^2 \\
&\quad + 2\left(1+\sigma\right)^{m_0}\left(1-\theta\mu_g\right)^{m_0} \left\|y_{mh} - y_{mh}^*\right\|^2 .
\end{aligned}
\tag{64}
$$

Since Lemma G.4 reveals that under the appropriate step-size setting $\lambda \sim \mathcal{O}(\frac{1}{m^4})$ and $\theta \sim \mathcal{O}(\frac{1}{m})$, $\tilde{\kappa}(j) \leq \tilde{\kappa}$ for $\tilde{\kappa} \sim \Omega(\frac{L_{gy}}{\mu_{gy}})$, we also choose $\sigma > 0$ such that

$$
\iota := \max\left\{\frac{\sqrt{\tilde{\kappa}} - 1}{\sqrt{\tilde{\kappa}} + 1}, \sqrt{\left(1+\sigma\right)\left(1-\theta\mu_g\right)}\right\} < 1
\tag{65}
$$

Additionally, set the warm-up steps $m_0$ to satisfy

$$
\iota^{-2m_0} \geq \max\left\{ \left(\frac{L_{fx}^2\mu_g^2 + L_{gxy}^2 C_{fg}^2}{L_{gx}^2\mu_g^2}\right)^{-1} \left(\frac{L_{fx}^2\mu_g^2 + L_{gxy}^2 C_{fg}^2}{L_{gx}^2\mu_g^2} + m\right), \right.
$$
$$
\left. \frac{L_{gy}^2}{\mu_g^2}\tilde{\kappa}\left(\frac{\mu_g L_{fy} + C_{fy}L_{gyy}}{\mu_g}\left(1 + \frac{L_{gx}}{\mu_g}\right)\right)^2, 2\omega_\varphi \right\},
$$

which means $m_0 \sim \Omega\left(\log m\right)$. In this manner, adding $\left(\frac{L_{fx}^2\mu_g^2 + L_{gxy}^2 C_{fg}^2}{L_{gx}^2\mu_g^2}\right)\|y_k - y_k^*\|^2$ on both sides of (63) and incorporating the estimation for $\epsilon_j^h$ (64) yield that

$$
\begin{aligned}
\delta_k &\leq \iota^{2(j-m_0)}\left(\delta_{mh}\right) + \left\|y_{mh} - y_{mh}^*\right\|^2 + 6m^2\lambda^2\omega_\varphi\left(\sum_{t=m_0}^{j-1} \left\|\widetilde{\nabla}\varphi\left(x_{mh+t}\right)\right\|^2 + \left\|\widetilde{\nabla}\varphi(x_{mh})\right\|^2\right) \\
&\leq \iota^{2(j-m_0)}\left(\delta_{mh}\right) + \iota^{2m}\left\|y_{m(h-1)} - y_{m(h-1)}^*\right\|^2 \\
&\quad + 6m^2\lambda^2\omega_\varphi\Big(\sum_{t=m_0}^{j-1} \|\widetilde{\nabla}\varphi(x_{mh+t})\|^2 + \|\widetilde{\nabla}\varphi(x_{mh})\|^2 \\
&\quad + \sum_{t=m_0}^{m-1} \|\widetilde{\nabla}\varphi(x_{m(h-1)+t})\|^2 + \|\widetilde{\nabla}\varphi(x_{m(h-1)})\|^2\Big),
\end{aligned}
\tag{66}
$$

where we apply Lemma H.1 recursively to obtain the second inequality, and adopt the notation

$$
\begin{aligned}
\omega_\varphi &:= \max\left\{\frac{\omega_\varepsilon}{m}\left(1 + \frac{L_{gx}}{\mu_g}\right)^2, \frac{\tilde{\kappa}\iota}{m^2\mu_g^2}\left(\frac{\mu_g L_{fy} + C_{fy}L_{gyy}}{\mu_g}\left(1 + \frac{L_{gx}}{\mu_g}\right)\right)^2, \right. \\
&\quad \left. \frac{1}{m^2}\left(1 + \frac{1}{\sigma}\right)\left(1-\theta\mu_g\right)\left(\frac{L_{fx}^2\mu_g^2 + L_{gxy}^2 C_{fg}^2}{L_{gx}^2\mu_g^2} + m\right)\left(\frac{L_{gx}}{\mu_g}\right)^2\right\} \sim \mathcal{O}\left(1\right).
\end{aligned}
$$

Consequently, expanding $\delta_{mh}$ in (66) in the same way derives (62). Regarding the upper-bound of the hyper-gradient estimation error, we have

$$\left\| \widetilde{\nabla}\varphi\left(x_k\right) - \nabla\varphi\left(x_k\right)\right\|^2 \le 3\left\|\nabla_x f\left(x_k, y_k\right) - \nabla_x f\left(x_k, y_k^*\right)\right\|^2 + 3\left\|\nabla_{xy}^2 g\left(x_k, y_k\right)\right\|^2 \left\|v_k - v_k^*\right\|^2$$

$$+ 3\left\|\nabla_{xy}^2 g\left(x_k, y_k\right) - \nabla_{xy}^2 g\left(x_k, y_k^*\right)\right\|^2 \left\|v_k^*\right\|^2$$

$$\le 3L_{fx}^2 \left\|y_k - y_k^*\right\|^2 + 3L_{gx}^2 \left\|v_k - v_k^*\right\|^2 + 3L_{gxy}^2 \left(\frac{C_{fy}}{\mu_g}\right)^2 \left\|y_k - y_k^*\right\|^2$$

$$= 3L_{gx}^2 \delta_k.$$

$\square$

**Theorem H.3.** *Suppose Assumptions 3.1, 3.2, 3.3, 3.6 and 3.9 hold. Within each epoch, if we set the step size $\theta \sim \mathcal{O}(1/m)$ a constant for $y$, and the step size for $x$ as zero in the first $m_0$ steps and the others as an appropriate constant $\lambda \sim \mathcal{O}(1/m^4)$, the iterates $\{x_k\}$ generated by Algorithm 2 satisfy*

$$\frac{m}{K\left(m - m_0\right)} \sum_{\substack{k=0, \\ (k\bmod m) > m_0}}^{K} \left\|\nabla\varphi\left(x_k\right)\right\|^2 = \mathcal{O}\left(\frac{m\lambda^{-1}}{K\left(m - m_0\right)}\right),$$

*where $m_0 \sim \Omega(\log m)$ is a constant and $m$ is the subspace dimension.*

*Proof.* Consider the Lyapunov function $\mathcal{L}_k := \varphi(x_k) + \delta_k$. According to Lemma E.2, a gradient descent step leads to the decrease in the hyper-function:

$$\varphi\left(x_{k+1}\right) - \varphi\left(x_k\right) \le \left\langle\nabla\varphi\left(x_k\right), x_{k+1} - x_k\right\rangle + \frac{L_\varphi}{2}\left\|x_{k+1} - x_k\right\|^2. \tag{67}$$

Then, telescoping $\mathcal{L}_{k+1} - \mathcal{L}_k$ over the index set $\mathcal{I} := \{k : 0 \le k \le K, (k\bmod m) > m_0\}$,

$$\sum_{k\in\mathcal{I}} \mathcal{L}_{k+1} - \mathcal{L}_k$$

$$= \sum_{k\in\mathcal{I}} \varphi(x_{k+1}) - \varphi(x_k) + \sum_{\substack{k=1, \\ ((k-1)\bmod m) > m_0}}^{K+1} \delta_k - \sum_{k\in\mathcal{I}} \delta_k$$

$$\overset{(i)}{\le} \sum_{k\in\mathcal{I}} \left\langle\nabla\varphi(x_k), x_{k+1} - x_k\right\rangle + \left(\frac{L_\varphi}{2} + 36m^3\omega_\varphi\right)\|x_{k+1} - x_k\|^2 + \sum_{e=0}^{h-1} 2m\iota^{2(m-m_0)}\delta_{me} - \sum_{k\in\mathcal{I}} \delta_k$$

$$\overset{(ii)}{\le} \sum_{k\in\mathcal{I}} -\left(\frac{\lambda}{2} - \lambda^2(L_\varphi + 72m^3\omega_\varphi)\right)\|\nabla\varphi(x_k)\|^2$$

$$+ \left(\frac{\lambda}{2} + \lambda^2(L_\varphi + 72m^3\omega_\varphi)\right)\|\nabla\varphi(x_k) - \widetilde{\nabla}\varphi(x_k)\|^2 + \sum_{e=0}^{h-1} 2m\iota^{2(m-m_0)}\delta_{me} - \sum_{k\in\mathcal{I}} \delta_k$$

$$\overset{(iii)}{\le} \sum_{k\in\mathcal{I}} -\left(\frac{\lambda}{2} - \lambda^2(L_\varphi + 72m^3\omega_\varphi)\right)\|\nabla\varphi(x_k)\|^2$$

$$+ 3L_{gx}^2\left(\frac{\lambda}{2} + \lambda^2(L_\varphi + 72m^3\omega_\varphi)\right)\sum_{k\in\mathcal{I}} \delta_k + \sum_{e=0}^{h-1} 2m\iota^{2(m-m_0)}\delta_{me} - \sum_{k\in\mathcal{I}} \delta_k, \tag{68}$$

where $(i)$ follows from inequalities (62) and (67); $(ii)$ results from the update rule $x_{k+1} = x_k - \lambda\widetilde{\nabla}\varphi(x_k)$ and Young's inequality; $(iii)$ comes from (61). Taking the coefficients of $\delta_k$ into account, we set the dimension parameters $(m, m_0)$ satisfying $m\iota^{2(m-m_0)} < 1/4$ and the step size $\lambda$ such that

$$\lambda \le \min\left\{\frac{1}{6L_{gx}^2}, \frac{1}{\left(12\left(L_\varphi + 72m^3\omega_\varphi\right)L_{gx}^2\right)^{1/2}}, \frac{1}{4\left(L_\varphi + 72m^3\omega_\varphi\right)}\right\}. \tag{69}$$

In this way, we obtain the following result from (68),

$$\mathcal{L}_{K+1} - \mathcal{L}_0 = \sum_{\substack{k=0, \\ (k \bmod m) > m_0}}^{K} \mathcal{L}_{k+1} - \mathcal{L}_k \leq \sum_{\substack{k=0, \\ (k \bmod m) > m_0}}^{K} - \left( \frac{\lambda}{2} - \lambda^2 \left( L_\varphi + 72 m^3 \omega_\varphi \right) \right) \| \nabla \varphi (x_k) \|^2.$$

Rearrange the above inequality and denote $\varphi^* := \min_{x \in \mathbb{R}^{d_x}} \varphi(x)$,

$$\sum_{\substack{k=0, \\ (k \bmod m) > m_0}}^{K} \| \nabla \varphi (x_k) \|^2 \leq \frac{4 \left( \varphi (x_0) - \varphi^* \right)}{\lambda} + \frac{4 \delta_0}{\lambda},$$

which completes the proof by dividing both sides by $\frac{m-m_0}{m} K$. □

# I    DETAILS ON EXPERIMENTS

## I.1    GENERAL SETTINGS

We conduct experiments to empirically validate the performance of the proposed algorithms. We test on a synthetic problem, a hyper-parameters selection task, and a data hyper-cleaning task. We compare the proposed SubBiO and LancBiO with the existing algorithms in bilevel optimization: stocBiO (Ji et al., 2021), AmIGO-GD and AmIGO-CG (Arbel and Mairal, 2022), SOBA (Dagréou et al., 2022) and TTSA (Hong et al., 2023), F2SA (Kwon et al., 2023) and HJFBiO (Huang, 2024). The experiments are produced on a workstation that consists of two Intel® Xeon® Gold 6330 CPUs (total $2{\times}28$ cores), 512GB RAM, and one NVIDIA A800 (80GB memory) GPU. The synthetic problem and the deep learning experiments are carried out on the CPUs and the GPU, respectively. For wider accessibility and application, we have made the code available on https://github.com/UCAS-YanYang/LancBiO.

For the proposed LancBiO, we initiate the subspace dimension at 1, and gradually increase it to $m = 10$ for the deep learning experiments and to $m = 80$ for the synthetic problem. For all the compared algorithms, we employ a grid search strategy to optimize the parameters. The optimal parameters yield the lowest loss. The experiment results are averaged over 10 runs. Note that Assumption 3.9 for initialization is not used in practice, for which we treat it as a theoretical assumption rather than incorporating it into Algorithm 2 in this paper.

In this paper, we consider the algorithms SubBiO and LancBiO in the deterministic scenario, so we initially compare them against the baseline algorithms with a full batch (i.e., deterministic gradient). In this setting, LancBiO yields favorable numerical results. Moreover, in the data hyper-cleaning task, to facilitate a more effective comparison with algorithms designed for stochastic applications, we implement all compared methods with a small batch size, finding that the proposed methods show competitive performance.

## I.2    DATA HYPER-CLEANING

The data hyper-cleaning task (Shaban et al., 2019), conducted on the MNIST dataset (LeCun et al., 1998), aims to train a classifier in a corruption scenario, where the labels of the training data are randomly altered to incorrect classification numbers at a certain probability $p$, referred to as the corruption rate. The task is formulated as follows,

$$\min_\lambda \quad \mathcal{L}_{val}(\lambda, w^*) := \frac{1}{|\mathcal{D}_{\text{val}}|} \sum_{(x_i, y_i) \in \mathcal{D}_{\text{val}}} L(w^* x_i, y_i)$$

$$\text{s.t.} \quad w^* = \arg\min_w \mathcal{L}_{tr}(w, \lambda)$$

$$:= \frac{1}{|\mathcal{D}_{\text{tr}}|} \sum_{(x_i, y_i) \in \mathcal{D}_{\text{tr}}} \sigma(\lambda_i) L(w x_i, y_i) + C_r \|w\|^2,$$

where $L(\cdot)$ is the cross-entropy loss, $\sigma(\cdot)$ is the sigmoid function, and $C_r$ is a regularization parameter. In addition, $w$ serves as a linear classifier and $\sigma(\lambda_i)$ can be viewed as the confidence of each data.

In the deterministic setting, where we implement all compared methods with full-batch, the training set, the validation set and the test set contain 5000, 5000 and 10000 samples, respectively. For algorithms that incorporate inner iterations to approximate $y^*$ or $v^*$, we select the inner iteration number from the set $\{5i \mid i = 1, 2, 3, 4\}$. The step size of inner iteration is selected from the set $\{0.01, 0.1, 1, 10\}$ and the step size of outer iteration is chosen from $\{5 \times 10^i \mid i = -3, -2, -1, 0, 1, 2, 3\}$. Regarding the Hessian/Jacobian-free algorithm HJFBiO, we set the step size $\delta = 1 \times 10^{-5}$ to implement finite difference methods. The results are presented in Figure 4. Note that LancBiO is crafted for approximating the Hessian inverse vector product $v^*$, while the solid methods stocBiO, TTSA, F2SA, and HJFBiO are not. Consequently, concerning the residual norm of the linear system, i.e., $\|A_k v_k - b_k\|$, we only compare the results with AmIGO-GD, AmIGO-CG and SOBA. Observe that the proposed subspace-based LancBiO achieves the lowest residual norm and the best test accuracy, and SubBiO is comparable to the other algorithms. Specifically, in Figure 4, the efficiency of LancBiO stems from its accurate approximation of the linear system. Furthermore, we implement the solvers designed for the stochastic setting using mini-batch to enable a broader comparison in Figure 8. It is shown that the stochastic algorithm SOBA tends to converge faster initially, but algorithms employing a full-batch approach achieve higher accuracy.

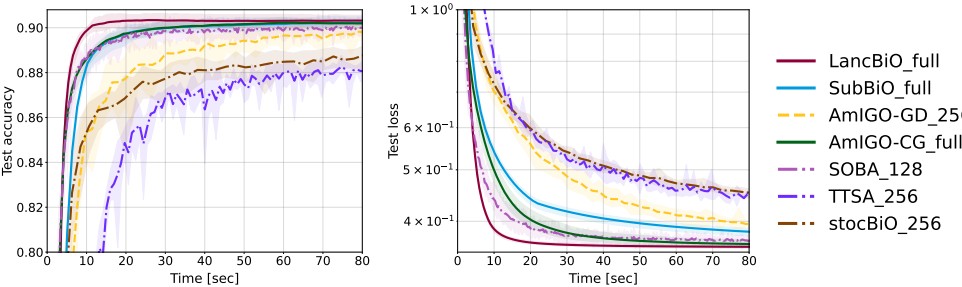

Figure 8: Comparison of the bilevel algorithms on data hyper-cleaning task with mini-batch when $p = 0.5$. The training set, the validation set and the test set contain 5000, 5000 and 10000 samples, respectively. The post-fix of legend represents the batch size. **Left:** test accuracy; **Right:**test loss.

To explore the potential for extending our proposed methods to a stochastic setting, we also conduct an experiment with stochastic gradients. In this setting, where we implement all compared methods with mini-batch, the training set, the validation set and the test set contain 20000, 5000 and 10000 samples, respectively. For algorithms that incorporate inner iterations to approximate $y^*$ or $v^*$, we select the inner iteration number from the set $\{3i \mid i = 1, 2, 3, 4\}$. The step size of inner iteration is selected from the set $\{0.01, 0.1, 1, 10\}$, the step size of outer iteration is chosen from $\{1 \times 10^i \mid i = -3, -2, -1, 0, 1, 2, 3\}$ and the batch size is picked from $\{32 \times 2^i \mid i = 0, 1, 2, 3\}$. AmIGO-CG is not presented since it fails in this experiment in our setting. The results in Figure 9 demonstrate that LancBiO maintains reasonable performance with stochastic gradients, exhibiting fast convergence rate, although the final convergence accuracy is slightly lower.

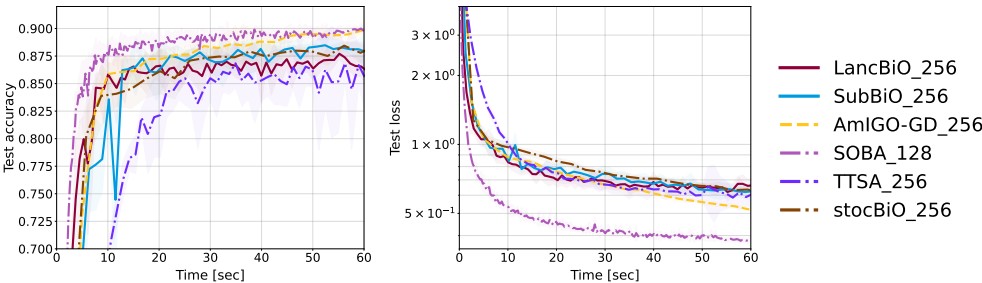

Figure 9: Comparison of the bilevel algorithms on data hyper-cleaning task with mini-batch when $p = 0.5$. The training set, the validation set and the test set contain 20000, 5000 and 10000 samples, respectively. The post-fix of legend represents the batch size. **Left:** test accuracy; **Right:**test loss.

Additionally, we also evaluate the performance of bilevel algorithms on Fashion-MNIST (Xiao et al., 2017) and Kuzushiji-MNIST (Clanuwat et al., 2018) datasets, both of which present more complexity compared to MNIST. Specifically, Fashion-MNIST serves as a modern replacement for MNIST, featuring grayscale images of clothing items across 10 categories, and Kuzushiji-MNIST is a culturally rich dataset of handwritten Japanese characters. The results, reported in Figures 10 and 11, reveal that LancBiO performs better than other algorithms and showcases robustness across various datasets, and SubBiO delivers a comparable convergence property.

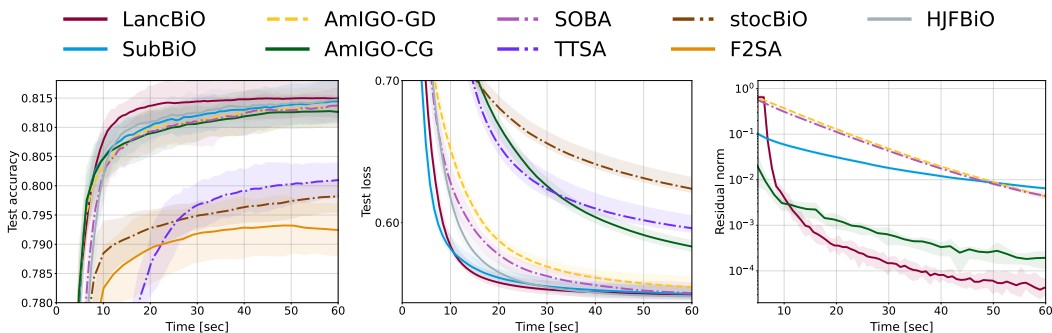

Figure 10: Data hyper-cleaning task tested on the Fashion-MNIST dataset when $p = 0.5$. **Left:** test accuracy; **Center**: test loss; **Right**: residual norm of the linear system, $\|A_k v_k - b_k\|$.

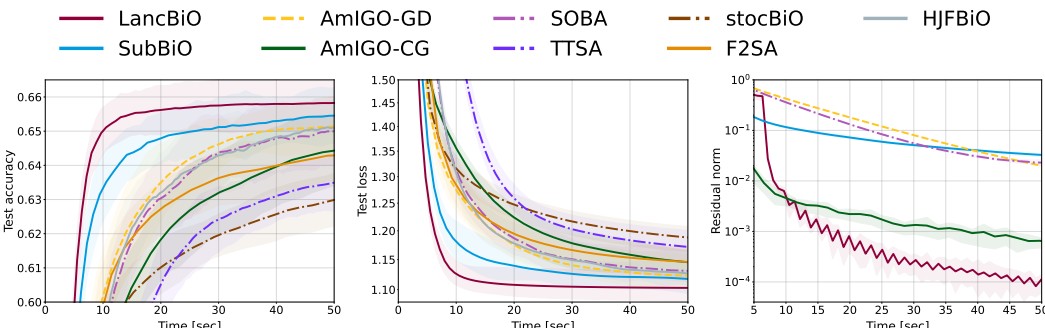

Figure 11: Data hyper-cleaning task tested on the Kuzushiji-MNIST dataset when $p = 0.6$. **Left:** test accuracy; **Center**: test loss; **Right**: residual norm of the linear system, $\|A_k v_k - b_k\|$.

### I.3 SYNTHETIC PROBLEMS

We concentrate on a synthetic scenario in bilevel optimization:

$$
\begin{aligned}
\min_{x \in \mathbb{R}^d} \quad & f(x, y^*) := c_1 \cos\left(x^\top D_1 y^*\right) + \frac{1}{2}\left\|D_2 x - y^*\right\|^2, \\
\text{s.t.} \quad & y^* = \arg\min_{y \in \mathbb{R}^d} g(x, y) \\
& := c_2 \sum_{i=1}^d \sin(x_i + y_i) + \log\left(\sum_{i=1}^d e^{x_i y_i}\right) + \frac{1}{2} y^\top \left(D_3 + G\right) y,
\end{aligned}
\tag{70}
$$

where we incorporate the trigonometric and log-sum-exp functions to enhance the complexity of the objective functions. In addition, we utilize the positive-definite matrix $G$ to ensure a strongly convex lower-level problem, and diagonal matrices $D_i$ $(i = 1, 2, 3)$ to control the condition numbers of both levels.

In this experiment, we set the problem dimension $d = 10^4$ and the constants $c_1 = 0.1$, $c_2 = 0.5$. $G$ is constructed randomly with $d$ eigenvalues from 1 to $10^5$. We generate entries of $D_1$, $D_2$ and $D_3$ from uniform distributions over the intervals $[-5, 5]$, $[0.1, 1.1]$ and $[0, 0.5]$, respectively. Taking into

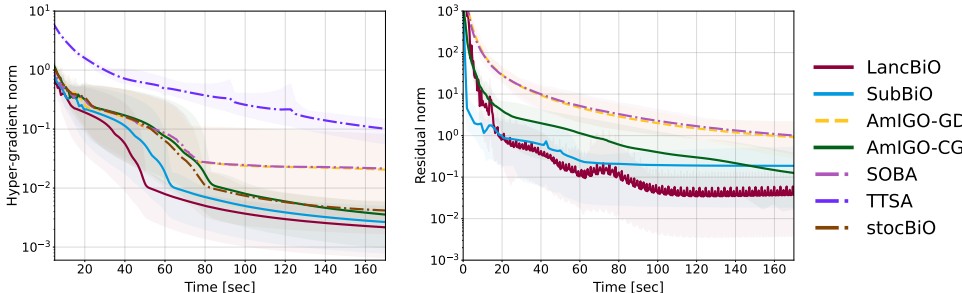

Figure 12: Comparison of the bilevel algorithms on the synthetic problem. **Left:** norm of the hyper-gradient; **Right**: residual norm of the linear system, $\|A_k v_k - b_k\|$.

account the condition numbers dominated by $D_i$ ($i = 1, 2, 3$) and $G$, we choose $\lambda = 1$ and $\theta = 10^{-5}$ for all algorithms compared after a manual search.

It can be seen from Figure 12 that LancBiO achieves the final accuracy the fastest, which benefits from the more accurate $v^*$ estimation. Figure 5 illustrates how variations in $m$ and $I$ influence the performance of LancBiO and AmIGO, tested across a range from 10 to 150 for $m$, and from 2 to 10 for $I$. For clarity, we set the seed of the experiment at 4, and present typical results to encapsulate the observed trends. It is observed that the increase of $m$ accelerates the decrease in the residual norm, thus achieving better convergence of the hyper-gradient, which aligns with the spirit of the classic Lanczos process.

When $m = 50$, the estimation of $v^*$ is sufficiently accurate to facilitate effective hyper-gradient convergence, which is demonstrated in Figure 5 that for $m \geq 50$, further increases in $m$ merely enhance the convergence of the residual norm. Under the same outer iterations, to attain a comparable convergence property, $I$ for AmIGO-CG should be set to 10. Furthermore, given that the number of Hessian-vector products averages at $(1 + 1/m)$ per outer iteration for LancBiO, whereas AmIGO requires $I \geq 2$, it follows that LancBiO is more efficient.

Table 2: Comparison on the synthetic problem (70). The dimension of problems is denoted by $d$. Results are averaged over 10 runs.

| Algorithm | $d = 10$ | | $d = 100$ | | $d = 1000$ | | $d = 10000$ | |
|---|---|---|---|---|---|---|---|---|
| | UL Val. | Time (S) | UL Val. | Time (S) | UL Val. | Time (S) | UL Val. | Time (S) |
| LancBiO | 4.52e−2 | 0.32 | 6.37e−2 | 0.53 | 5.29e−2 | 1.30 | −1.21e−2 | 16.36 |
| SubBiO | 3.73e−2 | 1.00 | 7.19e−2 | 1.17 | 4.63e−2 | 1.72 | −2.91e−2 | 21.26 |
| AmIGO-GD | 1.67e−1 | 2.44 | 1.05e−1 | 3.48 | 1.05e−1 | 1.46 | 4.96e−2 | 46.53 |
| AmIGO-CG | 5.56e−2 | 0.40 | 7.65e−2 | 1.90 | 5.73e−2 | 2.44 | 3.68e−2 | 25.00 |
| SOBA | 1.70e−1 | 0.60 | 1.28e−1 | 1.64 | 1.03e−1 | 2.52 | 3.49e−2 | 33.89 |
| TTSA | 5.66e−2 | 0.47 | 5.52e−2 | 0.89 | 6.52e−2 | 2.81 | 1.87e−1 | 121.07 |
| stocBiO | 6.24e−2 | 0.29 | 6.02e−2 | 0.45 | 5.21e−2 | 1.34 | −1.30e−2 | 20.66 |

Moreover, to illustrate how the proposed methods scale with increasing dimensions, we present the convergence time and the final upper-level (UL) value under different problem dimensions $d = 10^i, i = 1, 2, 3, 4$ in Table 2. The results show the robustness of the proposed methods across varying problem dimensions.

In addition, as discussed in Appendix D, to address the bilevel problem where the lower-level problem exhibits an indefinite Hessian, the framework LancBiO (Algorithm 2) requires a minor modification. Specifically, line 13 in Algorithm 2, which solves a small-size tridiagonal linear system, will be replaced by solving a low-dimensional least squares problem. We test the modified method LancBiO-MINRES on the following bilevel example borrowed from Liu et al. (2023a) with a non-convex

lower-level problem,

$$
\begin{aligned}
\min_{x \in \mathbb{R}} \quad & f(x, y^*) := \|x - a\|^2 + \|y^* - a - c\|^2 \\
\text{s.t.} \quad & y_i^* \in \arg\min_{y_i \in \mathbb{R}} \sin(x + y_i - c_i), \text{ for } i = 1, 2, \ldots, d,
\end{aligned}
\tag{71}
$$

where the subscript $i$ denotes the $i$-th component of a vector, while $a \in \mathbb{R}$ and $c \in \mathbb{R}^d$ are parameters. The results, reported in Figure 13, imply the potential of extending our work to non-convex scenarios.

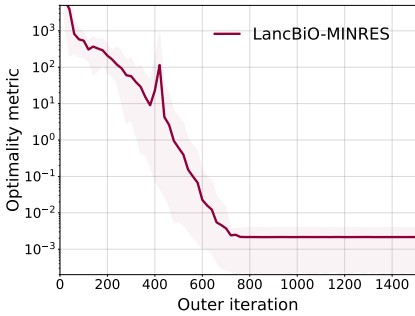

Figure 13: Test LancBiO-MINRES on the synthetic problem (71) with $d = 100$. The metric follows from the necessary conditions developed for bilevel problems without lower-level convexity in Theorem 1 of Xiao et al. (2023).

### I.4 LOGISTIC REGRESSION ON 20NEWSGROUP

Consider the hyper-parameters selection task on the 20Newsgroups dataset (Grazzi et al., 2020), which contains $c = 20$ topics with around $18000$ newsgroups posts represented in a feature space of dimension $l = 130107$. The goal is to simultaneously train a linear classifier $w$ and determine the optimal regularization parameter $\zeta$. The task is formulated as follows,

$$
\begin{aligned}
\min_{\lambda} \quad & \mathcal{L}_{val}(\zeta, w^*) := \frac{1}{|\mathcal{D}_{\text{val}}|} \sum_{(x_i, y_i) \in \mathcal{D}_{\text{val}}} L(w^* x_i, y_i) \\
\text{s.t.} \quad & w^* = \arg\min_{w} \mathcal{L}_{tr}(w, \zeta) \\
& := \frac{1}{|\mathcal{D}_{\text{tr}}|} \sum_{(x_i, y_i) \in \mathcal{D}_{\text{tr}}} L(w x_i, y_i) + \frac{1}{cl} \sum_{i=1}^{c} \sum_{j=1}^{l} \zeta_j^2 w_{ij}^2.
\end{aligned}
$$

where $L(\cdot)$ is the cross-entropy loss and $\{\zeta_j^2\}$ are the non-negative regularizers.

The experiment is implemented in the deterministic setting, where we implement all compared methods with full-batch, the training set, the validation set and the test set contain $5657$, $5657$ and $7532$ samples, respectively. For algorithms that incorporate inner iterations to approximate $y^*$ or $v^*$, we select the inner iteration number from the set $\{5i \mid i = 1, 2, 3, 4\}$. To guarantee the optimality condition of the lower-level problem, we adopt a decay strategy for the outer iteration step size, i.e., $\lambda_k = \lambda / k^{0.4}$, for all algorithms. The constant step size $\theta$ of inner iteration is selected from the set $\{0.01, 0.1, 1, 10\}$ and the initial step size $\lambda$ of outer iteration is chosen from $\{5 \times 10^i \mid i = -3, -2, -1, 0, 1, 2, 3\}$. The results are presented in Figure 6. In this setting, AmIGO-CG exhibits slightly better performance in reducing the residual norm. Nevertheless, under the same time, LancBiO implements more outer iterations to update $x$, which optimizes the hyper-function more efficiently.

