# OpenReview forum: "LancBiO: Dynamic Lanczos-aided Bilevel Optimization via Krylov Subspace"
_ICLR.cc/2025/Conference — ICLR 2025 Poster_

### Official Review · Reviewer_CLSW · 2024-11-01

**Soundness:** 2
**Presentation:** 3
**Contribution:** 2
**Rating:** 5
**Confidence:** 3

**Summary:**

This work proposes two novel algorithms, SubBiO and LancBiO, for solving bilevel optimization problems in which the lower-level problem is strongly convex. The main contribution lies in incorporating the Krylov subspace and the Lanczos process into bilevel optimization, achieving a more accurate hypergradient estimate by efficiently and dynamically solving the associated linear system. Under certain conditions, the authors establish non-asymptotic convergence for LancBiO and conduct an empirical study to validate the efficiency of both SubBiO and LancBiO.

**Strengths:**

$\textbf{S1:}$ The introduction of subspace techniques into bilevel optimization is well-motivated. The explanations of how the proposed SubBiO and LancBiO algorithms relate to existing methods are helpful.

$\textbf{S2:}$ A convergence rate guarantee is provided for LancBiO under certain conditions.

$\textbf{S3:}$ To empirically justify the improvement achieved by accurately solving the associated linear system, the experiments report the residual norm of the linear system.

**Weaknesses:**

For the theory, the main concerns are as follows:

$\textbf{W1: Assumption 3.10}.$ Based on (10), Assumption 3.10 depends on the step size, $\lambda$. However, in Lemma 3.11 and Theorem 3.12, $\lambda$ is specified to satisfy $\lambda\sim \mathcal{O}(\frac{1}{m^4})$. Under this setting, it seems that Assumption 3.10 may not hold when $m$ is sufficiently large. Therefore, in general, how can Assumption 3.10 be checked—either theoretically or empirically? A bit more discussion on these issues would be helpful.

$\textbf{W2: Theoretical results lack discussion}.$ In reading Theorem 3.12, it is unclear how the results compare to other bilevel algorithms, as some existing algorithms (e.g., deterministic SOBA) also reach an \epsilon-stationary point within $\mathcal{O}(\epsilon^{-1})$ outer iterations.
Additionally, there is no complexity analysis. For example, what is the number of oracle calls required to reach an $\epsilon$-stationary point? Typically, bilevel optimization literature provides the number of gradient, Hessian-vector, and Jacobian-vector products required to reach a stationary point with precision $\epsilon$.


$\textbf{W3: Lack of convergence analysis for SubBiO}.$ Given that SubBiO has a simpler structure than LancBiO, it would be beneficial to include a theoretical analysis or, if not feasible, to discuss the reasons for its absence.

For the experiments, the main concern is as follows:

$\textbf{W4: Experiments could be expanded}.$ The experiments could be more comprehensive. For example, it would be beneficial to include additional competing bilevel methods, especially Hessian-free algorithms like F2SA (Kwon et al., ICML 2023). Additionally, using more datasets in the data hyper-cleaning and logistic regression tasks would help validate the efficiency of both SubBiO and LancBiO.

**Questions:**

Apart from the questions raised in the Weaknesses section, some additional questions are as follows:

$\textbf{Q1:}$ In the implementation of SubBiO, how is the two-dimensional subproblem in line 4 solved? A bit more discussion on these choices would be helpful.

$\textbf{Q2:}$ In the convergence analysis of LancBiO, the hyperparameter $m_0$ plays an important role, but it is not included in the experiments. Could the authors clarify why?

$\textbf{Q3:}$ Can the authors provide more detail on why the Lanczos process in Algorithm 2 does not affect the final convergence guarantee? A brief proof sketch of Theorem 3.12 would be helpful.

Suggestions for improvement that did not affect the score:

$\textbf{About $m_0$ in Lemma 3.11 and Theorem 3.12:}$ First, $m_0$ in Lemma 3.11 should satisfy $m_0 = \Omega(1)$, meaning $m_0$ must be greater than some positive constant, as implied at the end of the proof of Lemma G.4. Second, $m_0$ in Theorem 3.12 should be set to $m_0 = \Omega(\log m)$, following from lines 1643–1647 and equation (63) in the proof of Lemma H.2.

---

> ### Author Response · Authors · 2024-11-23
>
> (Rebuttal part 1/3)
>
> ### **Overall**
> We would like to appreciate for all the time and effort the reviewer has spent on checking our proofs, along with the advice on the work.
>
> ### **Strengths**
> We are grateful for the reviewer's feedback on our work's clarity, motivation, theoretical grounding, and empirical performance.
>
> ### **Weaknesses**:
> 1. > W1: Assumption 3.10. Based on (10), Assumption 3.10 depends on the step size, $\lambda$. However, in Lemma 3.11 and Theorem 3.12, $\lambda$ is specified to satisfy $\lambda\sim\mathcal{O}(1/m^4)$. Under this setting, it seems that Assumption 3.10 may not hold when $m$ is sufficiently large. Therefore, in general, how can Assumption 3.10 be checked—either theoretically or empirically? A bit more discussion on these issues would be helpful.
>
>     Thanks to the reviewers’ comments, we are enlightened that **Assumption 3.10 can be circumvented** by a more detailed analysis. Specifically, in the original version, only the proof of Lemma H.2 resorts to Assumption 3.10. Therefore, in the updated version, we remove Assumption 3.10 and modify the proof of Lemma H.2 locally. The modification does not alter the convergence result of this paper.
>
>     In the updated PDF, the theoretical improvements are outlined in Appendix J to facilitate comparison with the original proof, and **they will be merged into Appendix H in the next version** of our paper.
>
> 2. > W2: Theoretical results lack discussion. In reading Theorem 3.12, it is unclear how the results compare to other bilevel algorithms, as some existing algorithms (e.g., deterministic SOBA) also reach an $\epsilon$-stationary point within $\mathcal{O}(\epsilon^{-1})$ outer iterations.
>
>     This is an interesting point, which inspires the following discussion.
>
>     Theoretically, we start by the **rationale for the better ability of LancBiO** to approximately solve linear systems.
>
>     Generally, consider the scenario of solving a standard linear system $Ax=b$ with positive-definite $A$, the condition number $\kappa:=\lambda_{max}(A)/\lambda_{min}(A)$ and the exact solution $x^*:=A^{-1}b$. The convergence properties of the Lanczos process and gradient descent have been well-studied.
>
>     Theorem 3.3 in [1] derives the convergence rate of the **gradient descent** method for solving symmetric linear systems,
>      $$
>      \frac{||x_j^{GD}-x^*||_A}{||x_0-x^*||_A} \le \left(\frac{\kappa-1}{\kappa+1}\right)^j.
>      $$
>      Section 1.3 in [2] discusses the convergence rate of the **Lanczos algorithm** for solving symmetric linear systems,
>      $$
>      \frac{||x_j^{Lanc}-x^*||_A}{||x_0-x^*||_A} \le 2\left(\frac{\sqrt{\kappa}-1}{\sqrt{\kappa}+1}\right)^j.
>      $$
>      Lanczos process performs better since $\frac{\sqrt{\kappa}-1}{\sqrt{\kappa}+1}<\frac{\kappa-1}{\kappa+1}$.
>
>     LancBiO, in a sense, **reflects this principle** within the context of bilevel optimization (see Lemma 3.11 in the paper), underscoring that the dynamic Lanczos-aided approach requires less effort to approximate the Hessian inverse vector product than compared methods like SOBA.
>
>     In bilevel optimization, regarding the proof of SOBA (see the latest arxiv version [3] with Appendix) , the convergence analysis relies on the descent property of a Lyapunov function $\mathcal{L}_k$, which **includes the term** $||v_k-v^*_k||$. It means the accuracy of solving the linear systems plays a crucial role in the convergence property. Similarly, for our LancBiO, the analysis for Lemma H.2 in the submission bounds $||\widetilde{\nabla}\varphi(x_k)-\nabla\varphi(x_k)||$ by $\delta_k$, which also **includes the term** $||v_k-v^*_k||$. Therefore, it holds that **the more descent of** $||v_k-v^*_k||$, **the better convergence of convergence behavior** of the algorithms.
>
>     Practically,  Figure 1 verifies that the higher quality of the estimate of $v^*$, the more enhanced descent of the objective function within the same number of outer iterations, which is one of the motivations of our work. The numerical experiments verify this point, e.g., in Figure 4, the proposed LancBiO outperforms SOBA by several orders of magnitude in terms of the residual norm, and thus LancBiO converges faster in loss and accuracy.
>
>     > Additionally, there is no complexity analysis. For example, what is the number of oracle calls required to reach an $\epsilon$-stationary point? Typically, bilevel optimization literature provides the number of gradient, Hessian-vector, and Jacobian-vector products required to reach a stationary point with precision $\epsilon$.
>
>     We discuss this point regarding the Algorithm 2. In each iteration, it invokes two gradient oracles (lines 2 and 16), one Hessian-vector product oracle (line 11), and one Jacobian-vector product (line 15).  Therefore, given the iteration complexity developed in Theorem 3.12, the proposed method requires $\mathcal{O}(\epsilon^{-1})$ invocations of each kinds of the mentioned oracles to reach a stationary point with precision $\epsilon$.

---

> ### Author Response · Authors · 2024-11-23
>
> (Rebuttal part 2/3)
>
> ### **Weaknesses**
>
> 3. > W3: Lack of convergence analysis for SubBiO. Given that SubBiO has a simpler structure than LancBiO, it would be beneficial to include a theoretical analysis or, if not feasible, to discuss the reasons for its absence.
>
>      We appreciate the reviewer's comments regarding the convergence analysis of SubBiO, which will help make our paper more rigorous. **As discussed in Section 2.3, SubBiO shares the same update rule as SOBA, except for the update of $v$**, which tracks the solution of $\min_{v} \frac{1}{2}v^\top A_kv-v^\top b_k$. Specifically, SOBA adopts the update rule as follows,
>      $$
>      v_{k}=v_{k-1}-\eta \left( A_{k}v_{k-1}-b_{k} \right)=(I-\eta A_k)v_{k-1} + \eta b_k,
>      $$
>      while the proposed SubBiO constructs a two-dimensional subspace, $S_{k}=\operatorname{span}<b_{k},(I-\eta A_{k}v_{k-1})>$. It is worth noting that the updated $v_k$ in SOBA belongs to the subspace $S_{k}$. Therefore, in the sense of solving the two-dimensional subproblem, **SubBiO selects the optimal solution** $v$ in the subspace.
>
>      Consequently, to give a convergence analysis for SubBiO, it remains to modifies the bound for $\||v_k-v^*_k\||$ in the proof for SOBA （see the analysis in [3]). Given the discussion above, a more straightforward manner is to use the bound $\||v_k^{SubBiO}-v^*_k\||\le\||v^{SOBA}_k-v^*_k\||$.
>
>      In our work, SubBiO serves as a motivation for LancBiO, and  **tackling the inherent instability of constructing Krylov subspaces** in bilevel optimization is the most novel theoretical part (see Appendix F and G). Therefore, we omit the detailed convergence analysis of SubBiO for simplicity of the presentation.
>
> 4. > W4: Experiments could be expanded. The experiments could be more comprehensive. For example, it would be beneficial to include additional competing bilevel methods, especially Hessian-free algorithms like F2SA (Kwon et al., ICML 2023). Additionally, using more datasets in the data hyper-cleaning and logistic regression tasks would help validate the efficiency of both SubBiO and LancBiO.
>
>     We sincerely thank the reviewer for the valuable suggestions regarding the scope of our experiments. We have **enriched the experiments** from the following aspects and made the **code publicly** available, which is attached to the submission.
>
>     (1) Two Hessian-free algorithms, F2SA [4] and HJFBiO [5], are evaluated in the experiments, providing a more comprehensive comparison.
>
>     (2) The data hyper-cleaning task is evaluated on two additional datasets, Fashion-MNIST and Kuzushiji-MNIST (see Figure 8 and Figure 9). The proposed LancBiO performs better than other algorithms and showcases robustness across various datasets.
>
>     (3) We test algorithms on a synthetic problem under different problem dimensions (see Table 2 for illustration).
>
> ### **Questions**
>
> 1. > Q1: In the implementation of SubBiO, how is the two-dimensional subproblem in line 4 solved? A bit more discussion on these choices would be helpful.
>
>     Consider the scenario where the upper-level variable $x\in\mathbb{R}^{d_x}$ and the lower-level variable $y\in\mathbb{R}^{d_y}$. As discussed in Section 2.2 in the PDF, solving the two-dimensional subproblem in line 4 of SubBiO reduces to solving
>      $$
>      \min_{z\in\mathbb{R}^2}\frac{1}{2}z^\top(S_k^\top A_kS_k)z - b_k^\top S_k z,
>      $$
>     where $S_k:=[b_k\ \left( I-\eta A_k  \right)v_{k-1}]\in\mathbb{R}^{d_y\times 2}$ is the basis matrix of the two-dimensional subspace. Therefore, the solution $z^*\in\mathbb{R}^2$ enjoys a closed-form expression $z^*=(S_k^\top A_k S_k)^{-1}(S_k^\top b_k)$, resulting in the variable $v_k=S_k z^*$. In fact, the complexity of computing the inverse of $S_k^\top A_k S_k\in\mathbb{R}^{2\times 2}$ is very small compared to the main computational bottleneck--- the two Hessian-vector products $S_k^\top A_k$ and $S_k^\top b_k$, which dominate the cost of this process.
>
> 2. > Q2: In the convergence analysis of LancBiO, the hyperparameter $m_0$ plays an important role, but it is not included in the experiments. Could the authors clarify why?
>
>     Theoretically, the $m_0$ in Theorem 3.12 is proposed to circumvent the inherent instability of constructing Krylov subspaces [6] in the context of bilevel optimization, which never appears in prior bilevel research.
>
>     However, in all numerical experiments, we set the warm-up steps $m_0=0$, i.e., the implementation of LancBiO behaves the same as single-loop algorithms. In this manner, LancBiO still shows efficient and stable results, as presented in Section 4 and Appendix I of the submission. Therefore, the configuration of $m_0$ functions theoretically, and it is omitted in the experiment discussion for simplicity. We have made all the **code publicly** available in the submission, and details can be verified there.

---

> ### Author Response · Authors · 2024-11-23
>
> (Rebuttal part 3/3)
>
> ### **Questions**
>
> 3. > Q3: Can the authors provide more detail on why the Lanczos process in Algorithm 2 does not affect the final convergence guarantee? A brief proof sketch of Theorem 3.12 would be helpful.
>
>     This is a good question, and we structure the answer into the following three steps.
>
>     (1) The proposed method, LancBiO, maintains an auxiliary variable $v_k$ to chase the Hessian inverse vector product $A_k^{-1}b_k$, as some existing methods do, e.g., SOBA.
>
>     (2) The dynamic Lanczos process in Algorithm 2, borrowing the intuition from the standard linear system (see Weakness 2), aims to obtain a "better" $v_k$ via the subspace technique. However, it may suffer from the inherent instability of constructing Krylov subspaces [6]
>
>     (3) Therefore, in Section 2.2, we propose two new strategies, restart mechanism and residual minimization, which serve as a cornerstone for the dynamic Lanczos process in Algorithm 2. In this manner, we establish an analysis framework that has never appeared in prior bilevel research to **tackle the inherent instability of constructing Krylov subspaces [6]** in bilevel optimization; see Appendix F and Appendix G. Consequently, final convergence results hold.
>
>     Thanks for your advice, we have added the proof sketch for Theorem 3.12 at the beginning of Appendix H.
>
> 4. > About $m_0$ in Lemma 3.11 and Theorem 3.12: First, $m_0$ in Lemma 3.11 should satisfy $m_0=\Omega(1)$, meaning $m_0$ must be greater than some positive constant, as implied at the end of the proof of Lemma G.4. Second, $m_0$ in Theorem 3.12 should be set to $m_0=\Omega(\log ⁡m)$, following from lines 1643–1647 and equation (63) in the proof of Lemma H.2.
>
>     We have made corresponding revisions based on the valuable advice, rendering the presentation more accurate.
>
> ### **Concluding remarks**
> Please respond to our post and let us know if the above clarifications adequately address your concerns about our work. We are happy to address any remaining points during the discussion phase; if the above responses are sufficient, we are grateful if you consider raising the grade.
>
> [1] Numerical optimization. 2006.
>
> [2] The Lanczos algorithm for solving symmetric linear systems. 1982.
>
> [3] A framework for bilevel optimization that enables stochastic and global variance reduction algorithms. arXiv preprint arXiv:2201.13409, 2022.
>
> [4] A fully first-order method for stochastic bilevel optimization.  ICML, 2023
>
> [5] Optimal Hessian/Jacobian-free nonconvex-PL bilevel optimization. ICML(oral), 2024
>
> [6] The Lanczos and conjugate gradient algorithms in finite precision arithmetic. Acta Numerica, 2006.

---

> ### Author Response · Authors · 2024-12-01
>
> Dear Reviewer CLSW
>
> We sincerely appreciate your great efforts in reviewing our work and your valuable comments. Your insights have been incredibly helpful in improving our submission.
>
> We would like to kindly follow up as we have not heard back from you since our last response. We are keen to know whether our responses have sufficiently addressed your concerns. If there are any remaining questions, please let us know, and we would be happy to provide further clarification.
>
> Best,
>
> Authors

---

### Official Review · Reviewer_nGhD · 2024-11-03

**Soundness:** 3
**Presentation:** 3
**Contribution:** 2
**Rating:** 5
**Confidence:** 3

**Summary:**

This work presents an approach for bilevel optimization using Krylov subspace methods and the Lanczos process to approximate inverse-Hessian vector products. The method constructs low-dimensional Krylov subspaces and solves tridiagonal linear systems, achieving convergence to an $\epsilon$-stationary point with $\mathcal{O}(\epsilon^{-1})$ complexity. Experimental evaluations are conducted to illustrate its performance.

**Strengths:**

The paper introduces a novel modification to the Lanczos process by re-linearising the objective functions within the iteration, specifically for bilevel optimization. This reduces large-scale subproblems to smaller tridiagonal linear systems, and the explanation provides a potential framework for applying subspace techniques in this context.

**Weaknesses:**

I identify two weaknesses:

1.To my knowledge, many recent advances in bilevel optimization have focused on addressing problems where the function is not necessarily strongly convex, or even non-convex. The strong convexity assumption in this paper may be overly stringent, potentially limiting the method's applicability to a broader range of optimization problems.

2.The proposed method is technically constrained to deterministic settings. While LancBiO can be extended to stochastic scenarios, its performance in these settings has been inconsistent. It remains uncertain whether LancBiO can be effectively adapted for stochastic environments, which is crucial for many practical applications. For example, despite including experiments in stochastic settings, SOBA seems to perform better, indicating challenges in extending LancBIO effectively.

**Questions:**

Could the authors provide further details on extending LancBiO to stochastic scenarios? Additionally, could they elaborate on how the Lanczos process ensures convergence in stochastic settings?

---

> ### Author Response · Authors · 2024-11-23
>
> ### **Overall**
> We would like to appreciate for all the time and effort the reviewer has spent on checking our submission, along with the valuable advice on the extension of our work.
>
> ### **Strengths**
> Thank you for your recognition of the novelty and contributions of our work.
>
> ### **Weaknesses**:
> We sincerely appreciate the reviewer's careful examination of our work.
>
> 1. > To my knowledge, many recent advances in bilevel optimization have focused on addressing problems where the function is not necessarily strongly convex, or even non-convex. The strong convexity assumption in this paper may be overly stringent, potentially limiting the method's applicability to a broader range of optimization problems.
>
>     This is an interesting point worth further discussion.
>
>     In addition to the specific algorithm LancBiO, **a main principle of this work is the dynamic Lanczos process in bilevel optimization**, which dynamically approximates the Hessian-inverse vector product $A_k^{-1}b_k$---a persistent computational bottleneck in recent advances in bilevel optimization, even when the lower-level problem is not strongly convex. To illustrate this, we take some recent related work as examples and provide a discussion.
>
>       - The lower-level problems in [1] and [2] are subject to linear constraints. Both equation (4) in [1] and equation (8) in [2] involve a term $A_k^{-1}b_k$, which play a crucial role in the update rule.
>       - The work [3] tackles the bilevel problem where the lower level is a constrained problem. Equation  (12) in it also involves a term $A_k^{-1}b_k$.
>       - The work [4] focuses on the lower-level problem that satisfies PL condition, and equation (8) in it involves a term $A_k^{-1}b_k$.
>       - When the lower level is a Riemannian optimization problem, equation (2) in [5] includes a term $A_k^{-1}b_k$ for updating variables.
>
>     Therefore, this work paves the way for adopting subspace techniques in future bilevel optimization research, both algorithmically and theoretically, and thus holds potential for broader applications.
>
> 2. > The proposed method is technically constrained to deterministic settings. While LancBiO can be extended to stochastic scenarios, its performance in these settings has been inconsistent. It remains uncertain whether LancBiO can be effectively adapted for stochastic environments, which is crucial for many practical applications. For example, despite including experiments in stochastic settings, SOBA seems to perform better, indicating challenges in extending LancBIO effectively.
>
>     We sincerely value the reviewer’s meticulous examination and  suggestions of our proposed framework, LancBiO. The detailed discussion is delayed to **Questions** part.
>
> ### **Questions**
>
> > Could the authors provide further details on extending LancBiO to stochastic scenarios? Additionally, could they elaborate on how the Lanczos process ensures convergence in stochastic settings?
>
>   Thanks for the reviewer's valuable advice and we recognize the importance of further adapting LancBiO to the stochastic setting for broader applications.
>
>   Theoretically, the main challenge of extending LancBiO to stochastic scenarios lies in the **coupled errors** introduced by stochastic gradients and those arising from the approximate dynamic subspaces. As revealed in Appendix F and Appendix G, we establish an analysis framework to tackle the inherent instability of constructing Krylov subspaces [6] in the context of bilevel optimization, which never appears in prior bilevel research.
>
>   Therefore, extending the proposed LancBiO to the stochastic version technically reduces to two steps:
>
>   1. By building our analysis on Appendix F and Appendix G for the dynamic Lanczos process and resorting to the recent analysis for the stochastic Lanczos process [7], we can develop a dynamic Lanczos process that accommodates stochastic oracles.
>   2. Translating existing literature on stochastic bilevel optimization, e.g., [8,9], we can bound the error of the stochastic hyper-gradient, and thus extending our work.
>
> ### **Concluding remark**
>
>   Please respond to our post and let us know if the above clarifications adequately address your concerns about our work. We are happy to address any remaining points during the discussion phase. If the above responses are sufficient, we appreciate your time reading our post and would appreciate it if you consider raising your score.

---

> ### Author Response · Authors · 2024-11-23
> **Reference**
>
> [1]  An implicit gradient-type method for linearly constrained bilevel problems. ICASSP, 2022
>
> [2] Alternating projected SGD for equality-constrained bilevel optimization. AISTATS, 2023
>
> [3] Double momentum method for lower-level constrained bilevel optimization. ICML, 2024
>
> [4] Optimal Hessian/Jacobian-free nonconvex-PL bilevel optimization. ICML(oral), 2024
>
> [5] A framework for bilevel optimization on Riemannian manifolds. NeurIPS, 2024
>
> [6] The Lanczos and conjugate gradient algorithms in finite precision arithmetic. Acta Numerica, 15:471–542, 2006.
>
> [7] Analysis of stochastic Lanczos quadrature for spectrum approximation. ICML, 2021
>
> [8] Bilevel optimization: convergence analysis and enhanced design. ICML, 2021.
>
> [9] A single-timescale method for stochastic bilevel optimization. AISTATS, 2022.

---

> > ### Comment · Reviewer_nGhD · 2024-11-28
> >
> > Thank you for the detailed response. The paper can benefit from a substantial revision before publishing. At this point, I still believe a score of 5 reflects the quality of this paper, therefore I will keep my score.

---

> ### Author Response · Authors · 2024-11-28
>
> Dear reviewer nGhD
>
> We would like to thank you again for all the time and effort you have spent on our submission, along with the prompt response. We have enriched experiments **following reviewers' advice** and removed Assumption 3.10 **with local modification** on the proof of Lemma H.2, in the current version of the PDF. We are also happy  to keep on addressing concerns in the remaining discussion phase.
>
> Best,
>
> Authors

---

### Official Review · Reviewer_H14N · 2024-11-03

**Soundness:** 3
**Presentation:** 3
**Contribution:** 3
**Rating:** 6
**Confidence:** 3

**Summary:**

The paper addresses the complexities of bilevel optimization, a framework widely used in machine learning applications. It introduces a novel approach that utilizes the Lanczos process to construct low-dimensional Krylov subspaces, aiming to alleviate the computational challenges associated with hypergradient calculations. By avoiding the direct computation of the Hessian inverse, the proposed method demonstrates improved efficiency and achieves a convergence rate of $ O(\epsilon^{-1}) $. The authors present a theoretical foundation for their approach and validate it through experiments on synthetic problems and deep learning tasks.

**Strengths:**

- **Innovative Approach**: The incorporation of subspace techniques into bilevel optimization is novel and contributes significantly to the field, potentially opening new avenues for research.
- **Theoretical Rigor**: The paper offers a solid theoretical framework that is well-justified, providing confidence in the proposed method's validity and effectiveness.
- **Empirical Validation**: The experimental results show promising performance improvements over existing methods, suggesting practical applicability in real-world scenarios.
- **Clarity of Presentation**: The paper is well-organized, making complex concepts accessible to readers, which enhances its impact.

**Weaknesses:**

- **Limited Scope of Experiments**: While the experimental results are promising, they are conducted on a limited set of problems. Broader validation across diverse benchmarks would strengthen the paper's claims.
- **Assumptions in Theory**: The theoretical results rely on certain assumptions that may not hold in all contexts, potentially limiting the generalizability of the findings.
- **Lack of Comparison with More Methods**: The paper could benefit from comparisons with additional state-of-the-art bilevel optimization methods to contextualize the improvements more clearly.

**Questions:**

1. How does the proposed method perform in scenarios where the lower-level problem is not strongly convex?
2. Can the authors elaborate on how the method scales with increasing dimensions in the bilevel optimization problems?
3. Are there any limitations observed when applying the proposed method to non-standard bilevel optimization problems, such as those with noisy or sparse data?
4. Could the authors provide more detailed insights into the computational complexity of solving the small-size tridiagonal linear system mentioned?

---

> ### Author Response · Authors · 2024-11-23
>
> (Rebuttal part 1/2)
>
> ### **Overall**:
> We appreciate all the time and effort the reviewer has spent on checking our submission, along with the advice on our work. Regarding the raised weaknesses, we have provided some appropriate explanations.
>
> ### **Strengths**:
> We are grateful for the reviewer's feedback on our work's novelty, originality, theoretical grounding, empirical performance, and clarity of presentation.
>
> ### **Weaknesses**:
> We sincerely appreciate the reviewer's careful examination of our work.
>
> 1. > Limited Scope of Experiments: While the experimental results are promising, they are conducted on a limited set of problems. Broader validation across diverse benchmarks would strengthen the paper's claims.
>
>     We sincerely thank the reviewer for the valuable suggestions regarding the scope of our experiments. We have enriched the experiments from the following aspects and made the **code publicly** available, which is attached to the submission.
>
>     - Two Hessian-free algorithms, F2SA [1] and HJFBiO [2], are evaluated in the experiments, providing a more comprehensive comparison.
>
>     - The data hyper-cleaning task is evaluated on two additional datasets, Fashion-MNIST and Kuzushiji-MNIST (see Figure 8 and Figure 9). The proposed LancBiO performs better than other algorithms and showcases robustness across various datasets.
>
>     - We test algorithms on a synthetic problem under different problem dimensions (see Table 2 for illustration).
>
> 2. > Assumptions in Theory: The theoretical results rely on certain assumptions that may not hold in all contexts, potentially limiting the generalizability of the findings.
>
>     Thanks to the reviewers’ comments, we are enlightened that **Assumption 3.10 can be circumvented** by a more detailed analysis. Specifically, in the original version, only the proof of Lemma H.2 resorts to Assumption 3.10. Therefore, in the updated version, we remove Assumption 3.10 and modify the proof of Lemma H.2 locally. The modification does not alter the convergence result of this paper.
>
>     In the updated PDF, the theoretical improvements are outlined in Appendix J to facilitate comparison with the original proof, and **they will be merged into Appendix H in the next version** of our paper.
>
> 3. > Lack of Comparison with More Methods: The paper could benefit from comparisons with additional state-of-the-art bilevel optimization methods to contextualize the improvements more clearly.
>
>     See Weakness 1.
>
> ### **Questions**
>
> 1. > How does the proposed method perform in scenarios where the lower-level problem is not strongly convex?
>
>     This is an interesting point worth further discussion.
>
>     The Lanczos process is known for its efficiency of constructing Krylov subspaces and is capable of solving indefinite linear systems [3]. Heuristically, its counterpart in the context of bilevel optimization, LancBiO also holds the potential. **As discussed in Appendix D**, to address the bilevel problem where the lower-level problem exhibits an indefinite Hessian, the framework LancBiO (Algorithm 2) requires a minor modification. Specifically, line 13 in Algorithm 2, which solves a small-size tridiagonal linear system, will be replaced by solving a low-dimensional least squares problem. We test the modified method LancBiO-MINRES on a numerical example borrowed from [4]. **The result is reported in Figure 12** in Appendix I, which implies the potential of extending our work to non-convex scenarios.
>
>     In addition to the specific algorithm LancBiO, **a main principle of this work is the dynamic Lanczos process in bilevel optimization**, which dynamically approximates the Hessian-inverse vector product $A_k^{-1}b_k$---a persistent computational bottleneck in recent advances in bilevel optimization, even when the lower-level problem is not strongly convex. This work paves the way for adopting subspace techniques in future bilevel optimization research, both algorithmically and theoretically, and thus holds potential for broader applications.
>
> 2. > Can the authors elaborate on how the method scales with increasing dimensions in the bilevel optimization problems?
>
>     **Theoretically**, our proof of Theorem 3.12 demonstrates that the iteration complexity of LancBiO is independent of the problem dimension, aligning with existing bilevel methods. The additional complexity introduced by higher dimensions arises primarily from vector and matrix operations.
>
>     **Numerically**, we test algorithms on a synthetic problem, illustrating how the method scales with increasing dimensions. **The results, outlined in Table 2**, showcase the robustness of the proposed LancBiO across varying problem dimensions.

---

> ### Author Response · Authors · 2024-11-23
>
> (Rebuttal part 2/2)
>
> ### **Questions**
>
> 3. > Are there any limitations observed when applying the proposed method to non-standard bilevel optimization problems, such as those with noisy or sparse data?
>
>     In the submission, we conduct a stochastic version of the data hyper-cleaning task (see Figure 7 in Appendix I). The results demonstrate that LancBiO maintains reasonable performance with mini-batches, exhibiting a fast convergence rate. However, the final convergence accuracy is slightly lower. Theoretically, tackling the errors introduced by (stochastic) noise and approximate subspaces simultaneously will be a central focus of our future work.
>
>     Additionally, if the problem exhibits a sparse structure---e.g., the gradient $\nabla f, \nabla g$ and the Hessian/Jacobian $\nabla^2 f,\nabla^2g,$ are sparse---then Algorithm 2 indicates that the intermediate variables, e.g., $Q_k$ and $w_k$, will also be sparse, as their computation primarily involves sparse matrix/vector multiplications and additions.
>
> 4. > Could the authors provide more detailed insights into the computational complexity of solving the small-size tridiagonal linear system mentioned?
>
>     Consider the scenario where the upper-level variable $x\in\mathbb{R}^{d_x}$ and the lower-level variable $y\in\mathbb{R}^{d_y}$. Then line 13 in Algorithm 2, $\Delta v_k=Q_k(T_k)^{-1}Q^\top_k r_k$, corresponds to solving the small-size tridiagonal linear system. Denoting $j=k\ \texttt{mod}\ m$, we have the subspace basis $Q_k\in\mathbb{R}^{d_y\times j}$, the tridiagonal matrix  $T_k\in\mathbb{R}^{j\times j}$, and the residual $r_k\in\mathbb{R}^{d_y\times 1}$. Therefore, the matrix-vector multiplications $Q^\top_k [r_k]$ and $Q_k[(T_k)^{-1}Q^\top_k r_k]$ cost $\mathcal{O}(jd_y)$. The operation $(T_k)^{-1}[Q^\top_k r_k]$ essentially tackles the small-size tridiagonal linear system $T_k z = Q^\top_k r_k$, which can be exactly solved by the famous Thomas algorithm (see https://en.wikipedia.org/wiki/Tridiagonal_matrix_algorithm). This process costs $\mathcal{O}(j)$. In summary, the complexity of line 13 in Algorithm 2 is $\mathcal{O}(jd_y)$, which is very small compared to the main computational bottleneck, i.e., large-scale matrix-vector multiplications.
>
> ### **Concluding remarks**
>
>   Please respond to our post and let us know if the above clarifications adequately address some of your concerns about our work. We are happy to address any remaining points during the discussion phase; if the above responses are sufficient, we appreciate your time reading our post and are grateful if you consider raising your score
>
>
> [1] A fully first-order method for stochastic bilevel optimization.  ICML, 2023
>
> [2] Optimal Hessian/Jacobian-free nonconvex-PL bilevel optimization. ICML(oral), 2024
>
> [3] On solving indefinite symmetric linear systems by means of the Lanczos method. Comput. Math. Math. Phys, 1999
>
> [4] Value-function-based sequential minimization for bi-level optimization. TPAMI, 2023

---

> ### Author Response · Authors · 2024-12-01
>
> Dear Reviewer H14N
>
> We sincerely appreciate your great efforts in reviewing our work and your valuable comments. Your insights have been incredibly helpful in improving our submission.
>
> In the rebuttal, we have made every effort to address your concerns by providing thorough responses and revising the submission based on the feedback. Could you kindly let us know whether there are any remaining questions? We would be happy to provide further clarification.
>
> Best,
>
> Authors

---

### Official Review · Reviewer_sZwj · 2024-11-04

**Soundness:** 3
**Presentation:** 2
**Contribution:** 2
**Rating:** 5
**Confidence:** 4

**Summary:**

This paper investigates a gradient based method for solving bilevel optimization problems with a strongly convex lower-level objective. To enhance the efficiency of hypergradient computation, the paper employs the Krylov subspace method alongside the Lanczos process to accelerate the solution of the linear systems involved. The paper presents non-asymptotic convergence results for the proposed method. Numerical experiments on data hyper-cleaning, synthetic problems, and logistic regression demonstrate its effectiveness.

**Strengths:**

1. This work leverages the Krylov subspace method and the Lanczos process to solve the linear system involved in hypergradient computation, providing a new approach for efficiently approximating the hypergradient.

2. A non-asymptotic convergence rate of $O(\epsilon^{-1})$ is established.

**Weaknesses:**

1. The description of Algorithm 2 is somewhat unclear, particularly in relation to the theoretical convergence result in Theorem 3.12. Theorem 3.12 requires the step size for $x$ to be zero during the initial few steps in each epoch, implying that before updating $x$, the algorithm must obtain a $y_k$ sufficiently close to $y^*(x_k)$, as also reflected in the theoretical analysis. This indicates that the epoch $h$ functions primarily as an outer loop index, where at each iteration $h$, the algorithm first solves the lower level problem to obtain a $y_k$ that is sufficiently close to the lower level solution before proceeding to update $x$. However, in other gradient based algorithms, such as SOBA, this requirement for $y_k$ to be close to the lower-level solution is not imposed.

2. In the convergence analysis, the proof of Theorem 3.12 , the step size for $x$ must be very small to ensure stability of the Krylov subspace method and the dynamic Lanczos process in LancBiO as $x_k$ is updated. It appears that the inclusion of the dynamic Lanczos process constrains the step size for $x$, potentially making it smaller than in algorithms without this process, such as SOBA. However, the paper lacks a discussion of this issue.

3. The parameter $m$, which defines the dimension of the Krylov subspace in LancBiO, should play a crucial role in the efficiency of the Krylov subspace method and hence the performance of LancBiO. However, the convergence analysis does not show how the choice of $m$ impacts the performance of LancBiO.

4. The convergence analysis requires an unusual assumption, Assumption 3.10, which is not present in other works, such as SOBA. Why is Assumption 3.10 necessary? Is it because $y_k$ generated by LancBiO cannot be guaranteed to converge to $y^*_k$ ?

**Questions:**

Is the proposed algorithm applicable in a stochastic setting?

---

> ### Author Response · Authors · 2024-11-23
>
> (Rebuttal part 1/2)
>
> ### **Overall**
> We appreciate the time and effort the reviewer has spent on our submission. We provide individual explanations for the questions raised.
>
> ### **Strengths**
> Thanks for the recognition of our work's novelty and theoretical grounding.
>
> ### **Weaknesses and Questions**
> We are grateful for the reviewer's thorough examination of our work.
>
> 1. > The description of Algorithm 2 is somewhat unclear, particularly in relation to the theoretical convergence result in Theorem 3.12. Theorem 3.12 requires the step size for x to be zero during the initial few steps in each epoch, implying that before updating $x$, the algorithm must obtain a $y_k$ sufficiently close to $y^∗(x_k)$, as also reflected in the theoretical analysis. This indicates that the epoch $h$ functions primarily as an outer loop index, where at each iteration $h$, the algorithm first solves the lower level problem to obtain a $y_k$ that is sufficiently close to the lower level solution before proceeding to update $x$. However, in other gradient based algorithms, such as SOBA, this requirement for $y_k$ to be close to the lower-level solution is not imposed.
>
>     Thanks for your accurate insight that $h$ functions like an outer loop index. It is admitted that the proposed LancBiO behaves slightly differently from purely single-loop methods,  such as SOBA.
>
>     **From the theoretical perspective**, to guarantee convergence, Theorem 3.12 requires that each epoch begins by updating $y$ for several steps and then updates $x$ and $y$ alternatively. Specifically, we establish an analysis framework to tackle the inherent instability of constructing Krylov subspaces [1] in the context of bilevel optimization, which never appears in prior bilevel research. Therefore, the mechanism of LancBiO can be seen as a trade-off between single- and double-loop frameworks.
>
>     **Moreover, in all numerical experiments**, we set the warm-up steps $m_0=0$, i.e., the implementation of LancBiO behaves similarly to single-loop algorithms. In this manner, LancBiO shows efficient and stable results, as presented in Section 4 and Appendix I of the submission. We have made all the **code publicly** available in the submission, and more details can be seen there.
>
>     In summary, we adopt the indices $h$ and $k$ in Algorithm 2 for clarity, which is compatible with the parameter configuration in theoretical analysis but also supports a practical implementation of the proposed algorithm.
>
> 2. > In the convergence analysis, the proof of Theorem 3.12, the step size for $x$ must be very small to ensure stability of the Krylov subspace method and the dynamic Lanczos process in LancBiO as $x_k$ is updated. It appears that the inclusion of the dynamic Lanczos process constrains the step size for $x$, potentially making it smaller than in algorithms without this process, such as SOBA. However, the paper lacks a discussion of this issue.
>
>     Theoretically, Theorem 3.12 suggests that $m$ can be in the order of $\mathcal{O}(1)$, and adopting a moderate step size $\lambda=\mathcal{O}(1/m^4)$ guarantees the stability of the dynamic Lanczos process. The specific inequalities $\lambda$ should obey are specified in the proof. When turning to SOBA  (see the latest Arxiv version [2] with Appendix), its step sizes are required to satisfy (98) and (99) in the Arxiv manuscript, which are complicated assemblies of various problem parameters, e.g., different Lipschitz constants. Therefore, it seems unclear to compare step sizes directly in theory.
>
>     Practically, for LancBiO, the step sizes $\lambda$ and $\theta$ are set as constants in the numerical experiments, with the parameter configuration summarized as follows. Specifically, the step sizes are in the same order as the other compared methods, which implies the requirement imposed by the dynamic Lanczos process on the step sizes is not strict in practice. We have made all the **code publicly** available in the submission, and the details can be verified there.
>     |    Experiment     | Synthetic problem | Hyper-parameters selection (full-batch) | Data hyper-cleaning (full-batch) | Data hyper-cleaning (mini-batch） |
>     | :-: | :-: | :--: | :------------------------------: | :-----------------------------: |
>     | $\lambda$ for $x$ |         1         |                   100                   |               500                |               100               |
>     | $\theta$ for $y$  |      0.00001      |                   100                   |               0.1                |               0.1               |

---

> ### Author Response · Authors · 2024-11-23
>
> (Rebuttal part 2/2)
>
> ### **Weaknesses and Questions**
>
> 3. > The parameter $m$, which defines the dimension of the Krylov subspace in LancBiO, should play a crucial role in the efficiency of the Krylov subspace method and hence the performance of LancBiO. However, the convergence analysis does not show how the choice of $m$ impacts the performance of LancBiO.
>
>     Theoretically, to guarantee the convergence, Theorem 3.13 suggests to employ a moderate step size $\lambda=\mathcal{O}(1/m^4)$. Taking it into the equation in line 426 of the submission specifies the influence of $m$ on the overall convergence, i.e., $\frac{1}{K}\sum_{k}||\nabla \phi(x_k)||^2=\mathcal{O}(\frac{m^4}{K})$.
>
>     In numerical experiments, we set the subspace dimension $m$ as $1$ during the initial several outer iterations. We then progressively increase this dimension to a constant. The setting is listed in the following table. Additionally, the influence of $m$ on the performance of the proposed LancBiO is visualized in Figure 8 and Figure 9. We have made all the **code publicly** available in the submission; more details can be seen there.
>
>     | Experiment | Synthetic problem | Hyper-parameters selection (full-batch) | Data hyper-cleaning (full-batch) | Data hyper-cleaning (mini-batch) |
>     | :--------: | :---------------: | :-------------------------------------: | -------------------------------- | -------------------------------- |
>     |     m      |        80         |                   10                    | 10                               | 50                               |
>
>  4. > The convergence analysis requires an unusual assumption, Assumption 3.10, which is not present in other works, such as SOBA. Why is Assumption 3.10 necessary? Is it because $y_k$ generated by LancBiO cannot be guaranteed to converge to $y_k^∗$ ?
>
>     Originally, Assumption 3.10 was put forward specifically to exploit subspace techniques in bilevel optimization. We establish an analysis framework that has never appeared in prior bilevel research to **tackle the inherent instability of constructing Krylov subspaces [1]** in bilevel optimization; see Appendix F and Appendix G.
>
>     Thanks to the reviewers’ comments, we are enlightened that **Assumption 3.10 can be circumvented** by a more detailed analysis. Specifically, in the original version, only the proof of Lemma H.2 resorts to Assumption 3.10. Therefore, in the updated version, we remove Assumption 3.10 and modify the proof of Lemma H.2 locally. The modification does not alter the convergence result of this paper.
>
>     In the updated PDF, the theoretical improvements are outlined in Appendix J to facilitate comparison with the original proof, and they will be merged into Appendix H in the next version of our paper.
>
> ### **Questions**
> > Is the proposed algorithm applicable in a stochastic setting?
>
>   In the submission, we conduct a stochastic version of the data hyper-cleaning task (see Figure 7 in Appendix I). The results demonstrate that LancBiO maintains reasonable performance with mini-batches, exhibiting a fast convergence rate, though the final convergence accuracy is slightly lower.
>
>   Extending our approach to accommodate stochastic gradients will be a central focus of our future work.
>
> ### **Concluding remarks**
> We are happy to address any remaining points during the discussion phase; if the above responses are sufficient, we appreciate your time reading our post and would be grateful if you consider raising the score.
>
> [1] The Lanczos and conjugate gradient algorithms in finite precision arithmetic. Acta Numerica.
>
> [2] A framework for bilevel optimization that enables stochastic and global variance reduction algorithms. arXiv preprint arXiv:2201.13409, 2022.

---

> ### Author Response · Authors · 2024-12-01
>
> Dear Reviewer sZwj
>
> We sincerely appreciate your great efforts in reviewing our work and your valuable comments. Your insights have been incredibly helpful in improving our submission.
>
> We would like to kindly follow up as we have not heard back from you since our last response. We are keen to know whether our responses have sufficiently addressed your concerns. If there are any remaining questions, please let us know, and we would be happy to provide further clarification.
>
> Best,
>
> Authors

---

> > ### Comment · Reviewer_sZwj · 2024-12-02
> >
> > Thank you to the authors for their rebuttal. After going through the response, I noticed a big gap between the theoretical convergence analysis and the implementation of the proposed method in the numerical experiments. Specifically, the authors' convergence analysis requires warm-up steps $m_0> 0$. However, as the authors’ metntion that  in all their experiments, they set $m_0 = 0$. This discrepancy implies that the method implemented in the numerical experiments does not align with the one analyzed theoretically, and the convergence results derived in the analysis do not directly support the implementation used in the experiments.
> >
> > There’s also a mismatch regarding the role of $m$. The authors explain that the overall convergence complexity is $O(\frac{m^4}{K})$ which suggests that
> > $m = 1$ should give the lowest complexity. However, $m = 1$ corresponds to a Krylov subspace dimension of 1, which is generally of limited interest.  Moreover, the numerical experiments suggest that larger values of $m$ tend to improve performance, which doesn’t align with the theoretical findings.
> >
> > In summary, there appears to be a big disconnect between the theoretical analysis and the numerical experiments. It is not clear which algorithm the authors intend to present: the one analyzed in the convergence theory or the one implemented in the experiments. I will keep my original score.

---

> > > ### Author Response · Authors · 2024-12-04
> > >
> > > We would like to thank you for your time and effort dedicated to reviewing our submission, as well as for your insightful comments. The theoretical analysis establishes an upper bound for the hyperparameter $m_0$, and the practical implementations validate performance when setting $m_0=0$. More importantly, your critique motivates us to further enhance the overall convergence results, particularly the dependency on the subspace dimension $m$. Through a more detailed analysis, we find that introducing the subspace technique reduces the oracle complexity. Here, the oracle refers to the gradient oracle, Hessian-vector product oracle, and Jacobian-vector product oracle.
> > >
> > > Thank you again for your valuable reviews, which will inspire us to further revise and improve the paper.

---

### Author Response · Authors · 2024-11-23
**General Response**

We sincerely thank all the reviewers for their insightful critiques and valuable suggestions regarding our manuscript. We thank all the reviewers for acknowledging our contributions and finding our work novel. We are the first to introduce subspace techniques to bilevel optimization, which is provable and of great potential. We have responded to the reviews individually and **updated a new manuscript version**. Here, we provide a general summary of our improvements to enhance the work.

- We have **enriched the experiments** from the following aspects and made the **code publicly** available, which is attached to the submission.
    1. Two Hessian-free algorithms, F2SA [1] and HJFBiO [2], are evaluated in the experiments, providing a more comprehensive comparison.
    2. The data hyper-cleaning task is evaluated on two additional datasets, Fashion-MNIST and Kuzushiji-MNIST (see Figure 8 and Figure 9). The proposed LancBiO performs better than other algorithms and showcases robustness across various datasets.
    3. We test algorithms on a synthetic problem under different problem dimensions (see Table 2 for illustration).

- Thanks to the reviewers’ comments, we are enlightened that **Assumption 3.10 can be circumvented** by a more detailed analysis. Specifically, in the original version, **only the proof of Lemma H.2** resorts to Assumption 3.10. Therefore, in the updated version, we remove Assumption 3.10 and modify the proof of Lemma H.2 locally. The modification does not alter the convergence result of this paper.

In the updated PDF, the modified content is highlighted in blue for easier identification. Specifically, the extension on experiments is recorded in **Appendix I**. The theoretical improvements are outlined in **Appendix J** to facilitate comparison with the original proof, and they will be merged into Appendix H in the next version of our paper.

[1] A fully first-order method for stochastic bilevel optimization.  ICML, 2023

[2] Optimal Hessian/Jacobian-free nonconvex-PL bilevel optimization. ICML(oral), 2024

---

> ### Author Response · Authors · 2024-11-28
>
> Dear all
>
> As the deadline for uploading the revised PDF approaches, we have **merged the modified proof** without the original Assumption 3.10 into Appendix H **in the updated PDF.** We recommend that reviewers and chairs who are interested in comparing this with the original proof refer to the previous version (in the Revisions history), where the theoretical improvements are outlined step-by-step in Appendix J.
>
> Best,
>
> Authors

---

### Meta-Review · Area_Chair_XfXj · 2024-12-20

**Metareview:**

This paper investigates bilevel optimization problems with a strongly convex lower-level objective, introducing novel algorithms, SubBiO and LancBiO, that leverage Krylov subspace methods and the Lanczos process to efficiently approximate hypergradients without directly computing the Hessian inverse. The proposed methods construct low-dimensional Krylov subspaces to solve tridiagonal linear systems, achieving improved efficiency and convergence to stationary points with theoretical non-asymptotic guarantees. Empirical evaluations on synthetic problems, data hyper-cleaning, and logistic regression tasks validate the methods’ effectiveness, demonstrating their potential for applications in machine learning.

**Additional Comments On Reviewer Discussion:**

The reviewers raised issues about the experiments and the presentation of the paper. I feel that the authors did a good job in the rebuttal.

Unfortunately, the topic of this paper may stray too far from core machine learning topics into optimization, scientific computing, and applied linear algebra. I think the paper is solid, as I also come from a similar background, but I take the reviewers' ambivalence as a negative signal.

---

### Decision · Program_Chairs · 2025-01-22

Accept (Poster)